# MoRE: Mixture of Remapping Experts for Irreversible Feature-Level Unlearning

## Abstract

Machine unlearning (MU) has emerged as a critical paradigm for enabling models to erase unwanted knowledge, not only at the instance level but also across entire concepts or classes, thereby addressing broader concerns of privacy, safety, and robustness. However, existing approaches face three persistent challenges: (1) they often degrade utility on the remain data, (2) they leave residual feature-level knowledge that makes unlearning reversible, and (3) they incur prohibitive computational or memory costs, limiting scalability. Recent works have partially addressed these issues through adversarial regularization or subspace erasure, yet both fall short of providing irreversible and scalable feature-level unlearning. We propose MoRE: Mixture of Remapping Experts, a novel framework for exact feature-level unlearning. MoRE introduces three innovations: (i) prototype-orthogonal projection to preserve remain utility by decorrelating forget and remain prototypes prior to erasure, (ii) remapping with mixture experts to merge forget prototypes into multiple remain prototypes, eliminating their separability and impeding recovery via fine-tuning, and (iii) efficient activation-mean prototypes that reduce unlearning to a single forward pass, achieving linear computational complexity and constant memory. Extensive experiments demonstrate that MoRE preserves utility, ensures irreversibility at the feature level, and scales effectively to large models and datasets, thereby establishing a principled pathway toward trustworthy and scalable machine unlearning.

## 1 Introduction

The growing demand for data privacy and model safety has led to stronger regulatory frameworks such as the General Data Protection Regulation (GDPR) (Regulation, 2016), which not only grants individuals the right to request the removal of personal data but also restricts the processing of entire categories of sensitive information, such as racial or ethnic attributes, and children's data, to name a few. Enforcing such regulations could be crippling for AI businesses, as it is particularly difficult in deep learning to selectively ablate knowledge from a highly monolithic and polysemantic model (Elhage et al., 2022): the influence of sensitive data is deeply embedded and entangled across model parameters. This challenge has sparked the emergence of the field of *machine unlearning* (MU), which aims to remove the influence of designated forget data from trained models while preserving the utility of remain data (*i.e.*, the model's performance on remain data).

MU methods have achieved remarkable progress in recent years, but they still fall short of meeting real-world demands for *complete knowledge deletion* (Lee et al., 2025; Kim et al., 2025). In particular, prior studies have shown that traces of forget data can persist in embedding features, creating vulnerabilities that make forgotten concepts recoverable, effectively *reversing* unlearning (Graves et al., 2020). To mitigate this issue, Erasing Space Concept (ESC) (Lee et al., 2025) seeks to achieve *feature-level unlearning* by ensuring that no information about the forget data remains in the latent feature representation. ESC operationalizes this idea by first identifying the principal components or *prototypes* that characterize the forget concepts, obtained via singular value decomposition (SVD) on its embedding features. A projection layer is then inserted between the feature extractor and the classification head to remove the most influential $k$ prototypes, thereby erasing the subspace associated with the forget data. ESC is also computationally efficient, requiring only a single forward pass and an SVD computation; traditional MU methods require end-to-end training/finetuning. To date, it has achieved SOTA in both unlearning effectiveness and efficiency.

**Erasing Space Concept**  **Remapping (ours)**  **Mixture of Remapping Experts (ours)**

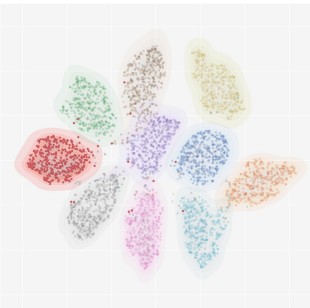 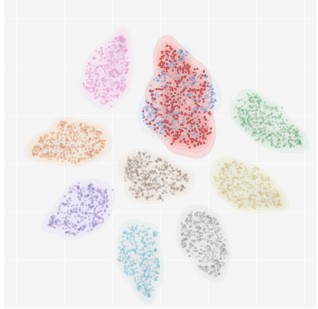 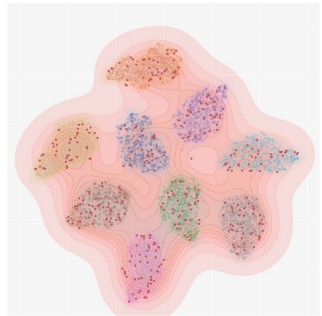

Figure 1: t-SNE visualization of the latent feature space of each unlearning method on the CIFAR-10 dataset with AllCNN. We removed the knowledge corresponding to the dots in red and remapped the prototype vector to the blue cluster. We can see that in ESC, forget features form a red cluster even after unlearning. Versus, when we remap the forget feature, the red cluster in absorbed into the blue target cluster, leaving it indistinguishable in the latent feature space. With multiple remapping experts, we can further scatter the feature vector across the latent space for irreversible unlearning.

While ESC marks an important step toward stronger unlearning guarantees, it still faces several critical challenges, which are also common to other MU methods. First, utility degradation remains a persistent issue: aggressively erasing feature subspaces often disrupts remain data representations, leading to significant performance drops. Second, residual forget knowledge in the latent space makes unlearning reversible: even after unlearning, forget data remain cohesive and separable from remain data in the latent feature space, making forgotten knowledge vulnerable to recovery through light fine-tuning (see Fig. 1). Third, unlearning inefficiency continues to hinder scalability. For traditional training-based MU methods, the computational overhead is substantial, limiting their adoption to small-scale models. ESC alleviates this issue by removing the need for retraining, but it suffers from memory inefficiency: storing activations and performing SVD scales poorly with the size of the forget set, restricting its applicability to small-scale datasets. These challenges highlight the need for a framework that preserves the utility of the remain dataset, ensures irreversible feature-level erasure, and is scalable to larger models and datasets.

To address these gaps, we propose **MoRE: Mixture of Remapping Experts**, a framework for *irreversible* feature-level unlearning. MoRE disrupts the residual cohesive–separable structure left by existing approaches: instead of allowing forget features to remain as a distinct cluster in the latent space, it remaps them into the distribution of the remain data, making the two indistinguishable at the feature level (see Fig. 1). To this end, MoRE introduces three core innovations:

- **Prototype-orthogonal projection for utility preservation.** Forget prototypes are often highly correlated with remain prototypes, causing naive erasure or remapping to harm utility on remain data. We introduce an orthogonalization step that decorrelates prototypes prior to erasure, ensuring that unlearning targets only forget data-specific information.

- **Remapping for irreversibility.** Beyond erasing prototypes, MoRE remaps forget prototypes to multiple remain prototypes (see Fig. 1). This scattering of forget features across the latent feature space breaks the separable-cohesive structure of forget clusters, significantly impeding the recovery of forgotten knowledge through fine-tuning or linear probing.

- **Unlearning Efficiency.** By using concept-wise activation means as prototypes, MoRE achieves linear time complexity in the number of data samples and constant space complexity with respect to the number of concepts/classes and feature dimensions. This lightweight design enables irreversible unlearning to scale efficiently to large datasets and models.

Through these innovations, MoRE addresses the long-standing limitations of MU by simultaneously preserving remain utility, ensuring irreversibility of unlearning at the feature level, and enabling scalability. Our extensive experiments demonstrate that MoRE establishes a principled and efficient pathway toward trustworthy and irreversible feature-level unlearning.

## 2 RELATED WORKS & BACKGROUNDS

**Machine Unlearning Definition.** The current formulation of MU has largely been shaped by privacy principles (Dwork et al., 2014), thereby framing the central objective as approximating the *retrain-from-scratch model* (*i.e.,* model trained solely on the remain data) that serves as the "*gold standard*" (Ginart et al., 2019). However, this problem formulation has raised two major concerns: i) reliability of the retrain-from-scratch model as the gold standard, and ii) lack of utility considerations to accommodate real-world scenarios.

First, the reliability of retrain-from-scratch as a reference has been called into question. Thudi et al. (2022b) theoretically demonstrate that models trained on different datasets can converge to similar parameter states, complicating direct comparisons. Similarly, Goel et al. (2022) shows that retrained models may vary significantly depending on hyperparameter choices, further undermining the reliability of retrain-from-scratch as a gold standard. Second, the current formulation overlooks the utility perspective that matters in real-world settings. In practice, users making unlearning requests often expect more than simply mimicking a retrained model (Kurmanji et al., 2023). For example, if requested knowledge is nominally erased but the model can still exploit it indirectly through related representations to produce the same output as before unlearning, users will not perceive the unlearning as successful (Kim et al., 2025).

**Extended Interpretation of MU: Knowledge Deletion (KD).** To address such concerns, Lee et al. (2025) proposed KD as an extended interpretation of MU. While KD inherits traditional evaluation metrics of the original MU standard, it augments them with explicit utility requirements to ensure that unlearning is both effective and perceivable. In this formulation, the objective is not only to remove the influence of the forget data but also to *minimize the usefulness of forget knowledge while maximizing that of remain knowledge*. By introducing utility metrics alongside privacy measures, KD no longer treats the retrain-from-scratch model as the sole point of reference, thereby enabling a more robust and practical evaluation that better reflects real-world expectations of unlearning. For those reasons, we also base our work on the KD task formulation.

**Erasing Space Concept (ESC).** *Erasing Space Concept (ESC)* (Lee et al., 2025) is a pioneering approach to achieving feature-level unlearning. The core idea is to remove the principal components, or prototypes, of the forget data from the latent feature space, thereby erasing knowledge at the feature level. Prototypes are representative vectors of the feature distribution, typically obtained through methods such as factorization or clustering. Concretely, the forget dataset is passed through the feature extractor $h_\psi(\cdot)$ of the original model to produce the feature matrix $\mathbf{Z}_f \in \mathbb{R}^{d \times N_f}$, where $d$ is the feature dimensions and $N_f$ is the number of forget data. This matrix is then decomposed via singular value decomposition (SVD), $\mathbf{Z}_f = \mathbf{U}\mathbf{\Sigma}\mathbf{V}^\top$, where the columns of $\mathbf{U}$ represent the principal directions of the forget feature space. ESC constructs a projection layer that suppresses these directions by pruning the top-$k$ directions. The resulting unlearned feature extractor and full ESC model are defined as

$$h_{\psi_p}(\mathbf{x}) = \mathbf{U}_p \mathbf{U}_p^\top h_\psi(\mathbf{x}), \quad f_{\text{ESC}}(\mathbf{x}) = g_\phi(h_{\psi_p}(\mathbf{x})), \tag{1}$$

where $\mathbf{U}_p = \mathbf{U}[k :]$ denotes the pruned subspace, and $g_\phi(\cdot)$ denotes the classification head. To further refine this process, ESC also proposed a training-based variant in which a learnable mask is applied over the left singular vectors (more details in Appendix §B.2). ESC has demonstrated state-of-the-art results across both conventional unlearning benchmarks and the new Knowledge Retention (KR) metric, which measures feature-level unlearning efficacy. Its efficiency is another strength: since it requires only a single forward pass and an SVD on the collected features, ESC is training-free and scalable to large models.

**Limitations of ESC.** Despite these advantages, ESC exhibits several limitations. First, it frequently leads to degraded utility on the remain dataset. This arises from the high correlation between the principal directions of forget and remain data, such that removing the former inevitably disrupts the latter (more details in §3.1). Second, even though ESC erases the forget subspace, the forget feature vectors often remain cohesive and separable in the latent space (see Fig. 1). This makes them vulnerable to recovery through light fine-tuning, raising concerns about privacy leakage. Third, although compute-efficient, ESC is not memory-efficient: storing all activations and performing SVD requires memory proportional to the number of forget samples and feature dimensions (i.e., $O(N_f d)$), which can cause out-of-memory failures on large-scale datasets. These challenges highlight the open problems that remain in achieving utility-preserving, irreversible, and scalable unlearning.

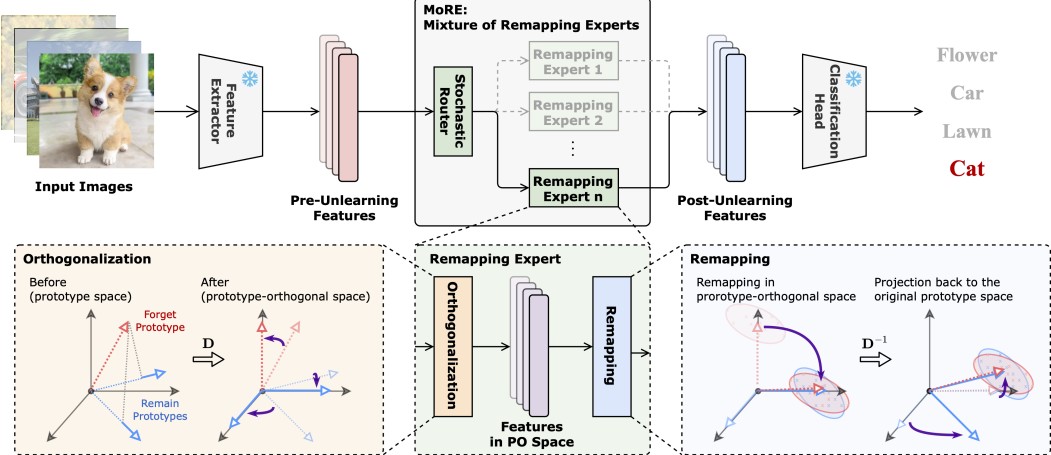

Figure 2: An overview of our proposed Mixture of Remapping Experts unlearning framework.

**Notations.** $\mathcal{D} = \{(\mathbf{x}_i, y_i)\}_{i=1}^N$ is a labeled training dataset with inputs $\mathbf{x}_i$ and labels $y_i$. We define $\mathcal{D}_f \subset \mathcal{D}$ as the forget training data and $\mathcal{D}_r = \mathcal{D} \setminus \mathcal{D}_f$ as the remain training data. Similarly, we define/denote forget test data, $\mathcal{D}_{ft}$, and train test data, $\mathcal{D}_{rt}$. The AI model parameterized by $\theta$ is denoted by $f_\theta$, which is expressed as a composition of a feature extractor and a classification head, $f_\theta = g_\phi \circ h_\psi$.

## 3 METHODS

To address the limitations of ESC, we propose **Mixture of Remapping Experts (MoRE)**, a novel unlearning framework that achieves irreversibility by remapping forget features into remain feature distributions. This is enabled through *prototype-orthogonal (PO) projection* and *erasing & remapping* operations in this disentangled PO space. With a single remapping expert, each forget prototype is mapped to one remain prototype, disrupting separability. Extending this to multiple experts disperses forget features across different remain prototypes, breaking cohesive structure and scattering the forget features across the latent feature space. This leaves little residual structure for linear probes to exploit, making recovery through probing significantly harder while preserving utility.

An overview of the proposed framework is shown in Fig. 2. The MoRE layer consists of two components: router and remapping experts. The router, either stochastic or conditional, selects which expert processes a given feature vector. The chosen expert applies two sequential operations: (i) projection into prototype-orthogonal space, and (ii) erasing and remapping of forget prototypes to remain prototypes. By distributing forget features across multiple experts, MoRE breaks separability and residual cohesion, while preserving overall utility.

### 3.1 PROTOTYPE-ORTHOGONAL PROJECTION

The first step of MoRE is prototype-orthogonal projection, which is essential for enabling stable forget prototype erasure and remapping. Our key observation is that forget and remain prototypes are often highly correlated in the latent feature space. As a result, directly erasing or modifying forget prototypes can inadvertently distort remain prototypes, leading either to degraded model utility or to incomplete unlearning.

Empirical evidence of this correlation is shown in Fig. 3. We compute the cosine similarity between class prototypes on CIFAR-10 with AllCNN architecture. On average, the cosine similarities between forget and remain prototypes are around 0.5, with maximum values reach-

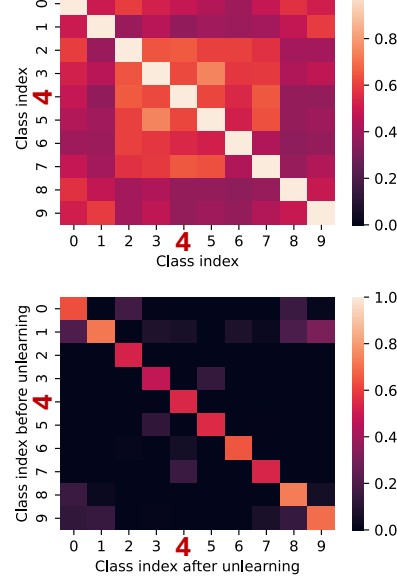

Figure 3: Cosine similarities between class-wise activation means before (*top*) and before & after (*bottom*) unlearning. Class **4** is unlearned.

ing as high as 0.77, confirming substantial overlap between forget and remain representations. To examine the effect of erasure, we apply ESC and measure the prototype correlations before and after unlearning. Ideally, the autocorrelation of remain prototypes should stay close to 1.0, indicating that their representations are preserved, while that of the forget prototype should drop to zero, indicating complete removal of its information. In practice, however, we observe a sharp decrease in both: for CIFAR-10, remain prototypes drop from 1.0 to 0.52, and the forget prototype falls to only 0.46, a level similar to the other remain prototypes. This highlights that erasing the forget prototype substantially disrupts the remain prototypes as well.

To mitigate this issue, we project input features into a *prototype-orthogonal space*, where the projected prototype vectors (both forget and remain) form an orthogonal basis. In this space, each prototype corresponds to an independent coordinate axis, so that editing or erasing a forget prototype does not influence other prototypes.[1] The goal is to construct a projection matrix $\mathbf{D}$ that enforces this condition. Mathematically, this can be phrased as $\mathbf{DP} = \mathbf{I}_k$, where $\mathbf{P} \in \mathbb{R}^{d \times k}$ is the prototype matrix whose columns are class prototypes, and $\mathbf{I}_k$ is the $k \times k$ identity matrix.

It can be easily noticed that $\mathbf{D}$ must be the pseudoinverse of $\mathbf{P}$, given that $\mathbf{P}$ is full-rank, in order to satisfy the above condition. Using the singular value decomposition (SVD) of $\mathbf{P}$, we obtain

$$\mathbf{P} = \mathbf{U}\,\mathbf{\Sigma}\,\mathbf{V}^\top, \qquad \mathbf{D} = \mathbf{P}^+ = \mathbf{V}\,\mathbf{\Sigma}^{-1}\,\mathbf{U}^\top, \tag{2}$$

where $\mathbf{U}$ and $\mathbf{V}$ are left and right singular vectors of $\mathbf{P}$ respectively, $\mathbf{\Sigma}$ is a diagonal matrix of singular values. Note that we compute the pseudoinverse using SVD rather than the normal-equation form $(\mathbf{P}^\top\mathbf{P})^{-1}\mathbf{P}^\top$, since the latter squares the condition number of $\mathbf{P}$ (i.e., $\kappa = \sigma_{\max}/\sigma_{\min}$) and can unnecessarily amplify fitting errors. Using the projection matrix $\mathbf{D}$, input features are mapped into a transformed space where prototype vectors are orthogonal by construction. In this space, erasing or remapping forget prototypes can be carried out with minimal distortion to remain prototypes, enabling better preservation of remain knowledge while simultaneously maximizing unlearning performance.

### 3.2 ERASING AND REMAPPING

With the input feature, $\mathbf{z} = h_\psi(\mathbf{x})$, projected onto the PO space (i.e., $\mathbf{m} = \mathbf{Dz}$), we can now perform erasing of the forget prototype. Here, the $i$-th element of $\mathbf{m}$, $m_i$, represents the membership of $\mathbf{z}$ to the subspace spanned by the $i$-th prototype, $\mathbf{p}_i$. We can then erase the contribution of forget prototypes by sparsifying the corresponding coordinates of $\mathbf{m}$ and then reconstructing the feature by projecting it back into the original prototype space. Formally, let $\mathbf{s} \in \{0, 1\}^k$ be a binary selector vector whose entries indicate which prototypes should be erased ($s_i = 1$ if prototype $i$ belongs to the forget set, otherwise $s_i = 0$). Then the unlearned feature, $\hat{\mathbf{z}}$, is given by:

$$\hat{\mathbf{z}} = \mathbf{P}\big(\mathbf{D} - \mathrm{diag}(\mathbf{s})\mathbf{D}\big)\mathbf{z}, \tag{3}$$

where $\mathrm{diag}(\mathbf{s})$ is the diagonal matrix constructed from $\mathbf{s}$.

One limitation of this formulation is that $\mathbf{D}$ projects $\mathbf{z}$ entirely into the low-rank prototype span, discarding all information outside it. To avoid this loss, we add a *complement-space projection* term, akin to a skip connection, that preserves the non-prototype components. Then the unlearned feature is given by

$$\hat{\mathbf{z}} = \mathbf{P}\big(\mathbf{D} - \mathrm{diag}(\mathbf{s})\mathbf{D}\big)\mathbf{z} + \big(\mathbf{I} - \mathbf{PD}\big)\mathbf{z}, \tag{4}$$

where the first term corresponds to prototype-space erasure, while the second term to complement space projection. This ensures a full-rank transformation: the influence of forget prototypes is erased, while all information outside the prototype span remains intact.

Above expression can be simplified to

$$\hat{\mathbf{z}} = (\mathbf{I} - \mathbf{P}_f\mathbf{D})\mathbf{z}, \tag{5}$$

where $\mathbf{P}_f = \mathbf{P}\,\mathrm{diag}(\mathbf{s})$ is a submatrix of $\mathbf{P}$ consisting only of the forget prototypes, with the remain prototypes zeroed out. Notice that $\mathbf{P}_f\mathbf{D}$ acts as a linear operator that surgically extracts the components of $\mathbf{z}$ aligned with the forget prototypes. Subtracting this term ensures that $\hat{\mathbf{z}}$ lies entirely in the subspace orthogonal to the forget prototypes, thereby erasing their influence.

---

[1]Strictly speaking, only orthogonality between forget and remain prototypes is required. Enforcing this selectively, however, is mathematically complex, so we adopt full mutual orthogonality for mathematical brevity.

To extend erasure into remapping, we expand the previous formulation to not only suppress the contribution of forget prototypes, but also to detect their presence and redirect it toward remain prototypes. Specifically, we define

$$\hat{\mathbf{z}} = \big(\mathbf{I} - \mathbf{P}_f\mathbf{D} + \mathbf{P}_t\,\mathrm{diag}(\mathbf{s})\mathbf{D}\big)\mathbf{z}, \tag{6}$$

where $\mathbf{P}_t$ denotes the target prototype matrix that specifies how each forget prototype is remapped. Concretely, $\mathbf{P}_t$ is constructed by assigning each forget prototype $\mathbf{p}_i$ (with $s_i = 1$) to a designated remain prototype $\mathbf{p}_j$. In this case, the $i$-th column of $\mathbf{P}_t$ is set to $\mathbf{p}_j$, while columns corresponding to non-forget indices remain zero. The first two terms $(\mathbf{I} - \mathbf{P}_f\mathbf{D})$ correspond to erasure as before. The additional term $\mathbf{P}\,\mathrm{diag}(\mathbf{1} - \mathbf{s})\mathbf{D}$ serves as a detector for the presence of forget prototypes: when a forget prototype is activated, the operation introduces a contribution from one of the remain prototypes. In effect, the erased coordinates are replaced with remain-aligned components, so that forget features are not only removed but actively remapped into the remain subspace. This process makes the latent representations of forget data indistinguishable from those of remain data, thereby strengthening the irreversibility of our proposed unlearning framework.

### 3.3 CONDITIONAL/STOCHASTIC REMAPPING WITH MULTIPLE EXPERTS

To further strengthen the irreversibility of our unlearning framework, we extend the remapping mechanism by introducing multiple remapping layers, each mapping a given forget prototype to different remain prototypes. We draw our inspiration from the conditional computation technique introduced in the Mixture of Experts (MoE) architecture. However, unlike the standard MoE, where experts improve prediction accuracy, in our proposed method, each expert specializes in remapping forget prototypes into distinct remain prototypes. The motivation for this design comes from a limitation we observe with single-prototype remapping. Although redirecting each forget prototype to one remain prototype helps blur separability, the remapped forget features yet remain cohesive. As mentioned previously, this residual cohesive structure can be exploited by carefully tuned linear probes. By scattering each forget prototype across multiple remain prototypes, MoRE breaks this residual cohesion and hence makes unlearning irreversible. Achieving this scattering requires a router that determines how inputs are assigned to experts. We consider two router designs: (i) a trained conditional router and (ii) a stochastic router.

The conditional router follows the standard design in MoE architectures: it is input-dependent and trained to select the most suitable expert for a given feature. However, a common issue in MoE training is *expert specialization collapse*, where the router favors a small subset of experts. While acceptable in traditional MoE settings, this behavior undermines the motivation of MoRE, since reduced expert diversity means less scattering and more residual cohesion. To mitigate this, we initialize the router with $\mathbf{P}^+$, the pseudo-inverse of the prototype matrix, which encourages more balanced expert usage and provides a stronger basis for training. The stochastic router, in contrast, is input-independent and routes each input randomly. This ensures an even workload among the experts and that forget features are widely scattered, maximizing the disruption of cohesive structures. Moreover, because it requires no training, the stochastic router is highly scalable to large models and datasets. For that reason, we adopt it as the default choice in our experiments. More details of our two router designs are provided in the Appendix §D.

### 3.4 UNLEARNING EFFICIENCY

MoRE constructs prototype vectors by *averaging concept-wise activations* from the feature extractor, a simple and effective strategy that consistently performs well in our image classification experiments. Although other methods, such as clustering or matrix factorization, could be used, we adopt activation means for their minimal computational overhead. All following unlearning operations are training-free and involve lightweight linear algebra. This design yields $O(Nd)$ computational complexity for prototype collection (with $N$ the number of samples) and $O(dk)$ memory complexity for storing prototypes (with $k$ the number of concepts), making MoRE scalable to large-scale models and datasets with negligible overhead compared to training-based unlearning methods.

## 4 EXPERIMENTS

**Datasets, Models, Settings, and Baselines.** We follow the standard experimental setup commonly adopted in the approximate unlearning and KD literature, ensuring fair comparison across baselines. We evaluate MoRE on CIFAR-10 using All-CNN, CIFAR-100 using ResNet-18, Tiny-ImageNet, and ImageNet using ViT-base-16. For class-wise unlearning, 10% of classes are removed; for

Table 1: Accuracy and KR performance in the KD settings are evaluated using CIFAR-10 with AllCNN, CIFAR-100 with ResNet-18, and Tiny-ImageNet with ViT. The table presents the mean and standard deviation (mean ± std) across three trials with the best value highlighted in bold.

**CIFAR-10**

| Method | D_f(↓) | D_r(↑) | D_ft(↓) | D_rt(↑) | HM(↑) | HM_t(↑) |
|---|---|---|---|---|---|---|
| Original | 99.92 | 99.94 | 91.80 | 91.07 | 0.16 | 15.05 |
| Retrain | 0.00 | 99.15 | 0.00 | 92.04 | 99.57 | 95.86 |
| Finetune | 0.00 ± 0.00 | 90.86 ± 2.66 | 0.00 ± 0.00 | 84.89 ± 2.39 | 95.20 ± 0.80 | 91.83 ± 0.82 |
| NG | 6.35 ± 0.08 | 89.62 ± 0.22 | 5.72 ± 0.11 | 81.26 ± 0.17 | 91.59 ± 0.03 | 87.28 ± 0.03 |
| RL | 0.00 ± 0.00 | 97.52 ± 0.29 | 0.00 ± 0.00 | 90.14 ± 0.06 | 98.74 ± 0.08 | 94.82 ± 0.02 |
| BS | 9.87 ± 0.02 | 95.23 ± 0.01 | 1.00 ± 0.01 | 85.75 ± 0.01 | 92.61 ± 0.01 | 87.83 ± 0.01 |
| Lau | 0.18 ± 0.00 | 86.57 ± 5.67 | 0.13 ± 0.01 | 79.51 ± 3.79 | 92.71 ± 1.84 | 88.53 ± 1.39 |
| ESC | 11.22 ± 1.18 | 88.84 ± 0.40 | 10.70 ± 0.87 | 81.27 ± 0.50 | 88.81 ± 0.74 | 85.10 ± 0.65 |
| ESC-T | 0.03 ± 0.02 | 88.88 ± 0.22 | 0.00 ± 0.00 | 81.21 ± 0.19 | 94.10 ± 0.12 | 89.63 ± 0.11 |
| **Remap** | 0.00 ± 0.00 | 99.87 ± 0.00 | 0.00 ± 0.00 | 91.16 ± 0.01 | **99.94 ± 0.00** | 95.38 ± 0.01 |
| **MoRE** | 0.00 ± 0.00 | 99.87 ± 0.01 | 0.00 ± 0.00 | 91.02 ± 0.11 | 99.93 ± 0.01 | 95.30 ± 0.06 |

**CIFAR-10 (KR setting; lr = 0.1)**

| Method | D_f(↓) | D_r(↑) | D_ft(↓) | D_rt(↑) | HM(↑) | HM_t(↑) |
|---|---|---|---|---|---|---|
| Original | 99.88 | 99.95 | 91.20 | 91.07 | 0.24 | 16.05 |
| Retrain | 72.62 | 97.06 | 72.90 | 88.01 | 42.71 | 41.44 |
| Finetune | 79.24 ± 11.93 | 93.80 ± 0.14 | 76.91 ± 3.90 | 87.30 ± 0.19 | 33.45 ± 21.55 | 36.36 ± 6.35 |
| NG | 94.34 ± 0.09 | 98.05 ± 0.01 | 83.97 ± 0.04 | 88.80 ± 0.00 | 10.69 ± 0.28 | 27.16 ± 0.07 |
| RL | 87.17 ± 1.69 | 97.53 ± 0.00 | 81.19 ± 0.83 | 89.38 ± 0.02 | 22.55 ± 4.07 | 31.03 ± 1.53 |
| BS | 96.43 ± 0.01 | 98.90 ± 0.00 | 85.50 ± 0.05 | 89.36 ± 0.00 | 6.88 ± 0.02 | 24.95 ± 0.10 |
| Lau | 90.60 ± 0.31 | 94.29 ± 0.10 | 80.70 ± 0.32 | 85.37 ± 0.03 | 17.06 ± 0.83 | 31.47 ± 0.54 |
| ESC | 82.97 ± 0.23 | 93.45 ± 0.21 | 73.60 ± 0.17 | 84.98 ± 0.16 | 28.80 ± 0.34 | 40.28 ± 0.19 |
| ESC-T | 87.34 ± 1.11 | 94.14 ± 0.35 | 78.80 ± 1.21 | 85.24 ± 0.31 | 22.31 ± 1.75 | 33.94 ± 1.58 |
| **Remap** | 33.20 ± 57.50 | 96.17 ± 6.40 | 29.83 ± 51.67 | 87.76 ± 5.92 | 66.89 ± 57.24 | 69.78 ± 44.34 |
| **MoRE** | 10.79 ± 9.85 | 95.96 ± 3.36 | 12.87 ± 8.96 | 86.82 ± 3.30 | **92.37 ± 6.91** | **86.91 ± 6.17** |

**CIFAR-100**

| Method | D_f(↓) | D_r(↑) | D_ft(↓) | D_rt(↑) | HM(↑) | HM_t(↑) |
|---|---|---|---|---|---|---|
| Original | 100.00 | 99.98 | 67.00 | 80.81 | 0.00 | 46.86 |
| Retrain | 0.00 | 96.64 | 0.00 | 72.00 | 98.29 | 83.72 |
| Finetune | 0.00 ± 0.00 | 87.41 ± 0.18 | 0.00 ± 0.00 | 69.39 ± 0.25 | 93.28 ± 0.06 | 81.93 ± 0.12 |
| NG | 29.19 ± 0.32 | 96.79 ± 0.02 | 5.65 ± 0.22 | 70.06 ± 0.09 | 81.79 ± 0.18 | 80.41 ± 0.04 |
| RL | 1.27 ± 1.34 | 82.19 ± 0.27 | 1.00 ± 2.00 | 67.58 ± 0.04 | 89.70 ± 0.43 | 80.33 ± 0.16 |
| BS | 1.93 ± 0.54 | 58.57 ± 10.75 | 1.00 ± 2.00 | 41.48 ± 4.33 | 73.34 ± 6.86 | 58.45 ± 5.39 |
| Lau | 2.60 ± 0.00 | 96.35 ± 0.01 | 0.00 ± 0.00 | 70.08 ± 0.01 | 96.87 ± 0.00 | 82.41 ± 0.01 |
| ESC | 0.50 ± 0.38 | 88.85 ± 0.32 | 0.00 ± 0.00 | 64.72 ± 0.17 | 93.77 ± 0.05 | 78.58 ± 0.10 |
| ESC-T | 0.00 ± 0.00 | 99.03 ± 0.00 | 0.00 ± 0.00 | 73.88 ± 0.09 | 99.51 ± 0.00 | 84.98 ± 0.00 |
| **Remap** | 0.00 ± 0.00 | 99.98 ± 0.00 | 0.00 ± 0.00 | 80.82 ± 0.00 | **99.99 ± 0.00** | **89.39 ± 0.00** |
| **MoRE** | 0.00 ± 0.00 | 99.98 ± 0.00 | 0.00 ± 0.00 | 80.22 ± 0.02 | **99.99 ± 0.00** | 89.03 ± 0.02 |

**CIFAR-100 (KR setting; lr = 0.1)**

| Method | D_f(↓) | D_r(↑) | D_ft(↓) | D_rt(↑) | HM(↑) | HM_t(↑) |
|---|---|---|---|---|---|---|
| Original | 100.00 | 99.98 | 67.00 | 80.41 | 0.00 | 46.80 |
| Retrain | 57.20 | 97.20 | 58.00 | 71.66 | 59.43 | 52.96 |
| Finetune | 66.31 ± 2.46 | 92.88 ± 0.03 | 53.66 ± 0.89 | 72.70 ± 0.10 | 49.37 ± 2.89 | 56.59 ± 0.41 |
| NG | 96.80 ± 0.03 | 99.65 ± 0.00 | 52.99 ± 0.67 | 74.94 ± 0.01 | 6.19 ± 0.09 | 57.76 ± 0.36 |
| RL | 70.59 ± 1.39 | 88.98 ± 0.12 | 56.85 ± 16.67 | 71.82 ± 0.13 | 44.16 ± 1.90 | 53.58 ± 9.79 |
| BS | 61.07 ± 15.71 | 86.49 ± 1.55 | 33.91 ± 6.00 | 60.65 ± 0.39 | 53.31 ± 11.80 | 63.18 ± 1.89 |
| Lau | 94.53 ± 0.04 | 99.39 ± 0.00 | 48.98 ± 2.00 | 74.18 ± 0.00 | 10.36 ± 0.12 | 60.42 ± 1.01 |
| ESC | 99.60 ± 0.00 | 99.92 ± 0.00 | 59.65 ± 1.56 | 75.90 ± 0.01 | 0.80 ± 0.00 | 52.65 ± 1.15 |
| ESC-T | 96.07 ± 0.06 | 99.89 ± 0.00 | 61.99 ± 0.67 | 75.35 ± 0.01 | 7.55 ± 0.21 | 50.51 ± 0.49 |
| **Remap** | 51.88 ± 15.94 | 94.23 ± 1.88 | 41.63 ± 10.56 | 76.24 ± 1.61 | 62.76 ± 14.71 | 65.84 ± 7.35 |
| **MoRE** | 0.07 ± 0.09 | 99.86 ± 0.18 | 1.27 ± 0.72 | 77.01 ± 0.06 | **99.89 ± 0.14** | **86.53 ± 0.31** |

**Tiny-ImageNet**

| Method | D_f(↓) | D_r(↑) | D_ft(↓) | D_rt(↑) | HM(↑) | HM_t(↑) |
|---|---|---|---|---|---|---|
| Original | 98.20 | 96.45 | 96.00 | 90.29 | 3.53 | 7.66 |
| Retrain | 0.00 | 99.98 | 0.00 | 85.23 | 99.99 | 92.03 |
| Finetune | 0.00 ± 0.00 | 66.89 ± 18.98 | 0.00 ± 0.00 | 55.49 ± 6.69 | 80.12 ± 9.42 | 71.35 ± 4.48 |
| NG | 0.01 ± 0.00 | 0.62 ± 0.00 | 0.00 ± 0.00 | 0.62 ± 0.00 | 1.24 ± 0.00 | 1.23 ± 0.00 |
| RL | 10.83 ± 4.26 | 96.98 ± 0.05 | 9.19 ± 5.45 | 85.14 ± 0.10 | 92.79 ± 1.17 | 87.73 ± 1.26 |
| BS | 36.07 ± 0.29 | 48.02 ± 0.19 | 35.43 ± 0.08 | 46.65 ± 0.16 | 54.84 ± 0.03 | 54.16 ± 0.05 |
| Lau | 0.00 ± 0.00 | 95.55 ± 0.00 | 0.00 ± 0.00 | 89.73 ± 0.00 | 97.73 ± 0.00 | 94.59 ± 0.00 |
| ESC | 0.10 ± 0.02 | 96.36 ± 0.05 | 0.03 ± 0.06 | 90.57 ± 0.05 | 98.10 ± 0.02 | 95.03 ± 0.05 |
| ESC-T | 0.00 ± 0.00 | 96.44 ± 0.03 | 0.00 ± 0.00 | 90.57 ± 0.04 | **98.19 ± 0.02** | **95.05 ± 0.02** |
| **Remap** | 0.03 ± 0.01 | 96.18 ± 0.01 | 0.00 ± 0.00 | 90.13 ± 0.03 | 98.04 ± 0.01 | 94.81 ± 0.02 |
| **MoRE** | 0.02 ± 0.01 | 96.16 ± 0.02 | 0.00 ± 0.00 | 90.01 ± 0.09 | 98.03 ± 0.01 | 94.74 ± 0.05 |

**Tiny-ImageNet (KR setting; lr = 0.1)**

| Method | D_f(↓) | D_r(↑) | D_ft(↓) | D_rt(↑) | HM(↑) | HM_t(↑) |
|---|---|---|---|---|---|---|
| Original | 98.20 | 96.83 | 96.00 | 90.10 | 3.53 | 7.66 |
| Retrain | 78.57 | 99.87 | 76.00 | 80.70 | 35.29 | 37.00 |
| Finetune | 53.69 ± 5.46 | 71.19 ± 13.19 | 51.67 ± 10.00 | 59.63 ± 4.77 | 55.97 ± 0.44 | 53.20 ± 1.33 |
| NG | 17.11 ± 0.11 | 19.84 ± 0.33 | 16.10 ± 0.03 | 18.62 ± 0.14 | 32.02 ± 0.53 | 30.48 ± 0.24 |
| RL | 91.21 ± 0.25 | 97.49 ± 0.04 | 84.90 ± 0.01 | 85.30 ± 0.24 | 16.10 ± 0.71 | 25.66 ± 0.02 |
| BS | 72.25 ± 0.02 | 74.26 ± 0.03 | 68.83 ± 0.12 | 69.97 ± 0.02 | 40.41 ± 0.02 | 43.12 ± 0.10 |
| Lau | 96.69 ± 0.00 | 96.86 ± 0.00 | 89.70 ± 0.00 | 90.18 ± 0.00 | 6.40 ± 0.00 | 18.49 ± 0.00 |
| ESC | 15.78 ± 0.84 | 96.88 ± 0.02 | 13.67 ± 1.06 | 90.54 ± 0.08 | 90.10 ± 0.49 | 88.38 ± 0.52 |
| ESC-T | 95.47 ± 0.12 | 96.78 ± 0.03 | 89.57 ± 0.21 | 90.23 ± 0.11 | 8.66 ± 0.21 | 18.70 ± 0.33 |
| **Remap** | 55.18 ± 15.74 | 93.57 ± 1.65 | 51.03 ± 14.00 | 87.00 ± 1.46 | 59.60 ± 14.98 | 61.96 ± 11.95 |
| **MoRE** | 0.50 ± 0.07 | 96.52 ± 0.01 | 0.43 ± 0.25 | 89.94 ± 0.07 | **97.99 ± 0.04** | **94.51 ± 0.14** |

instance-wise unlearning, 10% of samples are discarded. All experiments run on a single NVIDIA A100 GPU, with additional details in the Appendix §B. We compare MoRE against a wide set of approximate unlearning methods. As reference, we include **Original**, a model trained on the full dataset $\mathcal{D}$, and **Retrain**, a model trained only on the remain data $\mathcal{D}_r$. For unlearning approaches, we consider both classical and recent methods, with full details provided in the Appendix §B.2.

**Evaluations Metrics.** We evaluate unlearning performance by partitioning the dataset into remain training data $\mathcal{D}_r$, forget training data $\mathcal{D}_f$, remain test data $\mathcal{D}_{rt}$, and forget test data $\mathcal{D}_{ft}$. For each subset, we report accuracy after unlearning as well as the **Knowledge Retention (KR)** metric to measure feature-level unlearning performance (details in §B.3). Following prior work, we also employ the **Harmonic Mean (HM)** to measure balanced utility (details in §B.3) and **MIA** for sample-wise unlearning (details in §B.3).

## 4.1 EXPERIMENTAL RESULTS

**Model Utility Evaluation.** Table 1 reports comprehensive results for the KD task across multiple dataset–model pairs, comparing our method against recent MU baselines. Due to space constraints, we present the most competitive recent methods here, while full results—including all baselines, additional learning rate configurations, and ImageNet results—are provided in the Appendix §C.1. Here, we compare two variants of our method: **Remap**, a single-expert model facilitating one-to-one remapping, and **MoRE**, a multi-expert extension that enables one-to-many remapping for stronger unlearning. Our proposed methods consistently achieves the best performance across all settings (see HM and HM_t). Notably, it surpasses training-based unlearning methods that require orders of magnitude more compute. The gains are especially pronounced on CIFAR-10 and CIFAR-100, where MoRE not only maintains remain-set accuracy (D_r and D_rt) but in some cases slightly improves it. These results underscore the effectiveness of our method enabling both near-perfect forgetting of unwanted knowledge and preservation of remain knowledge (see HM and HM_t).

Original SD | CA | ESD | SLD-Med | RECE | UCE | **Ours**

Figure 4: Qualitative results of artistic style erasure of Van Gogh. The prompt is "*A depiction of a starry night over a quiet town, reminiscent of Van Gogh's famous painting*".

**KR Evaluation.** The impact of MoRE is even more striking under the KR evaluation. As shown in Table 1, existing baselines suffer from substantial recovery of forget accuracy (D_f and D_ft), approaching pre-unlearning performance. In contrast, our method keeps forget accuracy down to the level of random guessing, decisively outperforming all baselines and even the retrain model. This demonstrates that MoRE can achieve stronger unlearning objectives beyond the conventional gold standard, establishing a new utility-focused and irreversible unlearning paradigm.

**Unlearning Efficiency Evaluation.** Beyond unlearning effectiveness, MoRE is also highly efficient and scalable. On CIFAR-10 and CIFAR-100, MoRE performs complete unlearning in under 10 seconds while consuming less than 200 MB of GPU memory (see Fig. 5). Remarkably, despite such modest compute and memory requirements, MoRE still achieves SOTA performance and strong irreversibility. These traits underscore the scalability of our framework to larger datasets and models, making it a practical solution for real-world deployment.

**Concept Unlearning on Diffusion Models.** To demonstrate broader applicability, we present results on concept unlearning in text-to-image generation model, namely Stable Diffusion v1.4. Concept unlearning is closely related to class-wise unlearning and is especially relevant in the era of generative models. We believe this extension is more appropriate and impactful.

Table 2: LPIPS scores for artistic style erasure for different unlearning methods.

| Method | Training-free? | Remove "Van Gogh" | | | Remove "Kelly McKernan" | | |
|---|---|---|---|---|---|---|---|
| | | LPIPS_f (↑) | LPIPS_r (↓) | LPIPS_d (↑) | LPIPS_f (↑) | LPIPS_r (↓) | LPIPS_d (↑) |
| CA [2] | X | 0.3 | 0.13 | 0.17 | 0.22 | 0.17 | 0.05 |
| ESD [3] | X | 0.4 | 0.26 | 0.14 | 0.37 | 0.21 | 0.16 |
| SLD-Medium [4] | O | 0.31 | 0.55 | -0.24 | 0.39 | 0.47 | -0.08 |
| SAFREE [5] | O | 0.42 | 0.31 | 0.11 | 0.4 | 0.39 | 0.01 |
| RECE [6] | O | 0.31 | 0.08 | 0.23 | 0.29 | 0.04 | 0.25 |
| UCE [7] | O | 0.25 | 0.05 | 0.2 | 0.25 | 0.03 | 0.22 |
| **Ours** | O | 0.33 | 0.08 | **0.25** | 0.33 | 0.07 | **0.26** |

Following prior works (Gong et al., 2024; Gandikota et al., 2024), we apply prototype orthogonalization, erasure, and remapping to the cross-attention layers, using tokenized input prompts to construct prototypes. Our experimental setup also strictly follows the standard practice established by the SOTA diffusion unlearning methods Gandikota et al. (2023); Yoon et al. (2024); Gong et al. (2024); Gandikota et al. (2024), ensuring a fair comparison.

We evaluate unlearning performance in the artistic style erasure task, which has emerged as a standard benchmark for testing concept-level unlearning in generative models. Following prior works (Gong et al., 2024; Yoon et al., 2024), we construct an evaluation set using 20 prompts for each of 10 artists: 5 classical (Van Gogh, Pablo Picasso, Rembrandt, Andy Warhol, Caravaggio) and 5 modern (Kelly McKernan, Thomas Kinkade, Tyler Edlin, Kilian Eng, and the anime series Ajin: DemiHuman), all of whom are reported to be mimicked by SD. We apply MoRE to remove two artistic styles: Van Gogh and Kelly McKernan. And the unlearning performance is measured using the LPIPS scores (Zhang et al., 2018), which compares the generated outputs before and after unlearning to measure visual similarity (lower means more similar). We report three metrics:

- LPIPS_f (forget artists): LPIPS score computed on the forget artists (higher is better)

- LPIPS_r (remain artists): LPIPS score on the remain artists (lower is better)

- LPIPS_d = LPIPS_f - LPIPS_r: the overall tradeoff, capturing how well the method removes target styles while preserving unrelated ones

As shown in Table 2, our proposed method achieves highly competitive performance across all three LPIPS-based metrics: demonstrating strong unlearning of the target style (high LPIPS_f), minimal distortion to remain styles (low LPIPS_r), and the best overall tradeoff (highest LPIPS_d). Qualitative results shown in Figure 4 further support these findings. Notably, ours is the only method that successfully removes Van Gogh's iconic artistic style while faithfully adhering to the input prompt, generating a coherent image of a starry night over a town without reproducing the signature spiral patterns or brush stroke textures.

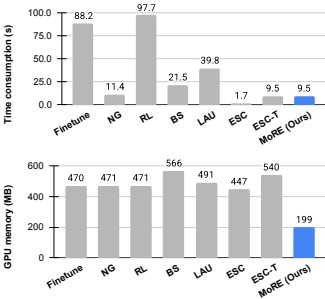

Figure 5: Comparison of time (*top*) and GPU memory consumption (*bottom*).

Table 3: Ablation table to demonstrate the effectiveness of the proposed techniques.

| CIFAR-10 | | | | | | | |
|---|---|---|---|---|---|---|---|
| Method | PO | D_f | D_r | D_ft | D_rt | HM | HM_t |
| Erase | X | 14.38 | 99.90 | 13.47 | 91.24 | 92.21 | 88.82 |
| Remap | | 0.00 | 89.52 | 0.00 | 79.64 | 94.47 | 88.67 |
| Erase | O | 0.00 | 99.94 | 0.00 | 91.67 | **99.97** | **95.65** |
| Remap | | 0.00 | 99.87 | 0.00 | 91.16 | 99.94 | 95.38 |
| MoRE | | 0.00 | 99.82 | 0.00 | 90.90 | 99.91 | 95.23 |
| CIFAR-10 (KR setting; lr = 0.1) | | | | | | | |
| Method | PO | D_f | D_r | D_ft | D_rt | HM | HM_t |
| Erase | X | 99.02 | 99.92 | 88.27 | 91.07 | 1.94 | 20.79 |
| Remap | | 99.87 | 99.94 | 91.27 | 90.97 | 0.27 | 15.94 |
| Erase | O | 99.01 | 99.92 | 88.27 | 91.07 | 1.97 | 20.79 |
| Remap | | 33.20 | 96.17 | 29.83 | 87.76 | 66.89 | 69.78 |
| MoRE | | 9.01 | 97.43 | 8.93 | 88.43 | **93.94** | **89.61** |

Table 4: Accuracy and MIA performance measured for random data forgetting task using ResNet-18 on CIFAR-10.

| CIFAR-10 | | | | | |
|---|---|---|---|---|---|
| Method | D_f | D_r | D_t | MIA | Avg. Gap |
| Retrain | 95.58 | 100.00 | 95.23 | 74.64 | - |
| Finetune | 99.85 | 100.00 | 95.58 | 87.73 | 4.43 |
| RL | 94.52 | 99.97 | 93.66 | 27.99 | 12.33 |
| SCRUB | 99.99 | 100.00 | 95.41 | 86.41 | 4.09 |
| BadT | 100.00 | 100.00 | 95.27 | 60.33 | 4.69 |
| SalUn | 100.00 | 99.99 | 95.18 | 63.61 | 3.88 |
| NG | 96.57 | 96.59 | 89.67 | 63.61 | 5.25 |
| ESC | 100.00 | 100.00 | 95.07 | 73.43 | 1.45 |
| ESC-T | 99.86 | 97.78 | 92.73 | 76.74 | 2.78 |
| Remap | 100.00 | 100.00 | 95.53 | 79.31 | 2.35 |

These results are highly significant; our proposed method is applied to diffusion models entirely out of the box, with no architecture-specific adaptation, no hyperparameter tuning and no additional engineering. Despite this, it outperforms SOTA diffusion model unlearning methods both quantitatively and qualitatively. Since our current implementation does not yet leverage any diffusion-specific components, we believe that modest, targeted adaptations could further amplify its effectiveness. This opens up a highly promising avenue for future research, with the potential to drive substantial impact across both the generative modeling and unlearning communities.

## 4.2 ABLATION STUDY

**Effectiveness of Prototype-Orthogonal Projection.** We perform ablation studies on the key components of our framework: PO projection, erasing, remapping, and the full MoRE architecture. Table 3 summarizes results on CIFAR-10, while additional results on CIFAR-100 are provided in the Appendix §C.2. Here, **Erase** refers to an ESC-like baseline that simply removes forget prototypes, while **Remap** denotes our proposed remapping strategy. **PO** indicates whether PO projection is applied. Without PO, erase fails to fully forget, with forget accuracy lingering around 14%. Remap can drive forget accuracy to zero, but at the cost of remain accuracy degradation.

Applying PO prior to unlearning yields the strongest results. By decorrelating prototype vectors, PO ensures that erasure and remapping precisely target forget prototypes without corrupting remain prototypes. Empirical evidence for this effect is shown in Fig. 6, which plots cosine similarities between prototypes after erasing and remapping. In contrast to Fig. 3, where unlearning perturbed remain prototypes, here remain prototypes preserve their autocorrelation close to one (diagonal entries). Meanwhile, the forget prototype exhibits a cosine similarity of 0 with itself under erasure, or a similarity close to 1 with the remain prototype of class 0 under remapping. These results validate the role of PO in disentangling prototypes and enabling precise editing of forget prototypes.

**Sensitivity Analysis for Target Remapping Class.** We also study the effect of the target prototype used for remapping (see Table 5). Using CIFAR-10 with class 4 as the forget class, we remap it to different remain classes. Performance remains strong across targets, though some yield slightly better results, suggesting mild preference. We leave deeper investigation to future work.

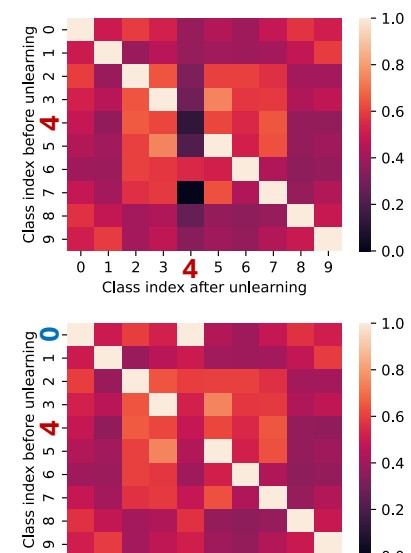

Figure 6: Cosine similarities between the mean activaitons after erasing (*top*) and remapping (*bottom*). Class **4** is unlearned and remapped to class **0**.

**Sensitivity Analysis for Number of Experts.** Figure 7 shows sensitivity analysis results with respect to the number of experts evaluated on CIFAR-10, CIFAR-100, and Tiny-ImageNet under the KR setting. HM performance is stable across a wide range of expert counts, indicating its robustness. The only exception is CIFAR-10 with a single expert, which reduces to basic remapping and causes a dip in performance. More detailed results are provided in the Appendix §C.3.

Figure 7: Impact of number of experts on unlearning performance.

Table 5: Unlearning performance measured with different target remapping classes using AllCNN on CIFAR-10.

| | CIFAR-10 | | | | | | | CIFAR-10 (KR setting; lr = 0.1) | | | | | |
|---|---|---|---|---|---|---|---|---|---|---|---|---|---|
| Target | D_f | D_r | D_ft | D_rt | HM | HM_t | Target | D_f | D_r | D_ft | D_rt | HM | HM_t |
| 0 | 0.00 | 99.87 | 0.00 | 91.16 | 99.94 | 95.38 | 0 | 33.20 | 96.17 | 29.83 | 87.76 | 66.89 | 69.78 |
| 1 | 0.00 | 99.87 | 0.00 | 90.76 | 99.93 | 95.16 | 1 | 33.27 | 96.20 | 30.03 | 87.36 | 66.75 | 69.34 |
| 2 | 0.00 | 99.84 | 0.00 | 91.06 | 99.92 | 95.32 | 2 | 66.29 | 92.48 | 62.00 | 84.46 | 34.06 | 40.38 |
| 3 | 0.00 | 99.75 | 0.00 | 90.68 | 99.87 | 95.11 | 3 | 36.40 | 95.63 | 34.23 | 87.34 | 65.00 | 66.02 |
| 5 | 0.00 | 99.74 | 0.00 | 90.68 | 99.87 | 95.11 | 5 | 33.51 | 96.14 | 31.27 | 87.69 | 66.67 | 67.39 |
| 6 | 0.00 | 99.77 | 0.00 | 90.94 | 99.88 | 95.26 | 6 | 33.26 | 96.14 | 31.03 | 87.49 | 66.75 | 67.77 |
| 7 | 0.00 | 99.85 | 0.00 | 91.11 | 99.93 | 95.35 | 7 | 66.49 | 92.50 | 61.33 | 84.19 | 33.67 | 41.49 |
| 8 | 0.00 | 99.86 | 0.00 | 90.76 | 99.93 | 95.16 | 8 | 67.05 | 92.49 | 60.60 | 83.93 | 33.42 | 43.37 |
| 9 | 0.00 | 99.84 | 0.00 | 90.69 | 99.92 | 95.12 | 9 | 89.95 | 89.62 | 80.90 | 81.34 | 15.24 | 29.26 |

**Stochastic vs. Conditional router.** While MoRE adopts stochastic routing as default for its compute efficiency, we also explored conditional variants, showcasing the potential of optimized routers (see Table 6). **MoRE-P** uses pseudo-inverse initialization of remain prototypes, while **MoRE-P-T-B** is a trained version with forget load balancing to prevent expert specialization collapse. We can see that the trained routers yield higher HM on CIFAR, but not consistently. Detailed analysis of methods and experiments are provided in the Appendix §D.

### 4.3 ADDITIONAL EXPERIMENTS

**Unlearning at Shallower Layers.** To assess generalizability, we applied unlearning at shallower layers (Table 7). It remains effective at the second-last layer but weakens at the third-last, highlighting sensitivity to layer choice and a promising avenue for future study. Additional KR results are given in Appendix §C.4.

**Random Data Forgetting.** We also evaluate MoRE on a random data forgetting task. For this setting, we adapt our method by computing prototypes (concept-wise feature means) separately for the forget and remain sets, and then remap each forget prototype to its counterpart from the remain set. As shown in Table 4, MoRE achieves comparable or superior performance to existing methods, outperforming most baselines in terms of average gap. This result is especially notable since our framework was not explicitly designed for random data forgetting, yet still delivers strong performance.

Table 6: Ablation table for stochastic random routing (MoRE) and conditional routing

| CIFAR-10 (N=8, KR setting; lr = 0.1) | | | | | | |
|---|---|---|---|---|---|---|
| Method | D_f | D_r | D_ft | D_rt | HM | HM_t |
| MoRE | 5.25 ± 3.77 | 96.35 ± 3.09 | 13.73 ± 6.64 | 84.27 ± 3.48 | 95.54 ± 3.42 | 85.24 ± 5.04 |
| MoRE-P | 5.89 ± 5.34 | 96.94 ± 2.04 | 6.33 ± 5.34 | 87.31 ± 2.05 | 95.47 ± 4.37 | 90.35 ± 4.01 |
| MoRE-P-T-B | 0.97 ± 0.37 | 95.48 ± 0.84 | 1.7 ± 0.61 | 86.08 ± 0.85 | **97.22 ± 0.61** | **91.79 ± 0.74** |

| CIFAR-100 (N=80, KR setting; lr = 0.1) | | | | | | |
|---|---|---|---|---|---|---|
| Method | D_f | D_r | D_ft | D_rt | HM | HM_t |
| MoRE | 0.03 ± 0.02 | 99.98 ± 0.00 | 1.90 ± 0.87 | 71.00 ± 0.19 | **99.97 ± 0.01** | 82.37 ± 0.30 |
| MoRE-P | 9.53 ± 1.09 | 98.13 ± 0.44 | 10.70 ± 0.75 | 68.15 ± 0.65 | 94.14 ± 0.71 | 81.25 ± 0.63 |
| MoRE-P-T-B | 4.53 ± 1.46 | 99.31 ± 0.16 | 6.30 ± 0.82 | 73.67 ± 0.42 | 97.34 ± 0.69 | **82.48 ± 0.56** |

Table 7: Unlearning performance when the proposed methods are applied to shallower layers.

| | | CIFAR-10 | | | | | |
|---|---|---|---|---|---|---|---|
| | Method | D_f | D_r | D_ft | D_rt | HM | HM_t |
| 2nd last | Erase | 0.95 ± 0.02 | 99.90 ± 0.00 | 0.50 ± 0.00 | 91.37 ± 0.01 | 99.47 ± 0.01 | **95.26 ± 0.00** |
| | Remap | 0.44 ± 0.00 | 99.52 ± 0.00 | 0.13 ± 0.06 | 90.53 ± 0.01 | **99.54 ± 0.00** | 94.97 ± 0.03 |
| | MoUE | 0.67 ± 0.02 | 99.60 ± 0.01 | 0.37 ± 0.06 | 90.73 ± 0.16 | 99.47 ± 0.02 | 94.98 ± 0.11 |
| 3rd last | Erase | 46.30 ± 0.15 | 98.07 ± 0.01 | 36.63 ± 0.38 | 88.87 ± 0.02 | 69.40 ± 0.13 | 73.98 ± 0.26 |
| | Remap | 24.26 ± 0.16 | 96.22 ± 0.05 | 18.43 ± 0.12 | 86.90 ± 0.04 | 84.76 ± 0.11 | 84.15 ± 0.08 |
| | MoUE | 25.28 ± 0.24 | 96.36 ± 0.09 | 19.47 ± 0.45 | 87.31 ± 0.10 | 84.17 ± 0.18 | 83.78 ± 0.25 |

| | | CIFAR-100 | | | | | |
|---|---|---|---|---|---|---|---|
| | Method | D_f | D_r | D_ft | D_rt | HM | HM_t |
| 2nd last | Erase | 5.20 ± 0.20 | 99.98 ± 0.00 | 0.60 ± 0.00 | 81.49 ± 0.00 | 97.32 ± 0.11 | **89.56 ± 0.00** |
| | Remap | 0.13 ± 0.12 | 99.98 ± 0.00 | 0.00 ± 0.00 | 80.71 ± 0.00 | **99.92 ± 0.06** | 89.33 ± 0.00 |
| | MoUE | 0.20 ± 0.20 | 99.98 ± 0.00 | 0.00 ± 0.00 | 80.38 ± 0.03 | 99.89 ± 0.10 | 89.12 ± 0.02 |
| 3rd last | Erase | 12.53 ± 1.33 | 99.78 ± 0.00 | 1.90 ± 0.00 | 79.13 ± 0.00 | 93.22 ± 0.75 | 87.60 ± 0.00 |
| | Remap | 8.67 ± 2.01 | 99.04 ± 0.00 | 1.60 ± 0.00 | 77.62 ± 0.01 | 95.02 ± 1.09 | 86.78 ± 0.00 |
| | MoUE | 10.00 ± 2.36 | 99.20 ± 0.01 | 2.07 ± 0.21 | 77.91 ± 0.17 | 94.36 ± 1.31 | 86.78 ± 0.18 |

## 5 CONCLUSION

We introduced MoRE (Mixture of Remapping Experts), a training-free framework for irreversible feature-level unlearning that combines prototype-orthogonal projection and expert-based remapping. Across diverse datasets and architectures, MoRE consistently outperforms existing methods, surpassing training-based baselines at a fraction of the cost. Unlike prior approaches that allow recovery of forgotten knowledge through linear probing, MoRE maintains accuracy near random-guess levels, delivering real-world unlearning guarantees stronger than retrain-from-scratch. These results position MoRE as both effective and efficient, paving the way for scalable deployment in practical systems. Beyond effectiveness, MoRE opens an entirely new avenues for research: how to train the router for unlearning experts, whether it can be both conditional and stochastic, how to better identify prototypes, how to optimally select experts and remapping vectors, and how the framework extends to generative models. By enabling both SOTA performance and rich opportunities for exploration, MoRE lays the groundwork for a new chapter in machine unlearning research.

ETHICS STATEMENT

This work adheres to the ICLR Code of Ethics. Our research does not involve human subjects, animal experiments, or the collection of sensitive personal data. The datasets used in this study are publicly available benchmarks that have been widely adopted in the research community and do not contain personally identifiable information. We have carefully considered the potential societal impacts of our method and believe that it does not pose direct risks of misuse or harmful applications. All computational experiments were conducted using institutional resources with appropriate permissions. The authors declare no conflicts of interest related to this work.

REPRODUCIBILITY STATEMENT

We provide all the details necessary to reproduce our method in the paper and appendix, including model architecture, training procedures, hyperparameters, evaluation metrics, and experimental setup. All datasets used are publicly available. Our code and trained models will be made publicly available upon acceptance to ensure full transparency and reproducibility of our results.

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

# APPENDIX

## A  PSEUDO CODE OF MoRE

---

**Algorithm 1** Construction of MoRE Layer

---

**Input:** Feature extractor $h_\psi(\cdot)$, classifier head $g_\phi(\cdot)$, dataset $\mathcal{D}$ with forget set $\mathcal{D}_f$ and remain set $\mathcal{D}_r$, number of experts $N_e$

**Output:** MoRE layer parameters $\{\mathbf{E}^{(e)}\}_{e=1}^{N_e}$

1: ▷ *Step 1: Compute prototypes via class-wise activation mean.*
2: Initialize $\mathbf{P}_{\text{sum}} \in \mathbb{R}^{d \times k} \leftarrow 0$, $\mathbf{C} \in \mathbb{R}^k \leftarrow 0$
3: **for** each $(x, y) \in \mathcal{D}$ **do**
4:     $\mathbf{z} \leftarrow h_\psi(x)$
5:     $\mathbf{P}_{\text{sum}}[:, y] \leftarrow \mathbf{P}_{\text{sum}}[:, y] + \mathbf{z}$
6:     $\mathbf{C}[y] \leftarrow \mathbf{C}[y] + 1$
7: **end for**
8: Normalize: $\mathbf{P}[:, c] \leftarrow \mathbf{P}_{\text{sum}}[:, c] / \max(\mathbf{C}[c], 1)$, $\forall c$
9:
10: ▷ *Step 2: Compute pseudo inverse of $\mathbf{P}$.*
11: Perform SVD: $\mathbf{P} = \mathbf{U}\boldsymbol{\Sigma}\mathbf{V}^T$
12: Compute projection matrix: $\mathbf{P}^+ \leftarrow \mathbf{V}\boldsymbol{\Sigma}^{-1}\mathbf{U}^T$
13:
14: ▷ *Step 3: Build remapping expert.*
15: Define binary selector $\mathbf{s} \in \{0, 1\}^k$, with $s_i = 1$ if class $i \in \mathcal{F}$
16: Compute $\mathbf{P}_f = \mathbf{P} \operatorname{diag}(\mathbf{s})$.
17: **for** $e = 1$ to $N_e$ **do**
18:     Initialize $\mathbf{P}_t^{(e)} \leftarrow 0$
19:     **for** each $i \in \mathcal{F}$ **do**
20:         Select $j \in \mathcal{R}$ uniformly at random
21:         $\mathbf{P}_t^{(e)}[:, i] \leftarrow \mathbf{P}[:, j]$
22:     **end for**
23:     Compute expert transform: $\mathbf{E}^{(e)} \leftarrow \mathbf{I} - \mathbf{P}_f \operatorname{diag}(\mathbf{s}) \mathbf{P}^+ + \mathbf{P}_t^{(e)} \operatorname{diag}(\mathbf{s}) \mathbf{P}^+$
24: **end for**
25:
26: **Return:** $\{\mathbf{E}^{(e)}\}_{e=1}^{N_e}$.

---

**Algorithm 2** MoRE Forward Pass (Stochastic Remapping)

---

**Input:** Sample $x$, feature extractor $h_\psi(\cdot)$, classifier $g_\phi(\cdot)$, expert transforms $\{\mathbf{E}^{(e)}\}_{e=1}^{N_e}$

**Output:** Prediction $\hat{y}$

1: Encode input: $\mathbf{z} \leftarrow h_\psi(x)$
2: Choose a random expert: $e \sim \text{Uniform}\{1, 2, \ldots, N_e\}$
3: Unlearn: $\hat{\mathbf{z}} \leftarrow \mathbf{E}^{(e)} \mathbf{z}$
4: Classify: $\hat{y} \leftarrow g_\phi(\hat{\mathbf{z}})$
5:
6: **Return:** $\hat{y}$

---

---

**Algorithm 3** Random Unlearning

---

**Input:** Feature extractor $h_\psi(\cdot)$, classifier head $g_\phi(\cdot)$, forget dataset $\mathcal{D}_f$, remain dataset $\mathcal{D}_r$, and weight parameters of the layer being unlearned $\mathbf{W}$

**Output:** Unlearned weight parameter $\hat{\mathbf{W}}$

1: ▷ *Step 1a: Compute forget prototypes via class-wise activation mean.*
2: Initialize $\mathbf{P}_{\text{sum},f} \in \mathbb{R}^{d \times k} \leftarrow 0$, $\mathbf{C} \in \mathbb{R}^k \leftarrow 0$
3: **for** each $(x, y) \in \mathcal{D}_f$ **do**
4:      $\mathbf{z} \leftarrow h_\psi(x)$
5:      $\mathbf{P}_{\text{sum},f}[:, y] \leftarrow \mathbf{P}_{\text{sum},f}[:, y] + \mathbf{z}$
6:      $\mathbf{C}[y] \leftarrow \mathbf{C}[y] + 1$
7: **end for**
8: Normalize: $\mathbf{P}_f[:, c] \leftarrow \mathbf{P}_{\text{sum},f}[:, c] / \max(\mathbf{C}[c], 1)$, $\forall c$
9:
10: ▷ *Step 1b: Compute retain prototypes via class-wise activation mean.*
11: Initialize $\mathbf{P}_{\text{sum},r} \in \mathbb{R}^{d \times k} \leftarrow 0$, $\mathbf{C} \in \mathbb{R}^k \leftarrow 0$
12: **for** each $(x, y) \in \mathcal{D}_r$ **do**
13:      $\mathbf{z} \leftarrow h_\psi(x)$
14:      $\mathbf{P}_{\text{sum},r}[:, y] \leftarrow \mathbf{P}_{\text{sum},r}[:, y] + \mathbf{z}$
15:      $\mathbf{C}[y] \leftarrow \mathbf{C}[y] + 1$
16: **end for**
17: Normalize: $\mathbf{P}_r[:, c] \leftarrow \mathbf{P}_{\text{sum},r}[:, c] / \max(\mathbf{C}[c], 1)$, $\forall c$
18:
19: ▷ *Step 2: Compute pseudo inverse of* $\mathbf{P}_f$.
20: Perform SVD: $\mathbf{P}_f = \mathbf{U}\boldsymbol{\Sigma}\mathbf{V}^T$
21: Compute projection matrix: $\mathbf{P}_f^+ \leftarrow \mathbf{V}\boldsymbol{\Sigma}^{-1}\mathbf{U}^T$
22:
23: ▷ *Step 3: Remap forget class-wise mean to retain class-wise mean.*
24: Compute remapping transform: $\mathbf{E} \leftarrow \mathbf{I} - \mathbf{P}_f \mathbf{P}_f^+ + \mathbf{P}_r \mathbf{P}_f^+$
25: Absorb remapping transform to weight: $\hat{\mathbf{W}} \leftarrow \mathbf{W}\mathbf{E}$
26:
27: **Return:** $\hat{\mathbf{W}}$.

---

## B   E­VALUATION D­ETAILS

### B.1   E­XPERIMENTAL S­ETUP

**Datasets, Models, and Setting.** We evaluate MoRE on a diverse set of image classification datasets, including CIFAR-10, CIFAR-100, Tiny-ImageNet, and ImageNet, to cover datasets of varying scale and complexity. For the class-wise unlearning task, we remove 10% of the classes, while for the instance-wise unlearning task, we randomly discard 10% of the training samples. Experiments are performed on representative architectures spanning different model families and sizes, namely All-CNN, ResNet-18, and Vision Transformer (ViT). All experiments are conducted on a single NVIDIA A100 GPU.

**Baselines.** We compare MoRE against a wide range of approximate unlearning methods. As reference points, we include **Original**, a model trained on the full dataset $\mathcal{D}$ either from scratch or from pre-trained weights, and **Retrain**, a model trained only on the remain data $\mathcal{D}_r$. For unlearning approaches, we consider both classical and recent methods. Classical baselines include Finetune, Fisher, and $\ell_1$-sparse. Recent specialized unlearning methods include NG, RL, BS, LAU, SCRUB, and SalUn.

**Model Training Settings.** We trained our AllCNN model with CIFAR-10 dataset from scratch, whereas ResNet-18 model was pretrained on ImageNet-1K and then trained on CIFAR-100, and ViT was pretrained on ImageNet-21K and then trained on TinyImageNet and ImageNet-1K.

**Evaluations Metrics.** We evaluate unlearning performance by partitioning the dataset into remain training data $\mathcal{D}_r$, forget training data $\mathcal{D}_f$, remain test data $\mathcal{D}_{rt}$, and forget test data $\mathcal{D}_{ft}$. For each

subset, we report accuracy after unlearning as well as the **Knowledge Retention (KR)** metric. Following prior work, we also employ the **Harmonic Mean (HM)** to provide a balanced measure of utility preservation and forgetting effectiveness.

## B.2 UNLEARNING BASELINES

**Finetune (Warnecke et al., 2021).** This method conducts additional training on the remain data only. This procedure reduces the model's dependence on the forget data while preserving utility on the rest.

**Fisher (Golatkar et al., 2020).** This method injects noise into model parameters with variance scaled by the Fisher Information Matrix of the remain data. The added noise removes information linked to the forget data while maintaining accuracy on the remain data.

**SCRUB (Kurmanji et al., 2023).** This method scrubs the model parameters so that they become indistinguishable from those of a model trained without the forget data. As a result, the influence of the forget data is removed without full retraining.

**$\ell_1$-sparse (Jia et al., 2023).** This method incorporates an $\ell_1$ regularization term into the optimization objective. The penalty shrinks irrelevant weights, which reduces the gap between approximate and exact unlearning.

**SalUn (Fan et al., 2024).** This method operates in two phases. First, it computes a weight saliency map from the gradients with respect to the NG loss. This step identifies the parameters that are the most responsible for the forget data. In the second phase, only the top k% of these salient parameters are updated and the RL loss is typically used as the forgetting signal.

**Negative Gradient (NG) (Thudi et al., 2022a).** This method updates the model parameters by applying gradient ascent to the forget data. The update direction maximizes the loss on the forget set, which directly counteracts the minimization objective used during the original training. As a result, NG enforces forgetting by pushing the model away from the learned decision boundaries associated with the forget data.

**Random Labeling (RL) (Golatkar et al. (2020)).** This method fine-tunes the original model with cross-entropy loss, but assigns random labels to the forget data. The use of incorrect labels disrupts the input label associations learned during training and gradually erases the influence of the forget set. In practice, RL is usually applied only to the random-labeled forget data. In some cases, RL also leverages the remain data to stabilize performance.

**Bounding Shrink (BS) (Chen et al. (2023)).** This method shifts the decision boundary of the model to replicate the behavior of a re-trained network. The forget class is pushed toward the borders of other clusters, while boundaries for the remain classes remain stable.

**Layer Attack Unlearning (LAU) (Kim et al. (2024)).** This method retrains the classification head while freezing the backbone. This approach efficiently reduces information in the output layer, but leaves residual knowledge within the feature extractor.

**ESC/ESC-T (Lee et al. (2025)).** ESC achieves feature-level unlearning by applying Singular Value Decomposition (SVD) to the representations of the forget set and projecting them onto the remain subspace, thereby pruning the leading principal directions. While this training-free approach is efficient, it may remove more knowledge than necessary.

To address this issue, ESC-T augments ESC with lightweight training. Instead of discarding entire principal directions, it learns a mask over the feature components, optimized with a penalized cross-entropy loss that discourages correct predictions on the forget set. After optimization, the mask is binarized to selectively suppress only the most critical directions related to the forget data. This fine-grained masking enables ESC-T to achieve a more precise trade-off between forgetting and retention compared to the original ESC.

## B.3 EVALUATION METRICS

**Knowledge Retention (KR) evaluation.** To evaluate feature-level unlearning performance, we adopt the Knowledge Retention Score (KR), a benchmark introduced in prior work on knowledge

deletion (Lee et al., 2025). KR is designed to quantify the extent to which unwanted knowledge persists in the feature extractor after unlearning. Unlike output-level metrics such as forget accuracy, remain/test accuracy, or MIA, KR directly evaluates the feature space of the unlearned model. A high KR score on the forget set indicates that the forget class is still linearly separable in the feature space, while a low score reflects effective removal of feature-level knowledge. At the same time, high KR scores in the remain set demonstrate that useful knowledge of the remain data has been preserved.

To implement KR, we employ linear probing (Kornblith et al., 2019; Lee et al., 2025). The feature extractor of the unlearned model is frozen and the classification head is randomly re-initialized, after which only the head is optimized. We train the classifier for 10 epochs with SGD using a batch size of 64 on the combined training set $D = D_f \cup D_r$. To evaluate robustness, we vary the learning rate across 0.1, 0.01, and 0.001, in addition to the setting used in the main results. These additional experiments under varying conditions enhance the reliability of the KR evaluation.

**HM Metric.** Evaluation of machine unlearning methods requires not only measuring how well the model forgets the target data but also how effectively it retains useful knowledge from the remain data. Evaluation based on a single metric, such as forget accuracy ($acc_f$) or remain accuracy ($acc_r$), is insufficient due to the inherent trade-off between forgetting and retaining performance. To address this, several trade-off metrics have been proposed in prior work, including the Arithmetic Mean, Geometric Mean, and Unlearning Score. Each of these metrics combines $acc_f$ and $acc_r$ into a single value, but their sensitivity to imbalance varies.

The Harmonic Mean (HM) has emerged as a particularly suitable choice because it penalizes large gaps between forgetting and retaining performance. HM combines $acc_f$ and $acc_r$ into a single balanced score. Formally, HM is defined as follows:

$$HM = \frac{2 \cdot (100 - acc_f) \cdot acc_r}{(100 - acc_f) + acc_r},$$

Here, $(100 - acc_f)$ represents the effectiveness of unlearning (lower accuracy in forget data is desirable), while $acc_r$ reflects the utility preserved in remain data. The harmonic mean ensures that high performance requires both aspects to be simultaneously satisfied. For example, a model that forgets perfectly but fails catastrophically in the remain set—or conversely, one that retains perfectly but does not forget at all—will still result in a low HM score. Only when both forgetting and retention are reasonably balanced does the HM achieve a high value, making it a robust indicator of unlearning quality.

We adopt the Harmonic Mean (HM) as the primary trade-off metric in our evaluation. Compared to alternative measures such as the Arithmetic or Geometric Mean, HM offers a stronger balance by penalizing extreme imbalances between forgetting and retaining performance. This property makes HM a reliable indicator of unlearning quality.

**MIA evaluation.** To evaluate the privacy guarantee of an unlearning method, we also adopt the widely used Membership Inference Attack (MIA). The attacker is a binary classifier trained to distinguish whether a given sample was part of the forget set $D_f$ or a held-out test set $D_t$, based on the cross-entropy loss values of the unlearned model. Specifically, the attacker is trained on a class-balanced dataset as follows:

$$D_{\text{train}}^b = \{(l(x_i, y_i), y_i^b)\},$$

where $l(x_i, y_i)$ is the cross-entropy loss of the unlearned model on sample $(x_i, y_i)$, and the binary label is set to 0 if $x_i \in D_t$, and 1 if $x_i \in D_f$. After training, the attacker is evaluated on a disjoint set $D_{\text{eval}}^b$, also balanced between $D_f$ and $D_t$. The unlearning model is considered vulnerable if the attacker achieves high accuracy on $D_{\text{eval}}^b$, as this indicates that the loss values of the forget set remain distinguishable, thereby indicating leakage of membership information.

On the other hand, an optimal defense corresponds to an attack accuracy of 50%, which is equivalent to random guessing and indicates that forget and test samples are indistinguishable. Retrain-from-scratch is regarded as the gold standard since it achieves this optimal defense in principle. In our evaluation, we report the attacker's success rate, where lower values indicate stronger privacy guarantees.

Table 8: Accuracy, KR and performance in the KD settings are evaluated using CIFAR-10 with AllCNN, CIFAR-100 with ResNet-18, and Tiny-ImageNet with ViT. The table presents the mean and standard deviation (mean ± std) across three trials with the best value highlighted in bold, as full results of previous experiments.

**CIFAR-10**

| Method | D_f | D_r | D_ft | D_rt | HM | HM_t | MIA |
|---|---|---|---|---|---|---|---|
| Original | 99.92 | 99.94 | 91.80 | 91.07 | 0.16 | 15.05 | 58.40 |
| Retrain | 0.00 | 99.15 | 0.00 | 92.04 | 99.57 | 95.86 | 50.80 |
| Finetune | 0.00 ± 0.00 | 90.86 ± 2.66 | 0.00 ± 0.00 | 84.89 ± 2.39 | 95.20 ± 0.80 | 91.83 ± 0.82 | 50.3 ± 0.65 |
| Fisher | 1.02 ± 0.53 | 93.16 ± 4.20 | 1.03 ± 0.62 | 84.91 ± 3.62 | 95.84 ± 1.63 | 91.39 ± 1.66 | 51.9 ± 0.78 |
| SCRUB | 0.09 ± 0.08 | 89.17 ± 0.81 | 1.00 ± 0.06 | 84.35 ± 0.18 | 94.17 ± 0.25 | 91.44 ± 0.07 | **50.8** ± 0.17 |
| ℓ-sparse | 0.00 ± 0.00 | 81.77 ± 3.56 | 0.00 ± 0.00 | 79.64 ± 3.65 | 89.97 ± 1.32 | 88.66 ± 1.42 | 48.5 ± 0.51 |
| SalUn | 1.37 ± 0.05 | 98.45 ± 0.01 | 1.29 ± 0.11 | 90.54 ± 0.08 | 98.53 ± 0.02 | 94.43 ± 0.03 | 52.0 ± 1.42 |
| NG | 6.35 ± 0.08 | 89.62 ± 0.22 | 5.72 ± 0.11 | 81.26 ± 0.17 | 91.59 ± 0.03 | 87.28 ± 0.03 | 49.5 ± 0.06 |
| RL | 0.00 ± 0.00 | 97.52 ± 0.29 | 0.00 ± 0.00 | 90.14 ± 0.06 | 98.74 ± 0.08 | 94.82 ± 0.02 | 49.7 ± 0.20 |
| BS | 9.87 ± 0.02 | 95.23 ± 0.01 | 1.00 ± 0.01 | 85.75 ± 0.01 | 92.61 ± 0.01 | 87.83 ± 0.01 | 50.1 ± 0.06 |
| Lau | 0.18 ± 0.00 | 86.57 ± 5.67 | 0.13 ± 0.01 | 79.51 ± 3.79 | 92.71 ± 1.84 | 88.53 ± 1.39 | 51.7 ± 0.50 |
| ESC | 11.22 ± 1.18 | 88.84 ± 0.40 | 10.70 ± 0.87 | 81.27 ± 0.50 | 88.81 ± 0.74 | 85.10 ± 0.65 | 53.4 ± 1.80 |
| ESC-T | 0.03 ± 0.02 | 88.88 ± 0.22 | 0.00 ± 0.00 | 81.21 ± 0.19 | 94.10 ± 0.12 | 89.63 ± 0.11 | 51.7 ± 1.10 |
| **Erase** | 0.00 ± 0.00 | 99.94 ± 0.00 | 0.00 ± 0.00 | 91.67 ± 0.01 | **99.97** ± 0.00 | **95.65** ± 0.00 | 55.3 ± 1.65 |
| **Remap** | 0.00 ± 0.00 | 99.87 ± 0.00 | 0.00 ± 0.00 | 91.16 ± 0.01 | 99.94 ± 0.00 | 95.38 ± 0.01 | 54.0 ± 1.53 |
| **MoRE** | 0.00 ± 0.00 | 99.87 ± 0.01 | 0.00 ± 0.00 | 91.02 ± 0.11 | 99.93 ± 0.01 | 95.30 ± 0.06 | 53.4 ± 0.70 |
| **MoRE-P** | 0.00 ± 0.00 | 99.76 ± 0.00 | 0.00 ± 0.00 | 90.84 ± 0.01 | 99.88 ± 0.00 | 95.20 ± 0.01 | 52.7 ± 2.75 |

**CIFAR-10 (KR setting; lr = 0.1)**

| Method | D_f | D_r | D_ft | D_rt | HM | HM_t |
|---|---|---|---|---|---|---|
| Original | 99.88 | 99.95 | 91.20 | 91.07 | 0.24 | 16.05 |
| Retrain | 72.62 | 97.06 | 72.90 | 88.01 | 42.71 | 41.44 |
| Finetune | 79.24 ± 11.93 | 93.80 ± 0.14 | 76.91 ± 3.90 | 87.30 ± 0.19 | 33.45 ± 21.55 | 36.36 ± 6.35 |
| Fisher | 97.31 ± 0.02 | 97.35 ± 0.10 | 87.56 ± 0.34 | 88.18 ± 0.26 | 5.23 ± 0.09 | 21.77 ± 0.79 |
| SCRUB | 67.37 ± 0.39 | 89.54 ± 1.47 | 65.03 ± 0.51 | 84.45 ± 0.71 | 47.82 ± 0.55 | 49.45 ± 0.42 |
| ℓ-sparse | 59.93 ± 2.32 | 85.61 ± 0.00 | 60.26 ± 0.95 | 83.39 ± 0.02 | 54.54 ± 1.99 | 53.81 ± 0.76 |
| SalUn | 93.19 ± 0.22 | 98.42 ± 0.00 | 86.13 ± 0.81 | 89.93 ± 0.02 | 12.70 ± 0.66 | 23.98 ± 1.83 |
| NG | 94.34 ± 0.09 | 98.05 ± 0.01 | 83.97 ± 0.04 | 88.80 ± 0.00 | 10.69 ± 0.28 | 27.16 ± 0.07 |
| RL | 87.17 ± 1.69 | 97.53 ± 0.00 | 81.19 ± 0.83 | 89.38 ± 0.02 | 22.55 ± 4.07 | 31.03 ± 1.53 |
| BS | 96.43 ± 0.01 | 98.90 ± 0.00 | 85.50 ± 0.05 | 89.36 ± 0.00 | 6.88 ± 0.02 | 24.95 ± 0.10 |
| Lau | 90.60 ± 0.31 | 94.29 ± 0.10 | 80.70 ± 0.32 | 85.37 ± 0.03 | 17.06 ± 0.83 | 31.47 ± 0.54 |
| ESC | 82.97 ± 0.23 | 93.45 ± 0.21 | 73.60 ± 0.17 | 84.98 ± 0.16 | 28.80 ± 0.34 | 40.28 ± 0.19 |
| ESC-T | 87.34 ± 1.11 | 94.14 ± 0.35 | 78.80 ± 1.21 | 85.24 ± 0.31 | 22.31 ± 1.75 | 33.94 ± 1.58 |
| **Erase** | 99.01 ± 0.04 | 99.92 ± 0.01 | 88.27 ± 0.12 | 91.07 ± 0.04 | 1.97 ± 0.08 | 20.79 ± 0.18 |
| **Remap** | 33.20 ± 57.50 | 96.17 ± 6.40 | 29.83 ± 51.67 | 87.76 ± 5.92 | 66.89 ± 57.24 | 69.78 ± 44.34 |
| **MoRE** | 10.79 ± 9.85 | 95.96 ± 3.36 | 12.87 ± 8.96 | 86.82 ± 3.30 | 92.37 ± 6.91 | 86.91 ± 6.17 |
| **MoRE-P** | 1.19 ± 0.67 | 99.21 ± 0.27 | 1.00 ± 0.72 | 90.16 ± 0.32 | **99.01** ± 0.45 | **94.37** ± 0.47 |

**CIFAR-100**

| Method | D_f | D_r | D_ft | D_rt | HM | HM_t | MIA |
|---|---|---|---|---|---|---|---|
| Original | 100.00 | 99.98 | 67.00 | 80.81 | 0.00 | 46.86 | 74.00 |
| Retrain | 0.00 | 96.64 | 0.00 | 72.00 | 98.29 | 83.72 | 54.00 |
| Finetune | 0.00 ± 0.00 | 87.41 ± 0.18 | 0.00 ± 0.00 | 69.39 ± 0.25 | 93.28 ± 0.06 | 81.93 ± 0.12 | 52.1 ± 22.22 |
| Fisher | 29.78 ± 185.24 | 50.61 ± 16.09 | 17.85 ± 102.89 | 39.33 ± 7.58 | 58.82 ± 6.49 | 53.19 ± 1.69 | 59.7 ± 28.67 |
| SCRUB | 3.17 ± 0.57 | 97.58 ± 0.01 | 1.00 ± 0.67 | 76.50 ± 0.01 | 97.20 ± 0.13 | 86.31 ± 0.15 | 66.9 ± 0.22 |
| ℓ-sparse | 0.00 ± 0.00 | 66.66 ± 3.49 | 0.00 ± 0.00 | 59.71 ± 2.30 | 80.00 ± 1.84 | 74.77 ± 1.43 | 48.3 ± 4.67 |
| SalUn | 1.71 ± 0.06 | 87.23 ± 0.76 | 1.26 ± 0.22 | 68.26 ± 0.36 | 92.43 ± 0.34 | 80.72 ± 0.12 | 58.6 ± 12.67 |
| NG | 29.19 ± 0.32 | 96.79 ± 0.02 | 5.65 ± 0.22 | 70.06 ± 0.09 | 81.79 ± 0.18 | 80.41 ± 0.04 | 74.6 ± 0.67 |
| RL | 1.27 ± 1.34 | 82.19 ± 0.27 | 1.00 ± 2.00 | 67.58 ± 0.04 | 89.70 ± 0.03 | 80.33 ± 0.16 | **54.2** ± 6.22 |
| BS | 1.93 ± 0.54 | 58.57 ± 10.75 | 1.00 ± 2.00 | 41.48 ± 4.33 | 73.34 ± 6.86 | 58.45 ± 5.39 | 63.9 ± 6.00 |
| Lau | 2.60 ± 0.00 | 96.35 ± 0.01 | 0.00 ± 0.00 | 70.08 ± 0.01 | 96.87 ± 0.00 | 82.41 ± 0.01 | 78.3 ± 0.22 |
| ESC | 0.50 ± 0.38 | 88.85 ± 0.32 | 0.00 ± 0.00 | 64.72 ± 0.17 | 93.77 ± 0.05 | 78.58 ± 0.10 | 75.5 ± 10.89 |
| ESC-T | 0.00 ± 0.00 | 99.03 ± 0.00 | 0.00 ± 0.00 | 73.88 ± 0.00 | 99.51 ± 0.00 | 84.98 ± 0.00 | 84.6 ± 2.89 |
| **Erase** | 0.00 ± 0.00 | 99.98 ± 0.00 | 0.00 ± 0.00 | 81.44 ± 0.00 | **99.99** ± 0.00 | **89.77** ± 0.00 | 52.3 ± 0.50 |
| **Remap** | 0.00 ± 0.00 | 99.98 ± 0.00 | 0.00 ± 0.00 | 80.82 ± 0.00 | **99.99** ± 0.00 | 89.39 ± 0.00 | 73.0 ± 1.53 |
| **MoRE** | 0.00 ± 0.00 | 99.98 ± 0.00 | 0.00 ± 0.00 | 80.22 ± 0.03 | **99.99** ± 0.00 | 89.03 ± 0.02 | 70.7 ± 0.98 |
| **MoRE-P** | 0.00 ± 0.00 | 99.98 ± 0.00 | 0.00 ± 0.00 | 79.68 ± 0.00 | **99.99** ± 0.00 | 88.69 ± 0.00 | 66.7 ± 0.90 |

**CIFAR-100 (KR setting; lr = 0.1)**

| Method | D_f | D_r | D_ft | D_rt | HM | HM_t |
|---|---|---|---|---|---|---|
| Original | 100.00 | 99.98 | 67.00 | 80.41 | 0.00 | 46.80 |
| Retrain | 57.20 | 97.50 | 58.00 | 71.60 | 59.43 | 52.96 |
| Finetune | 66.31 ± 2.46 | 92.88 ± 0.03 | 53.66 ± 0.89 | 72.70 ± 0.10 | 49.37 ± 2.89 | 56.59 ± 0.41 |
| Fisher | 52.58 ± 35.26 | 74.53 ± 4.88 | 34.09 ± 90.89 | 57.44 ± 2.05 | 57.28 ± 13.57 | 60.29 ± 15.43 |
| SCRUB | 85.76 ± 6.11 | 98.20 ± 0.00 | 56.63 ± 4.22 | 76.71 ± 0.06 | 24.41 ± 14.60 | 55.33 ± 2.69 |
| ℓ-sparse | 31.59 ± 13.63 | 76.38 ± 0.58 | 38.28 ± 4.22 | 67.36 ± 0.17 | 71.97 ± 4.50 | 64.36 ± 1.32 |
| SalUn | 81.26 ± 1.24 | 92.89 ± 0.11 | 56.38 ± 32.89 | 71.81 ± 0.04 | 31.12 ± 2.43 | 53.63 ± 19.94 |
| NG | 96.80 ± 0.03 | 99.65 ± 0.00 | 52.99 ± 0.67 | 74.94 ± 0.01 | 6.19 ± 0.09 | 57.76 ± 0.36 |
| RL | 70.59 ± 1.39 | 88.98 ± 0.12 | 56.85 ± 16.67 | 71.82 ± 0.13 | 44.16 ± 1.90 | 53.58 ± 9.79 |
| BS | 61.07 ± 15.71 | 86.49 ± 1.55 | 33.91 ± 6.00 | 60.65 ± 0.39 | 53.31 ± 11.80 | 63.18 ± 1.89 |
| Lau | 94.53 ± 0.04 | 99.39 ± 0.00 | 48.98 ± 2.00 | 74.18 ± 0.00 | 10.36 ± 0.12 | 60.42 ± 1.01 |
| ESC | 99.60 ± 0.00 | 99.92 ± 0.00 | 59.65 ± 1.56 | 75.90 ± 0.01 | 0.80 ± 0.00 | 52.65 ± 1.15 |
| ESC-T | 96.00 ± 0.00 | 99.03 ± 0.00 | 59.87 ± 0.06 | 75.35 ± 0.01 | 7.55 ± 0.21 | 50.51 ± 0.49 |
| **Erase** | 10.67 ± 0.51 | 99.98 ± 0.00 | 5.80 ± 1.65 | 80.90 ± 0.03 | 94.36 ± 0.29 | **87.04** ± 0.70 |
| **Remap** | 51.88 ± 15.94 | 94.23 ± 1.88 | 41.63 ± 10.56 | 76.24 ± 1.61 | 62.76 ± 14.71 | 65.84 ± 7.35 |
| **MoRE** | 0.07 ± 0.09 | 99.86 ± 0.18 | 1.27 ± 0.72 | 77.01 ± 0.06 | **99.89** ± 0.14 | 86.53 ± 0.31 |
| **MoRE-P** | 19.53 ± 6.20 | 97.26 ± 1.30 | 16.63 ± 4.10 | 76.40 ± 1.04 | 88.01 ± 4.25 | 79.71 ± 2.44 |

**Tiny-ImageNet**

| Method | D_f | D_r | D_ft | D_rt | HM | HM_t | MIA |
|---|---|---|---|---|---|---|---|
| Original | 98.20 | 96.45 | 96.00 | 90.29 | 3.53 | 7.66 | 56.00 |
| Retrain | 0.00 | 99.98 | 0.00 | 85.23 | 99.99 | 92.03 | 50.80 |
| Finetune | 0.00 ± 0.00 | 66.89 ± 18.98 | 0.00 ± 0.00 | 55.49 ± 6.69 | 80.12 ± 9.42 | 71.35 ± 4.48 | 58.0 ± 0.70 |
| NG | 0.01 ± 0.00 | 0.62 ± 0.00 | 0.00 ± 0.00 | 0.62 ± 0.00 | 1.24 ± 0.00 | 1.23 ± 0.00 | 49.5 ± 0.15 |
| RL | 10.83 ± 4.26 | 96.98 ± 0.05 | 9.19 ± 5.45 | 85.14 ± 0.10 | 92.79 ± 1.17 | 87.73 ± 1.26 | 57.6 ± 1.42 |
| BS | 36.07 ± 0.29 | 48.02 ± 0.19 | 35.43 ± 0.08 | 46.65 ± 0.16 | 54.84 ± 0.03 | 54.16 ± 0.05 | 62.1 ± 0.05 |
| Lau | 0.00 ± 0.00 | 95.55 ± 0.00 | 0.00 ± 0.00 | 89.73 ± 0.00 | 97.73 ± 0.00 | 94.59 ± 0.00 | **51.0** ± 0.02 |
| ESC | 0.10 ± 0.02 | 96.36 ± 0.05 | 0.03 ± 0.06 | 90.57 ± 0.05 | 98.10 ± 0.02 | 95.03 ± 0.05 | 50.2 ± 3.06 |
| ESC-T | 0.00 ± 0.00 | 96.44 ± 0.03 | 0.00 ± 0.00 | 90.57 ± 0.04 | **98.19** ± 0.02 | **95.05** ± 0.02 | 49.6 ± 0.69 |
| **Erase** | 0.13 ± 0.01 | 96.56 ± 0.01 | 0.27 ± 0.06 | 90.71 ± 0.01 | **98.19** ± 0.06 | 95.01 ± 0.03 | 55.9 ± 0.30 |
| **Remap** | 0.03 ± 0.01 | 96.18 ± 0.01 | 0.00 ± 0.00 | 90.13 ± 0.03 | 98.04 ± 0.01 | 94.81 ± 0.02 | 56.3 ± 1.18 |
| **MoRE** | 0.02 ± 0.01 | 96.16 ± 0.02 | 0.00 ± 0.00 | 90.01 ± 0.09 | 98.03 ± 0.01 | 94.74 ± 0.05 | 54.7 ± 1.79 |
| **MoRE-P** | 0.00 ± 0.00 | 95.19 ± 0.05 | 0.00 ± 0.00 | 89.12 ± 0.07 | 97.53 ± 0.03 | 94.25 ± 0.04 | 52.7 ± 2.42 |

**Tiny-ImageNet (KR setting; lr = 0.1)**

| Method | D_f | D_r | D_ft | D_rt | HM | HM_t |
|---|---|---|---|---|---|---|
| Original | 98.20 | 96.83 | 96.00 | 90.10 | 3.53 | 7.66 |
| Retrain | 78.57 | 99.87 | 76.00 | 80.70 | 35.29 | 37.00 |
| Finetune | 53.69 ± 5.46 | 71.19 ± 13.19 | 51.67 ± 10.00 | 59.63 ± 4.77 | 55.97 ± 0.44 | 53.20 ± 1.33 |
| NG | 17.11 ± 0.11 | 19.84 ± 0.33 | 16.10 ± 0.03 | 18.62 ± 0.14 | 32.02 ± 0.53 | 30.48 ± 0.24 |
| RL | 91.21 ± 0.25 | 97.49 ± 0.04 | 84.90 ± 0.01 | 85.30 ± 0.24 | 16.10 ± 0.71 | 25.66 ± 0.02 |
| BS | 72.25 ± 0.02 | 74.26 ± 0.03 | 68.83 ± 0.12 | 69.97 ± 0.02 | 40.41 ± 0.02 | 43.12 ± 0.10 |
| Lau | 96.69 ± 0.00 | 96.86 ± 0.00 | 89.70 ± 0.00 | 90.18 ± 0.00 | 6.40 ± 0.00 | 18.49 ± 0.00 |
| ESC | 15.78 ± 0.84 | 96.88 ± 0.02 | 13.67 ± 1.06 | 90.54 ± 0.08 | 90.10 ± 0.49 | 88.38 ± 0.52 |
| ESC-T | 95.47 ± 0.12 | 96.78 ± 0.03 | 89.57 ± 0.21 | 90.23 ± 0.11 | 8.66 ± 0.21 | 18.70 ± 0.33 |
| **Erase** | 32.91 ± 6.17 | 96.90 ± 0.03 | 28.20 ± 5.55 | 90.49 ± 0.07 | 79.18 ± 4.26 | 79.99 ± 3.41 |
| **Remap** | 55.18 ± 15.74 | 93.57 ± 1.65 | 51.03 ± 14.00 | 87.00 ± 1.46 | 59.60 ± 14.98 | 61.96 ± 11.95 |
| **MoRE** | 0.50 ± 0.07 | 96.52 ± 0.01 | 0.43 ± 0.25 | 89.94 ± 0.07 | **97.99** ± 0.04 | **94.51** ± 0.14 |
| **MoRE-P** | 52.13 ± 8.69 | 96.76 ± 0.04 | 44.00 ± 5.29 | 90.06 ± 0.08 | 63.74 ± 7.77 | 68.29 ± 3.99 |

# C   ADDITIONAL RESULTS

## C.1   MAIN RESULTS IN FULL

Here we present the main results table encompassing all unlearning baselines evaluated under various KR settings with learning rates of 0.001, 0.01, and 0.1. Table 8 shows results for no KR-Setting, and KR setting with learning rate set to 0.1. Table 9 shows results for KR settings with learning rate set to 0.001 and 0.01. Here we show 4 variants of our method, Erase which only performs erasing of the forget prototpye in the PO space, Remap which only performs one-to-one remapping for the forget prototype (can be also considered as a setting with one expert), MoRE which uses multiple remapping experts for one-to-many remapping of the forget prototypes and MoRE-P which uses conditional router initialized with the pseudo-inverse of **P**.

We also present results for ImageNet on ViT model. Due to limited compute resources, our investigation was limited to a few compute efficient unlearning methods. We also evaluate the methods under the KR setting with a learning rate set to 0.1. The results can be found in Table 10. We can

Table 9: Accuracy, KR and performance in the KD settings are evaluated using CIFAR-10 with AllCNN, CIFAR-100 with ResNet-18, and Tiny-ImageNet with ViT. The table reports results with learning rates 0.001 and 0.01 for KR, presented as mean and standard deviation (mean ± std) across three trials with the best value highlighted in bold

**CIFAR-10 (KR setting; lr = 0.001)**

| Method | D_f | D_r | D_ft | D_rt | HM | HM_t |
|---|---|---|---|---|---|---|
| Original | 99.94 | 99.88 | 91.80 | 90.87 | 0.12 | 15.04 |
| Retrain | 44.86 | 98.46 | 48.00 | 66.66 | 70.69 | 65.94 |
| Finetune | 37.18 ± 18.11 | 93.15 ± 0.08 | 36.82 ± 10.95 | 86.94 ± 0.12 | 74.72 ± 10.10 | 73.00 ± 5.49 |
| Fisher | 96.48 ± 0.62 | 96.54 ± 0.13 | 88.09 ± 1.09 | 87.50 ± 0.18 | 6.62 ± 2.13 | 20.87 ± 2.55 |
| SCRUB | 54.73 ± 30.34 | 88.09 ± 1.19 | 51.40 ± 28.92 | 83.23 ± 0.74 | 59.15 ± 24.32 | 60.80 ± 18.97 |
| $\ell_1$-sparse | 24.70 ± 8.19 | 84.84 ± 0.11 | 25.76 ± 5.66 | 82.71 ± 0.12 | 79.64 ± 2.14 | 78.15 ± 1.58 |
| SalUn | 83.39 ± 1.47 | 98.03 ± 0.00 | 77.06 ± 1.69 | 89.59 ± 0.03 | 28.32 ± 3.20 | 36.46 ± 2.75 |
| NG | 77.98 ± 0.28 | 97.12 ± 0.01 | 69.80 ± 0.18 | 87.96 ± 0.00 | 35.88 ± 0.49 | 44.96 ± 0.22 |
| RL | 62.36 ± 0.95 | 97.18 ± 0.02 | 57.13 ± 0.62 | 89.15 ± 0.02 | 54.24 ± 1.04 | 57.89 ± 0.51 |
| BS | 85.92 ± 0.00 | 98.62 ± 0.00 | 74.00 ± 0.02 | 89.06 ± 0.00 | 24.64 ± 0.00 | 40.25 ± 0.03 |
| Lau | 74.94 ± 0.80 | 93.39 ± 0.05 | 65.62 ± 1.87 | 85.17 ± 0.04 | 39.48 ± 1.16 | 48.94 ± 1.80 |
| ESC | 11.51 ± 0.47 | 93.45 ± 0.20 | 11.70 ± 0.69 | 85.30 ± 0.21 | 90.90 ± 0.32 | 86.78 ± 0.41 |
| ESC-T | 42.51 ± 1.64 | 93.21 ± 0.19 | 41.03 ± 1.31 | 84.80 ± 0.18 | 71.10 ± 1.24 | 69.56 ± 0.87 |
| **Erase** | 28.12 ± 1.73 | 99.94 ± 0.00 | 25.37 ± 1.25 | 91.62 ± 0.02 | 83.61 ± 1.17 | 82.26 ± 0.75 |
| **Remap** | 51.89 ± 15.14 | 94.19 ± 2.24 | 45.97 ± 13.65 | 85.97 ± 1.68 | 62.82 ± 14.56 | 65.76 ± 11.23 |
| **MoRE** | 6.02 ± 0.20 | 97.98 ± 0.08 | 9.43 ± 0.80 | 88.41 ± 0.09 | **95.94 ± 0.14** | 89.48 ± 0.40 |
| **MoRE-P** | 9.81 ± 1.17 | 96.33 ± 0.18 | 8.43 ± 1.25 | 87.62 ± 0.16 | 93.15 ± 0.71 | **89.55 ± 0.67** |

**CIFAR-10 (KR setting; lr = 0.01)**

| Method | D_f | D_r | D_ft | D_rt | HM | HM_t |
|---|---|---|---|---|---|---|
| Original | 99.94 | 99.94 | 91.50 | 91.07 | 0.12 | 15.55 |
| Retrain | 67.80 | 97.29 | 69.00 | 88.27 | 48.39 | 45.89 |
| Finetune | 67.55 ± 12.57 | 93.06 ± 0.09 | 66.16 ± 5.05 | 86.59 ± 0.09 | 47.75 ± 15.54 | 48.52 ± 5.63 |
| Fisher | 96.83 ± 0.35 | 97.09 ± 0.11 | 87.53 ± 0.82 | 87.99 ± 0.29 | 6.03 ± 1.21 | 21.78 ± 1.89 |
| SCRUB | 71.13 ± 3.45 | 88.81 ± 1.07 | 69.15 ± 2.63 | 83.81 ± 0.49 | 43.46 ± 4.34 | 45.02 ± 2.85 |
| $\ell_1$-sparse | 54.20 ± 3.27 | 84.69 ± 0.01 | 54.96 ± 0.83 | 82.50 ± 0.10 | 59.39 ± 2.31 | 58.25 ± 0.51 |
| SalUn | 91.81 ± 0.20 | 98.27 ± 0.00 | 85.59 ± 1.13 | 89.71 ± 0.02 | 15.09 ± 0.60 | 24.75 ± 2.50 |
| NG | 90.33 ± 0.12 | 97.91 ± 0.01 | 79.97 ± 0.10 | 88.61 ± 0.00 | 17.60 ± 0.33 | 32.67 ± 0.17 |
| RL | 81.85 ± 0.12 | 96.99 ± 0.01 | 76.46 ± 1.34 | 88.70 ± 0.02 | 30.57 ± 0.24 | 37.15 ± 2.11 |
| BS | 94.35 ± 0.00 | 98.89 ± 0.00 | 82.67 ± 0.01 | 89.25 ± 0.00 | 10.68 ± 0.01 | 29.03 ± 0.02 |
| Lau | 86.70 ± 0.41 | 94.03 ± 0.07 | 77.17 ± 0.11 | 85.36 ± 0.03 | 23.27 ± 0.94 | 36.03 ± 0.16 |
| ESC | 70.22 ± 0.88 | 93.34 ± 0.16 | 62.23 ± 1.21 | 85.16 ± 0.13 | 45.15 ± 0.99 | 52.32 ± 1.15 |
| ESC-T | 74.06 ± 0.96 | 93.17 ± 0.17 | 68.07 ± 0.93 | 84.70 ± 0.05 | 40.58 ± 1.17 | 46.37 ± 0.98 |
| **Erase** | 94.16 ± 0.01 | 99.94 ± 0.00 | 80.57 ± 0.02 | 91.36 ± 0.03 | 11.04 ± 0.19 | 32.05 ± 0.16 |
| **Remap** | 39.01 ± 49.89 | 95.71 ± 5.35 | 34.87 ± 44.31 | 87.29 ± 4.83 | 65.24 ± 49.26 | 68.42 ± 37.54 |
| **MoRE** | 8.88 ± 1.15 | 97.03 ± 0.46 | 11.43 ± 0.95 | 87.73 ± 0.44 | **93.98 ± 0.83** | 88.15 ± 0.64 |
| **MoRE-P** | 11.79 ± 3.68 | 96.31 ± 0.36 | 9.97 ± 3.57 | 87.55 ± 0.39 | 92.06 ± 2.20 | **88.76 ± 1.90** |

**CIFAR-100 (KR setting; lr = 0.001)**

| Method | D_f | D_r | D_ft | D_rt | HM | HM_t |
|---|---|---|---|---|---|---|
| Original | 100.00 | 99.98 | 67.00 | 79.67 | 0.00 | 46.67 |
| Retrain | 4.80 | 89.16 | 8.00 | 66.66 | 92.08 | 77.31 |
| Finetune | 10.17 ± 4.59 | 84.35 ± 0.63 | 10.49 ± 10.67 | 67.99 ± 0.16 | 86.87 ± 0.46 | 77.05 ± 1.01 |
| Fisher | 3.62 ± 9.34 | 26.00 ± 2.93 | 3.48 ± 4.67 | 22.30 ± 1.34 | 40.95 ± 4.42 | 36.23 ± 2.30 |
| SCRUB | 3.56 ± 2.73 | 95.30 ± 0.00 | 3.11 ± 1.56 | 74.19 ± 0.99 | 95.87 ± 0.68 | 84.03 ± 0.08 |
| $\ell_1$-sparse | 3.50 ± 3.24 | 63.23 ± 2.52 | 5.19 ± 1.56 | 57.73 ± 1.02 | 76.27 ± 1.05 | 71.71 ± 0.27 |
| SalUn | 73.35 ± 27.45 | 82.81 ± 1.31 | 56.83 ± 18.67 | 65.98 ± 0.65 | 40.32 ± 38.56 | 52.19 ± 10.08 |
| NG | 23.33 ± 0.06 | 97.65 ± 0.02 | 7.32 ± 0.22 | 71.70 ± 0.07 | 85.90 ± 0.02 | 80.85 ± 0.09 |
| RL | 60.17 ± 26.43 | 76.84 ± 1.43 | 51.67 ± 72.22 | 64.87 ± 0.26 | 52.46 ± 17.21 | 55.39 ± 33.28 |
| BS | 0.52 ± 0.17 | 51.70 ± 6.23 | 0.67 ± 0.22 | 39.27 ± 2.28 | 67.99 ± 5.15 | 56.28 ± 2.56 |
| Lau | 1.67 ± 0.43 | 97.28 ± 0.01 | 0.67 ± 0.22 | 71.71 ± 0.03 | 97.80 ± 0.11 | 83.29 ± 0.07 |
| ESC | 0.80 ± 0.43 | 81.60 ± 0.02 | 0.67 ± 0.89 | 59.64 ± 0.01 | 89.54 ± 0.12 | 74.53 ± 0.10 |
| ESC-T | 0.13 ± 0.01 | 99.39 ± 0.00 | 0.67 ± 0.00 | 74.80 ± 0.00 | 99.63 ± 0.00 | 85.34 ± 0.00 |
| **Erase** | 1.20 ± 0.67 | 99.98 ± 0.00 | 0.70 ± 0.26 | 81.19 ± 0.09 | 99.39 ± 0.34 | **89.34 ± 0.06** |
| **Remap** | 52.79 ± 1.63 | 94.20 ± 0.20 | 36.97 ± 0.80 | 76.05 ± 0.16 | 62.89 ± 1.48 | 68.93 ± 0.54 |
| **MoRE** | 0.00 ± 0.00 | 99.98 ± 0.00 | 0.00 ± 0.00 | 80.11 ± 0.14 | **99.99 ± 0.00** | 88.96 ± 0.09 |
| **MoRE-P** | 2.80 ± 0.60 | 99.43 ± 0.13 | 3.40 ± 0.66 | 78.89 ± 0.18 | 98.30 ± 0.36 | 86.83 ± 0.36 |

**CIFAR-100 (KR setting; lr = 0.01)**

| Method | D_f | D_r | D_ft | D_rt | HM | HM_t |
|---|---|---|---|---|---|---|
| Original | 100.00 | 99.98 | 67.00 | 80.23 | 0.00 | 46.76 |
| Retrain | 40.80 | 96.08 | 46.00 | 71.53 | 73.26 | 61.54 |
| Finetune | 53.25 ± 1.74 | 90.53 ± 0.03 | 48.65 ± 1.56 | 72.04 ± 0.28 | 61.66 ± 1.26 | 59.96 ± 1.01 |
| Fisher | 40.67 ± 94.89 | 66.95 ± 7.67 | 27.76 ± 62.00 | 53.01 ± 2.68 | 61.42 ± 17.13 | 60.41 ± 3.16 |
| SCRUB | 57.59 ± 9.79 | 97.59 ± 0.01 | 49.56 ± 10.89 | 76.50 ± 0.03 | 45.97 ± 13.88 | 60.80 ± 6.00 |
| $\ell_1$-sparse | 22.59 ± 6.41 | 73.66 ± 2.16 | 30.37 ± 17.56 | 65.41 ± 0.21 | 75.37 ± 0.20 | 67.22 ± 2.92 |
| SalUn | 76.87 ± 0.14 | 90.33 ± 0.03 | 56.28 ± 6.22 | 70.97 ± 0.02 | 36.83 ± 0.26 | 54.11 ± 3.46 |
| NG | 90.33 ± 0.01 | 99.25 ± 0.00 | 44.98 ± 2.00 | 74.17 ± 0.01 | 17.62 ± 0.02 | 63.18 ± 0.80 |
| RL | 67.79 ± 1.31 | 86.03 ± 0.30 | 59.47 ± 22.89 | 70.89 ± 0.24 | 46.83 ± 1.48 | 51.11 ± 14.13 |
| BS | 31.58 ± 24.83 | 80.74 ± 2.46 | 18.64 ± 14.00 | 57.20 ± 0.55 | 74.07 ± 4.79 | 67.17 ± 1.50 |
| Lau | 79.67 ± 0.00 | 98.73 ± 0.00 | 44.66 ± 0.22 | 73.64 ± 0.01 | 33.72 ± 0.02 | 63.19 ± 0.11 |
| ESC | 31.42 ± 3.10 | 99.68 ± 0.00 | 7.27 ± 0.89 | 76.23 ± 0.02 | 81.26 ± 1.50 | 83.67 ± 0.13 |
| ESC-T | 80.25 ± 2.04 | 99.72 ± 0.00 | 49.29 ± 4.22 | 75.16 ± 0.01 | 32.97 ± 4.00 | 60.56 ± 2.07 |
| **Erase** | 4.03 ± 1.35 | 99.98 ± 0.00 | 2.00 ± 0.89 | 81.31 ± 0.07 | 97.93 ± 0.70 | **88.88 ± 0.33** |
| **Remap** | 57.92 ± 5.00 | 93.41 ± 0.54 | 42.33 ± 4.38 | 75.56 ± 0.35 | 57.91 ± 4.90 | 65.34 ± 2.77 |
| **MoRE** | 0.02 ± 0.00 | 99.98 ± 0.00 | 1.57 ± 0.29 | 77.37 ± 0.14 | **99.98 ± 0.00** | 86.64 ± 0.18 |
| **MoRE-P** | 10.33 ± 3.13 | 98.21 ± 0.85 | 12.20 ± 4.56 | 76.35 ± 0.54 | 93.73 ± 2.09 | 81.65 ± 2.29 |

**Tiny-ImageNet (KR setting; lr = 0.001)**

| Method | D_f | D_r | D_ft | D_rt | HM | HM_t |
|---|---|---|---|---|---|---|
| Original | 98.20 | 96.45 | 96.00 | 90.29 | 3.53 | 7.66 |
| Retrain | 13.34 | 99.97 | 12.80 | 84.48 | 92.84 | 85.82 |
| Finetune | 10.82 ± 1.51 | 24.84 ± 5.25 | 9.64 ± 1.31 | 22.60 ± 2.40 | 38.82 ± 7.90 | 36.14 ± 4.12 |
| NG | 1.16 ± 0.02 | 2.06 ± 0.01 | 1.17 ± 0.18 | 1.04 ± 0.03 | 4.03 ± 0.03 | 3.72 ± 0.05 |
| RL | 73.15 ± 0.96 | 93.00 ± 0.37 | 69.86 ± 0.38 | 82.15 ± 0.17 | 41.63 ± 1.26 | 44.08 ± 0.36 |
| BS | 35.02 ± 0.08 | 42.18 ± 0.10 | 34.20 ± 0.32 | 41.48 ± 0.06 | 51.15 ± 0.03 | 50.88 ± 0.05 |
| Lau | 95.89 ± 0.00 | 94.76 ± 0.00 | 89.70 ± 0.00 | 88.67 ± 0.00 | 7.88 ± 0.00 | 18.46 ± 0.00 |
| ESC | 0.27 ± 0.11 | 94.64 ± 0.10 | 0.20 ± 0.17 | 88.88 ± 0.02 | 97.19 ± 0.06 | 94.02 ± 0.04 |
| ESC-T | 16.73 ± 3.14 | 95.63 ± 0.04 | 13.87 ± 2.14 | 89.90 ± 0.04 | 89.00 ± 1.77 | 87.97 ± 1.09 |
| **Erase** | 0.09 ± 0.06 | 95.84 ± 0.05 | 0.13 ± 0.23 | 90.12 ± 0.17 | **97.84 ± 0.01** | **94.74 ± 0.01** |
| **Remap** | 47.75 ± 2.73 | 90.73 ± 0.37 | 46.33 ± 2.93 | 85.02 ± 0.16 | 66.29 ± 2.32 | 65.77 ± 2.23 |
| **MoRE** | 0.00 ± 0.00 | 95.42 ± 0.04 | 0.00 ± 0.00 | 89.37 ± 0.21 | 97.66 ± 0.02 | 94.38 ± 0.12 |
| **MoRE-P** | 3.13 ± 0.76 | 93.92 ± 0.08 | 3.57 ± 0.84 | 88.12 ± 0.07 | 95.37 ± 0.35 | 92.09 ± 0.34 |

**Tiny-ImageNet (KR setting; lr = 0.01)**

| Method | D_f | D_r | D_ft | D_rt | HM | HM_t |
|---|---|---|---|---|---|---|
| Original | 98.00 | 96.21 | 96.00 | 90.11 | 3.92 | 7.66 |
| Retrain | 59.34 | 99.95 | 58.10 | 82.97 | 57.80 | 55.68 |
| Finetune | 31.75 ± 6.68 | 63.21 ± 18.79 | 29.60 ± 8.28 | 54.56 ± 6.98 | 65.47 ± 1.06 | 61.35 ± 0.63 |
| NG | 11.46 ± 0.23 | 10.95 ± 0.04 | 10.26 ± 0.83 | 10.30 ± 0.04 | 19.50 ± 0.09 | 18.48 ± 0.09 |
| RL | 87.56 ± 0.16 | 96.23 ± 0.09 | 83.43 ± 0.10 | 85.09 ± 0.23 | 22.02 ± 0.37 | 27.73 ± 0.20 |
| BS | 66.37 ± 0.09 | 67.10 ± 0.03 | 62.66 ± 0.22 | 64.94 ± 0.02 | 44.80 ± 0.07 | 47.41 ± 0.12 |
| Lau | 96.49 ± 0.00 | 96.18 ± 0.00 | 90.70 ± 0.00 | 90.04 ± 0.00 | 6.77 ± 0.00 | 16.86 ± 0.00 |
| ESC | 0.68 ± 0.10 | 96.30 ± 0.01 | 0.73 ± 0.23 | 90.50 ± 0.08 | 97.79 ± 0.05 | 94.68 ± 0.11 |
| ESC-T | 89.20 ± 0.40 | 96.29 ± 0.02 | 82.33 ± 0.45 | 90.36 ± 0.03 | 19.42 ± 0.65 | 29.55 ± 0.63 |
| **Erase** | 7.92 ± 2.92 | 96.44 ± 0.04 | 7.07 ± 3.28 | 90.70 ± 0.18 | 94.20 ± 1.53 | 91.79 ± 1.55 |
| **Remap** | 50.97 ± 10.69 | 92.50 ± 0.82 | 48.20 ± 9.93 | 86.66 ± 0.68 | 63.66 ± 9.24 | 64.50 ± 7.76 |
| **MoRE** | 0.02 ± 0.02 | 96.04 ± 0.01 | 0.00 ± 0.00 | 89.93 ± 0.07 | **97.97 ± 0.00** | **94.70 ± 0.04** |
| **MoRE-P** | 19.47 ± 6.74 | 96.21 ± 0.02 | 16.67 ± 3.40 | 90.10 ± 0.13 | 87.52 ± 4.11 | 86.55 ± 1.86 |

Table 10: Accuracy, KR and performance in the KD settings are evaluated using ImageNet-1K with ViT. The table reports results with learning rate 0.1 for KR.

| | **ImageNet-1K** | | | | | | | **ImageNet-1K (KR setting; lr = 0.1)** | | | | | |
|---|---|---|---|---|---|---|---|---|---|---|---|---|---|
| **Method** | **D_f** | **D_r** | **D_ft** | **D_rt** | **HM** | **HM_t** | **MIA** | **Method** | **D_f** | **D_r** | **D_ft** | **D_rt** | **HM** | **HM_t** |
| Original | 87.88 | 88.11 | 81.44 | 81.10 | 0.16 | 15.05 | 58.40 | Original | 87.88 | 88.11 | 81.44 | 81.10 | 0.24 | 16.05 |
| Lau | 0.12 | 9.94 | 0.04 | 8.98 | 18.08 | 16.49 | 51.70 | Lau | 89.16 | 88.48 | 79.48 | 78.21 | 19.32 | 32.51 |
| ESC | 0.46 | 83.93 | 0.64 | 77.12 | 91.07 | 86.84 | 49.76 | ESC | 65.56 | 86.52 | 60.12 | 78.97 | 49.27 | 53.00 |
| ESC-T | 0.00 | 86.77 | 0.00 | 79.95 | 92.92 | 88.86 | 50.22 | ESC-T | 85.12 | 86.25 | 78.84 | 78.63 | 25.38 | 33.35 |
| **Erase** | 0.24 | 86.47 | 0.08 | 79.61 | 92.64 | 88.62 | 49.94 | **Erase** | 84.32 | 85.49 | 78.02 | 77.52 | 26.50 | 34.25 |
| **Remap** | 1.56 | 86.66 | 1.38 | 79.80 | 92.18 | 88.22 | 49.66 | **Remap** | 82.54 | 84.34 | 75.80 | 76.58 | 28.93 | 36.78 |
| **MoRE** | 1.36 | 86.72 | 1.26 | 79.83 | 92.30 | 88.28 | 49.52 | **MoRE** | 44.00 | 82.59 | 39.94 | 74.78 | 66.74 | 66.62 |

see that the proposed method outperforms all baselines, showing its effectiveness in larger datasets such as ImageNet.

Table 11: Unlearning performance when the proposed methods are applied to the shallower layers. Accuracy, KR and performance in the KD settings are evaluated using CIFAR-10 with AllCNN, and CIFAR-100 with ResNet-18. The table reports results with learning rate 0.1 for KR, presented as mean and standard deviation (mean ± std) across three trials with the best value highlighted in bold

**CIFAR-10**

| Method | PO | D_f | D_r | D_ft | D_rt | HM | HM_t |
|---|---|---|---|---|---|---|---|
| Original | | 99.92 | 99.94 | 91.80 | 91.07 | 0.16 | 15.05 |
| Retrain | | 0.00 | 99.15 | 0.00 | 92.04 | 99.57 | 95.86 |
| Erase | X | 14.38 ± 0.07 | 99.90 ± 0.00 | 13.47 ± 0.15 | 91.24 ± 0.02 | 92.21 ± 0.04 | 88.82 ± 0.08 |
| Remap | X | 0.00 ± 0.00 | 89.52 ± 0.04 | 0.00 ± 0.00 | 79.64 ± 0.03 | 94.47 ± 0.02 | 88.67 ± 0.02 |
| Erase | O | 0.00 ± 0.00 | 99.94 ± 0.00 | 0.00 ± 0.00 | 91.67 ± 0.01 | **99.97 ± 0.00** | **95.65 ± 0.00** |
| Remap | O | 0.00 ± 0.00 | 99.87 ± 0.00 | 0.00 ± 0.00 | 91.16 ± 0.01 | 99.94 ± 0.00 | 95.38 ± 0.01 |
| MoRE | O | 0.00 ± 0.00 | 99.82 ± 0.02 | 0.00 ± 0.00 | 90.90 ± 0.08 | 99.91 ± 0.01 | 95.23 ± 0.05 |

**CIFAR-10 (KR setting; lr = 0.1)**

| Method | PO | D_f | D_r | D_ft | D_rt | HM | HM_t |
|---|---|---|---|---|---|---|---|
| Original | | 99.88 | 99.95 | 91.20 | 91.07 | 0.24 | 16.05 |
| Retrain | | 72.62 | 97.06 | 72.90 | 88.01 | 42.71 | 41.44 |
| Erase | X | 99.02 ± 0.02 | 99.92 ± 0.00 | 88.27 ± 0.12 | 91.07 ± 0.02 | 1.94 ± 0.04 | 20.79 ± 0.18 |
| Remap | X | 99.87 ± 0.01 | 99.94 ± 0.00 | 91.27 ± 0.25 | 90.97 ± 0.03 | 0.27 ± 0.02 | 15.94 ± 0.42 |
| Erase | O | 99.01 ± 0.04 | 99.92 ± 0.01 | 88.27 ± 0.12 | 91.07 ± 0.04 | 1.97 ± 0.08 | 20.79 ± 0.18 |
| Remap | O | 33.20 ± 57.50 | 96.17 ± 6.40 | 29.83 ± 51.67 | 87.76 ± 5.92 | 66.89 ± 57.24 | 69.78 ± 44.34 |
| MoRE | O | 9.01 ± 11.84 | 97.43 ± 3.14 | 8.93 ± 10.90 | 88.43 ± 2.96 | **93.94 ± 7.95** | **89.61 ± 6.92** |

**CIFAR-100**

| Method | PO | D_f | D_r | D_ft | D_rt | HM | HM_t |
|---|---|---|---|---|---|---|---|
| Original | | 100.00 | 99.98 | 67.00 | 80.81 | 0.00 | 46.86 |
| Retrain | | 0.00 | 96.64 | 0.00 | 72.00 | 98.29 | 83.72 |
| Erase | X | 0.00 ± 0.00 | 99.98 ± 0.00 | 0.00 ± 0.00 | 79.89 ± 0.00 | **99.99 ± 0.00** | 88.82 ± 0.00 |
| Remap | X | 0.00 ± 0.00 | 99.91 ± 0.00 | 0.00 ± 0.00 | 75.24 ± 0.00 | 99.96 ± 0.00 | 85.87 ± 0.00 |
| Erase | O | 0.00 ± 0.00 | 99.98 ± 0.00 | 0.00 ± 0.00 | 81.44 ± 0.00 | **99.99 ± 0.00** | **89.77 ± 0.00** |
| Remap | O | 0.00 ± 0.00 | 99.98 ± 0.00 | 0.00 ± 0.00 | 80.82 ± 0.00 | **99.99 ± 0.00** | 89.39 ± 0.00 |
| MoRE | O | 0.00 ± 0.00 | 99.98 ± 0.00 | 0.00 ± 0.00 | 80.17 ± 0.02 | **99.99 ± 0.00** | 88.99 ± 0.01 |

**CIFAR-100 (KR setting; lr = 0.1)**

| Method | PO | D_f | D_r | D_ft | D_rt | HM | HM_t |
|---|---|---|---|---|---|---|---|
| Original | | 100.00 | 99.98 | 67.00 | 80.41 | 0.00 | 46.80 |
| Retrain | | 57.20 | 97.50 | 58.00 | 71.60 | 59.43 | 52.96 |
| Erase | X | 100.0 ± 0.00 | 99.98 ± 0.00 | 80.37 ± 0.21 | 80.19 ± 0.09 | 0.00 ± 0.00 | 31.54 ± 0.26 |
| Remap | X | 100.0 ± 0.00 | 99.98 ± 0.00 | 78.77 ± 0.21 | 80.32 ± 0.07 | 0.00 ± 0.00 | 33.59 ± 0.27 |
| Erase | O | 10.67 ± 0.51 | 99.98 ± 0.00 | 5.80 ± 1.65 | 80.90 ± 0.03 | 94.36 ± 0.29 | 87.04 ± 0.70 |
| Remap | O | 51.88 ± 15.94 | 94.23 ± 1.88 | 41.63 ± 10.56 | 76.24 ± 1.61 | 62.76 ± 14.71 | 65.84 ± 7.35 |
| MoRE | O | 0.24 ± 0.21 | 99.73 ± 0.22 | 1.13 ± 0.42 | 78.39 ± 0.34 | **99.75 ± 0.21** | **87.44 ± 0.37** |

**Tiny-ImageNet**

| Method | PO | D_f | D_r | D_ft | D_rt | HM | HM_t |
|---|---|---|---|---|---|---|---|
| Original | | 98.20 | 96.45 | 96.00 | 90.29 | 3.53 | 7.66 |
| Retrain | | 0.00 | 99.98 | 0.00 | 84.05 | 99.99 | 91.33 |
| Erase | X | 0.00 ± 0.00 | 95.84 ± 0.02 | 0.00 ± 0.00 | 90.03 ± 0.01 | 97.87 ± 0.01 | 94.76 ± 0.01 |
| Remap | X | 0.00 ± 0.00 | 90.04 ± 0.07 | 0.00 ± 0.00 | 83.94 ± 0.13 | 94.76 ± 0.04 | 91.27 ± 0.08 |
| Erase | O | 0.13 ± 0.01 | 96.56 ± 0.01 | 0.27 ± 0.06 | 90.71 ± 0.04 | **98.19 ± 0.00** | **95.01 ± 0.03** |
| Remap | O | 0.03 ± 0.01 | 96.18 ± 0.01 | 0.00 ± 0.00 | 90.13 ± 0.03 | 98.04 ± 0.01 | 94.81 ± 0.02 |
| MoRE | O | 0.02 ± 0.01 | 96.11 ± 0.02 | 0.00 ± 0.00 | 89.92 ± 0.08 | 98.01 ± 0.01 | 94.69 ± 0.05 |

**Tiny-ImageNet (KR setting; lr = 0.1)**

| Method | PO | D_f | D_r | D_ft | D_rt | HM | HM_t |
|---|---|---|---|---|---|---|---|
| Original | | 98.20 | 96.83 | 96.00 | 90.10 | 3.53 | 7.66 |
| Retrain | | 81.00 | 99.97 | 84.00 | 83.28 | 31.93 | 26.84 |
| Erase | X | 96.20 ± 0.00 | 96.84 ± 0.02 | 90.13 ± 0.35 | 90.24 ± 0.06 | 7.31 ± 0.00 | 17.79 ± 0.57 |
| Remap | X | 96.53 ± 0.42 | 96.84 ± 0.04 | 90.00 ± 0.20 | 90.30 ± 0.10 | 6.69 ± 0.78 | 18.01 ± 0.33 |
| Erase | O | 32.91 ± 6.17 | 96.90 ± 0.03 | 28.20 ± 5.55 | 90.49 ± 0.07 | 79.18 ± 4.26 | 79.99 ± 3.41 |
| Remap | O | 55.18 ± 15.74 | 93.57 ± 1.65 | 51.03 ± 14.00 | 87.00 ± 1.46 | 59.60 ± 14.98 | 61.96 ± 11.95 |
| MoRE | O | 1.99 ± 0.92 | 96.63 ± 0.10 | 1.63 ± 0.65 | 90.11 ± 0.13 | **97.31 ± 0.50** | **94.06 ± 0.28** |

## C.2 Effectiveness of Prototype-Orthogonal Projection in Full

Here we present our full results for the ablation studies, which not only include results for CIFAR-10 dataset but also results for CIFAR-100 and Tiny-ImageNet datasets. The results can be found in Table 11.

## C.3 Sensitivity Analysis for Number of Experts in Full

Here we present our full results for the sensitivity analysis for the number of experts results with specific numbers including both mean and standard deviation. Results are shown in Table 12.

## C.4 Unlearning at Shallower Layers in Full

Here we present our full results for unlearning at shallower layers for CIFAR-100 dataset as well, using ResNet-18. Results are shown in Table 13.

## D Additional Experiments on Conditional Router

**Stochastic vs. Conditional Routing.** We adopt the stochastic random routing approach that assigns data randomly to each expert without implementing a dedicated router. While routers play a role in existing Mixture of Experts research, we experimented with conditional routing approaches, but eventually opted for random routing as the most reasonable solution, achieving satisfactory performance nonetheless. This section offers detailed experiments and analysis covering our different approaches. We discuss various router development attempts from two angles: (1) **Initialization** and (2) **Finetuning** strategies.

Table 12: Impact of number of experts on unlearning performance. Accuracy, KR and performance in the KD settings are evaluated using CIFAR-10 with AllCNN, CIFAR-100 with ResNet-18, and Tiny-ImageNet with ViT. The table reports results with learning rate 0.1 for KR, presented as mean and standard deviation (mean $\pm$ std) across three trials with the best value highlighted in bold

**CIFAR-10**

| # of Experts | $D_f$ | $D_r$ | $D_{ft}$ | $D_{rt}$ | HM | $HM_t$ |
|---|---|---|---|---|---|---|
| 1 | 0.00 ± 0.00 | 99.87 ± 0.00 | 0.00 ± 0.00 | 91.16 ± 0.01 | **99.93 ± 0.00** | **95.38 ± 0.01** |
| 2 | 0.00 ± 0.00 | 99.86 ± 0.01 | 0.00 ± 0.00 | 90.99 ± 0.07 | **99.93 ± 0.01** | 95.28 ± 0.04 |
| 3 | 0.00 ± 0.00 | 99.87 ± 0.01 | 0.00 ± 0.00 | 90.87 ± 0.02 | **99.93 ± 0.00** | 95.21 ± 0.01 |
| 4 | 0.00 ± 0.00 | 99.83 ± 0.03 | 0.00 ± 0.00 | 91.00 ± 0.16 | 99.92 ± 0.01 | 95.29 ± 0.09 |
| 5 | 0.00 ± 0.00 | 99.80 ± 0.00 | 0.00 ± 0.00 | 90.91 ± 0.17 | 99.90 ± 0.00 | 95.24 ± 0.09 |
| 6 | 0.00 ± 0.00 | 99.81 ± 0.02 | 0.00 ± 0.00 | 90.80 ± 0.15 | 99.90 ± 0.01 | 95.18 ± 0.04 |
| 7 | 0.00 ± 0.00 | 99.82 ± 0.01 | 0.00 ± 0.00 | 90.93 ± 0.13 | 99.91 ± 0.01 | 95.25 ± 0.07 |
| 8 | 0.00 ± 0.00 | 99.82 ± 0.01 | 0.00 ± 0.00 | 90.84 ± 0.08 | 99.91 ± 0.00 | 95.20 ± 0.04 |
| 9 | 0.00 ± 0.00 | 99.82 ± 0.01 | 0.00 ± 0.00 | 90.91 ± 0.02 | 99.91 ± 0.01 | 95.24 ± 0.05 |

**CIFAR-10 (KR setting; lr = 0.1)**

| # of Experts | $D_f$ | $D_r$ | $D_{ft}$ | $D_{rt}$ | HM | $HM_t$ |
|---|---|---|---|---|---|---|
| 1 | 33.19 ± 57.48 | 96.18 ± 6.40 | 29.77 ± 51.56 | 87.80 ± 5.88 | 66.92 ± 57.19 | 69.90 ± 44.16 |
| 2 | 9.05 ± 10.07 | 97.41 ± 2.44 | 9.47 ± 10.17 | 88.29 ± 2.06 | 93.95 ± 6.61 | 89.27 ± 6.10 |
| 3 | 5.37 ± 3.91 | 97.76 ± 1.64 | 7.87 ± 5.09 | 88.61 ± 1.35 | 96.16 ± 2.82 | **90.31 ± 3.16** |
| 4 | 6.34 ± 5.61 | 97.60 ± 1.94 | 8.33 ± 5.20 | 87.56 ± 2.01 | 95.56 ± 3.87 | 89.55 ± 3.55 |
| 5 | 3.59 ± 1.74 | 98.22 ± 0.85 | 7.40 ± 2.52 | 88.02 ± 0.87 | 97.30 ± 1.29 | 90.25 ± 1.65 |
| 6 | 3.17 ± 2.26 | 98.32 ± 1.07 | 8.13 ± 4.07 | 87.87 ± 1.51 | 97.57 ± 1.67 | 89.81 ± 2.74 |
| 7 | 2.99 ± 1.23 | 98.54 ± 0.64 | 8.13 ± 2.45 | 87.54 ± 0.83 | 97.77 ± 0.94 | 89.65 ± 1.59 |
| 8 | 2.65 ± 0.77 | 98.48 ± 0.65 | 7.80 ± 3.27 | 87.26 ± 1.13 | 97.91 ± 0.71 | 89.65 ± 2.14 |
| 9 | 1.96 ± 0.31 | 98.48 ± 0.53 | 8.17 ± 2.40 | 87.37 ± 1.11 | **98.26 ± 0.41** | 89.54 ± 1.72 |

**CIFAR-100**

| # of Experts | $D_f$ | $D_r$ | $D_{ft}$ | $D_{rt}$ | HM | $HM_t$ |
|---|---|---|---|---|---|---|
| 10 | 0.00 ± 0.00 | 99.98 ± 0.00 | 0.00 ± 0.00 | 80.16 ± 0.05 | **99.99 ± 0.00** | 88.99 ± 0.03 |
| 20 | 0.00 ± 0.00 | 99.98 ± 0.00 | 0.00 ± 0.00 | 80.22 ± 0.03 | **99.99 ± 0.00** | **89.03 ± 0.02** |
| 30 | 0.00 ± 0.00 | 99.98 ± 0.00 | 0.00 ± 0.00 | 80.21 ± 0.01 | **99.99 ± 0.00** | 89.02 ± 0.02 |
| 40 | 0.00 ± 0.00 | 99.98 ± 0.00 | 0.00 ± 0.00 | 80.16 ± 0.08 | **99.99 ± 0.00** | 88.98 ± 0.05 |
| 50 | 0.00 ± 0.00 | 99.98 ± 0.00 | 0.00 ± 0.00 | 80.16 ± 0.08 | **99.99 ± 0.00** | 88.99 ± 0.05 |
| 60 | 0.00 ± 0.00 | 99.98 ± 0.00 | 0.00 ± 0.00 | 80.21 ± 0.03 | **99.99 ± 0.00** | 89.02 ± 0.02 |
| 70 | 0.00 ± 0.00 | 99.98 ± 0.00 | 0.00 ± 0.00 | 80.17 ± 0.09 | **99.99 ± 0.00** | 89.00 ± 0.06 |
| 80 | 0.00 ± 0.00 | 99.98 ± 0.00 | 0.00 ± 0.00 | 80.17 ± 0.02 | **99.99 ± 0.00** | 88.99 ± 0.01 |
| 90 | 0.00 ± 0.00 | 99.98 ± 0.00 | 0.00 ± 0.00 | 80.23 ± 0.02 | **99.99 ± 0.00** | 89.03 ± 0.01 |

**CIFAR-100 (KR setting; lr = 0.1)**

| # of Experts | $D_f$ | $D_r$ | $D_{ft}$ | $D_{rt}$ | HM | $HM_t$ |
|---|---|---|---|---|---|---|
| 10 | 0.24 ± 0.21 | 99.73 ± 0.22 | 1.13 ± 0.42 | 78.39 ± 0.34 | 99.75 ± 0.21 | **87.44 ± 0.37** |
| 20 | 0.07 ± 0.09 | 99.86 ± 0.18 | 1.27 ± 0.72 | 77.01 ± 0.06 | 99.89 ± 0.14 | 86.53 ± 0.31 |
| 30 | 0.00 ± 0.00 | 99.97 ± 0.06 | 1.63 ± 0.15 | 75.64 ± 0.21 | **99.99 ± 0.00** | 85.52 ± 0.10 |
| 40 | 0.02 ± 0.00 | 99.97 ± 0.00 | 1.20 ± 0.35 | 74.19 ± 0.35 | 99.98 ± 0.00 | 84.74 ± 0.34 |
| 50 | 0.00 ± 0.00 | 99.98 ± 0.00 | 2.00 ± 0.66 | 73.43 ± 0.48 | **99.99 ± 0.00** | 83.96 ± 0.56 |
| 60 | 0.03 ± 0.02 | 99.98 ± 0.00 | 1.80 ± 0.26 | 72.63 ± 0.28 | 99.98 ± 0.01 | 83.50 ± 0.12 |
| 70 | 0.01 ± 0.01 | 99.98 ± 0.00 | 1.67 ± 0.61 | 71.43 ± 0.35 | 99.98 ± 0.01 | 82.75 ± 0.33 |
| 80 | 0.03 ± 0.02 | 99.98 ± 0.00 | 1.90 ± 0.87 | 71.00 ± 0.19 | 99.97 ± 0.01 | 82.37 ± 0.30 |
| 90 | 0.01 ± 0.01 | 99.98 ± 0.00 | 2.03 ± 0.42 | 70.37 ± 0.33 | **99.99 ± 0.01** | 81.91 ± 0.28 |

**Tiny-ImageNet**

| # of Experts | $D_f$ | $D_r$ | $D_{ft}$ | $D_{rt}$ | HM | $HM_t$ |
|---|---|---|---|---|---|---|
| 20 | 0.02 ± 0.01 | 96.11 ± 0.02 | 0.00 ± 0.00 | 89.92 ± 0.08 | 98.01 ± 0.01 | 94.69 ± 0.05 |
| 40 | 0.02 ± 0.00 | 96.13 ± 0.01 | 0.00 ± 0.00 | 89.94 ± 0.02 | 98.02 ± 0.01 | 94.70 ± 0.01 |
| 60 | 0.03 ± 0.01 | 96.14 ± 0.00 | 0.00 ± 0.00 | 89.95 ± 0.02 | 98.02 ± 0.00 | 94.71 ± 0.03 |
| 80 | 0.02 ± 0.01 | 96.15 ± 0.01 | 0.00 ± 0.00 | 89.91 ± 0.06 | 98.03 ± 0.01 | 94.69 ± 0.03 |
| 100 | 0.03 ± 0.01 | 96.16 ± 0.01 | 0.00 ± 0.00 | 89.94 ± 0.04 | 98.03 ± 0.01 | 94.70 ± 0.02 |
| 120 | 0.02 ± 0.01 | 96.17 ± 0.02 | 0.00 ± 0.00 | 89.92 ± 0.06 | 98.04 ± 0.01 | 94.69 ± 0.04 |
| 140 | 0.02 ± 0.01 | 96.15 ± 0.02 | 0.00 ± 0.00 | 89.93 ± 0.01 | 98.03 ± 0.01 | 94.70 ± 0.00 |
| 160 | 0.02 ± 0.01 | 96.16 ± 0.02 | 0.00 ± 0.00 | 90.01 ± 0.09 | 98.03 ± 0.01 | **94.74 ± 0.05** |
| 180 | 0.00 ± 0.00 | 96.18 ± 0.01 | 0.00 ± 0.00 | 89.89 ± 0.03 | **98.05 ± 0.00** | 94.68 ± 0.02 |

**Tiny-ImageNet (KR setting; lr = 0.1)**

| # of Experts | $D_f$ | $D_r$ | $D_{ft}$ | $D_{rt}$ | HM | $HM_t$ |
|---|---|---|---|---|---|---|
| 20 | 1.99 ± 0.92 | 96.63 ± 0.10 | 1.63 ± 0.65 | 90.11 ± 0.13 | 97.31 ± 0.50 | 94.06 ± 0.28 |
| 40 | 1.10 ± 0.15 | 96.63 ± 0.02 | 0.90 ± 0.56 | 90.04 ± 0.11 | 97.75 ± 0.07 | 94.35 ± 0.29 |
| 60 | 0.80 ± 0.11 | 96.63 ± 0.01 | 0.73 ± 0.21 | 90.01 ± 0.06 | 97.90 ± 0.05 | 94.41 ± 0.07 |
| 80 | 0.65 ± 0.04 | 96.58 ± 0.03 | 0.70 ± 0.20 | 89.92 ± 0.15 | 97.95 ± 0.01 | 94.38 ± 0.14 |
| 100 | 0.54 ± 0.10 | 96.56 ± 0.02 | 0.63 ± 0.06 | 89.93 ± 0.08 | 97.99 ± 0.06 | 94.41 ± 0.07 |
| 120 | 0.55 ± 0.08 | 96.54 ± 0.02 | 0.50 ± 0.20 | 89.94 ± 0.07 | 97.97 ± 0.04 | 94.48 ± 0.13 |
| 140 | 0.51 ± 0.10 | 96.55 ± 0.01 | 0.50 ± 0.17 | 89.96 ± 0.16 | **98.00 ± 0.05** | 94.49 ± 0.06 |
| 160 | 0.50 ± 0.07 | 96.52 ± 0.01 | 0.43 ± 0.25 | 89.94 ± 0.07 | 97.99 ± 0.04 | **94.51 ± 0.14** |
| 180 | 0.50 ± 0.14 | 96.53 ± 0.03 | 0.53 ± 0.15 | 89.81 ± 0.19 | 97.99 ± 0.06 | 94.39 ± 0.09 |

Table 13: Unlearning performance when the proposed methods are applied to shallower layers. Accuracy, KR, and KD performance are evaluated on CIFAR-10 with AllCNN and CIFAR-100 with ResNet-18. The table reports results with learning rate 0.1 for KR, presented as mean and standard deviation (mean $\pm$ std) across three trials with the best value highlighted in bold

**CIFAR-10**

| | Method | $D_f$ | $D_r$ | $D_{ft}$ | $D_{rt}$ | HM | $HM_t$ |
|---|---|---|---|---|---|---|---|
| 2nd last | Erase | 0.95 ± 0.02 | 99.90 ± 0.00 | 0.50 ± 0.00 | 91.37 ± 0.01 | 99.47 ± 0.01 | **95.26 ± 0.00** |
| 2nd last | Remap | 0.44 ± 0.00 | 99.52 ± 0.00 | 0.13 ± 0.06 | 90.53 ± 0.01 | **99.54 ± 0.00** | 94.97 ± 0.03 |
| 2nd last | MoUE | 0.67 ± 0.02 | 99.60 ± 0.01 | 0.37 ± 0.06 | 90.73 ± 0.16 | 99.47 ± 0.02 | 94.98 ± 0.11 |
| 3rd last | Erase | 46.30 ± 0.15 | 98.07 ± 0.01 | 36.63 ± 0.38 | 88.87 ± 0.02 | 69.40 ± 0.13 | 73.98 ± 0.26 |
| 3rd last | Remap | 24.26 ± 0.16 | 96.22 ± 0.05 | 18.43 ± 0.12 | 86.90 ± 0.04 | 84.76 ± 0.11 | 84.15 ± 0.08 |
| 3rd last | MoUE | 25.28 ± 0.24 | 96.36 ± 0.09 | 19.47 ± 0.45 | 87.31 ± 0.10 | 84.17 ± 0.18 | 83.78 ± 0.25 |

**CIFAR-10 (KR setting; lr = 0.1)**

| | Method | $D_f$ | $D_r$ | $D_{ft}$ | $D_{rt}$ | HM | $HM_t$ |
|---|---|---|---|---|---|---|---|
| 2nd last | Erase | 96.09 ± 0.75 | 99.57 ± 0.08 | 87.17 ± 1.33 | 89.84 ± 0.20 | 7.51 ± 1.38 | 22.44 ± 2.04 |
| 2nd last | Remap | 94.03 ± 2.51 | 99.08 ± 0.35 | 84.83 ± 2.93 | 89.97 ± 0.37 | 11.20 ± 4.52 | 25.88 ± 4.37 |
| 2nd last | MoUE | 86.01 ± 2.50 | 98.26 ± 0.33 | 79.30 ± 2.55 | 88.97 ± 0.42 | **24.44 ± 3.81** | 33.54 ± 3.39 |
| 3rd last | Erase | 98.16 ± 0.06 | 99.17 ± 0.01 | 88.83 ± 0.21 | 89.75 ± 0.07 | 3.61 ± 0.12 | 19.86 ± 0.33 |
| 3rd last | Remap | 92.53 ± 0.65 | 98.46 ± 0.04 | 82.30 ± 0.46 | 88.94 ± 0.10 | 13.89 ± 1.12 | 29.52 ± 0.64 |
| 3rd last | MoUE | 87.15 ± 0.69 | 97.20 ± 0.08 | 75.97 ± 0.90 | 87.43 ± 0.05 | 22.70 ± 1.09 | **37.70 ± 1.11** |

**CIFAR-100**

| | Method | $D_f$ | $D_r$ | $D_{ft}$ | $D_{rt}$ | HM | $HM_t$ |
|---|---|---|---|---|---|---|---|
| 2nd last | Erase | 5.20 ± 0.20 | 99.98 ± 0.00 | 0.60 ± 0.00 | 81.49 ± 0.01 | 97.32 ± 0.11 | **89.56 ± 0.00** |
| 2nd last | Remap | 0.13 ± 0.12 | 99.98 ± 0.00 | 0.00 ± 0.00 | 80.71 ± 0.00 | **99.92 ± 0.06** | 89.33 ± 0.00 |
| 2nd last | MoUE | 0.20 ± 0.20 | 99.98 ± 0.00 | 0.00 ± 0.00 | 80.38 ± 0.03 | 99.89 ± 0.10 | 89.12 ± 0.02 |
| 3rd last | Erase | 12.53 ± 1.33 | 99.78 ± 0.00 | 1.90 ± 0.00 | 79.13 ± 0.00 | 93.22 ± 0.75 | 87.60 ± 0.00 |
| 3rd last | Remap | 8.67 ± 2.01 | 99.04 ± 0.00 | 1.60 ± 0.00 | 77.62 ± 0.01 | 95.02 ± 1.09 | 86.78 ± 0.00 |
| 3rd last | MoUE | 10.00 ± 2.36 | 99.20 ± 0.01 | 2.07 ± 0.21 | 77.91 ± 0.17 | 94.36 ± 1.31 | 86.78 ± 0.18 |

**CIFAR-100 (KR setting; lr = 0.1)**

| | Method | $D_f$ | $D_r$ | $D_{ft}$ | $D_{rt}$ | HM | $HM_t$ |
|---|---|---|---|---|---|---|---|
| 2nd last | Erase | 52.87 ± 2.04 | 99.98 ± 0.00 | 42.40 ± 2.27 | 77.16 ± 0.02 | 64.05 ± 1.90 | 65.94 ± 1.50 |
| 2nd last | Remap | 81.73 ± 1.70 | 99.77 ± 0.26 | 52.70 ± 1.00 | 78.48 ± 0.29 | 30.86 ± 2.43 | 59.02 ± 0.74 |
| 2nd last | MoUE | 29.53 ± 2.50 | 99.83 ± 0.05 | 26.27 ± 2.06 | 75.06 ± 0.04 | **82.60 ± 1.73** | **74.38 ± 1.05** |
| 3rd last | Erase | 74.47 ± 1.81 | 99.77 ± 0.01 | 40.60 ± 0.36 | 77.36 ± 0.07 | 40.64 ± 2.29 | 67.20 ± 0.25 |
| 3rd last | Remap | 83.07 ± 4.75 | 99.68 ± 0.15 | 49.70 ± 3.30 | 76.22 ± 0.25 | 28.76 ± 6.79 | 60.56 ± 2.28 |
| 3rd last | MoUE | 59.73 ± 1.22 | 99.35 ± 0.05 | 34.73 ± 1.04 | 75.03 ± 0.05 | 57.30 ± 1.23 | 69.80 ± 0.60 |

## D.1 ROUTER INITIALIZATION

**Random Initialization.** Random initialization is the most commonly used router initialization method in existing MoE research. Similar to the approach used for router initialization in MoE

Upcycling reserach Komatsuzaki et al. (2023), we scale the standard deviation with $N(0, 0.1^2)$ to initialize values close to zero, allowing sufficient gradient flow during training.

**Pseudo Inverse Initialization.** The role of the router is crucial in MoE, and although randomly initialized routers are used, they are highly sensitive to training. If a specific expert is continuously selected during the initial training phase, the vector representing that expert in the router will receive more training influence compared to others, causing it to continuously dominate training samples. This sensitivity becomes particularly problematic in our MoRE unlearning tasks, where the remain set should be preserved while the forget set needs to be forgotten, making router training inherently unstable. A key consideration is that we want to accurately represent the space where the remain classes are distributed while erasing the space occupied by the forget classes.

Therefore, it is necessary to set initial router weights to represent the remain feature space as uniformly as possible, and we propose an initialization method using the pseudo inverse of remain prototypes. Our method applies SVD to the prototype set $\mathbf{P}_R \in \mathbb{R}^{d \times k_R}$ derived from $k_R$ remain classes, computes the pseudo-inverse $\mathbf{P}_R^+ \in \mathbb{R}^{k_R \times d}$, and initializes the router weight $\mathbf{W}_r \in \mathbb{R}^{N_E \times d}$ using the first $N_E$ rows of $\mathbf{P}_R^+$. Through this approach, we can initialize an optimal router that effectively represents the remain feature space according to the predefined number of experts $N_E$.

---

**Algorithm 4** Router Initialization with Remain Prototypes' Pseudo-Inverse

---

**Input:** Remain class prototypes $\mathbf{P}_R \in \mathbb{R}^{d \times k_R}$, number of experts $N_E$
**Output:** Initialized router weights $\mathbf{W}_r \in \mathbb{R}^{N_E \times d}$
 1: Perform Singular Value Decomposition: $\mathbf{U}, \mathbf{\Sigma}, \mathbf{V}^T \leftarrow \text{SVD}(\mathbf{P}_R)$
 2: Compute the pseudo-inverse: $\mathbf{P}_R^+ \leftarrow \mathbf{V}\mathbf{\Sigma}^{-1}\mathbf{U}^T$
 3: Set router weights to the top $N_E$ rows: $\mathbf{W}_r \leftarrow (\mathbf{P}_R^+)_{:N_E,:}$
 4: **return** $\mathbf{W}_r$

---

### D.2 ROUTER FINETUNING

**Remain and Forget Loss.** For the entire training dataset, we define individual loss functions for forget samples (which we aim to unlearn) and remain samples (which should preserve accuracy), as follows:

$$\mathcal{L}_R = \frac{1}{|\mathcal{B}_R|} \sum_{(\mathbf{x},y) \in \mathcal{B}_R} \mathcal{L}_{CE}(f_\theta(\mathbf{x}), y), \quad \mathcal{L}_F = -\frac{1}{|\mathcal{B}_F|} \sum_{(\mathbf{x},y) \in \mathcal{B}_F} \mathcal{L}_{CE}(f_\theta(\mathbf{x}), y),$$

where $\mathcal{B}_R$ denotes remain samples in the batch, $\mathcal{B}_F$ denotes forget samples in the batch, and $\mathcal{L}_{CE}$ represents the cross-entropy loss. The forget loss function penalizes correct predictions by increasing the loss value, thus preventing effective learning on these samples.

**Forget Load Balancing Loss.** MoE architectures commonly suffer from expert imbalance, where only a few experts handle most of the workload while others remain underutilized Fedus et al. (2022). This happens when certain experts gain an early advantage during training and subsequently attract more training data. Load balancing regularization is therefore essential in MoE training to ensure fair expert utilization. Futhermore, our MoRE unlearning scenario introduces a unique challenge in which remain and forget samples require fundamentally different routing strategies due to their opposing learning objectives. Remain samples can be concentrated on specific experts as long as they are routed toward directions that effectively reduce task loss. On the other hand, forget samples, which are intentionally designed to disrupt learning, create a coordination problem when they cluster on specific experts. Such clustering results in heavy penalization of those experts through forget loss, which subsequently interferes with their capacity to process remain samples effectively and creates a destructive feedback loop. To address this challenge, we apply load balancing loss exclusively to forget samples:

$$\mathcal{L}_{FLB} = N_E \sum_{e=1}^{N_E} (I_e \cdot L_e),$$

$$I_e = \frac{1}{|\mathcal{B}_F|} \sum_{(\mathbf{x},y) \in \mathcal{B}_F} \text{softmax}(\mathbf{W_r x})_e, \quad L_e = \frac{1}{|\mathcal{B}_F|} \sum_{(\mathbf{x},y) \in \mathcal{B}_F} \mathbf{1}[\text{TopK}(\text{softmax}(\mathbf{W_r x}))],$$

where $I_e$ represents the average probability computed through the router for forget samples, indicating their importance, while $L_e$ denotes the load measured by counting the actual number of experts selected in the top-K selection process. This serves to equalize expert utilization by encouraging both the probability of each expert selecting forget samples and the actual assignment probability to approach $\frac{1}{N_E}$ during training, thus preventing expert concentration for forget samples. The total loss function is formulated by integrating all the loss terms as follows:

$$\mathcal{L}_{total} = \mathcal{L}_R + \lambda_F \mathcal{L}_F + \lambda_{FLB} \mathcal{L}_{FLB},$$

where $\lambda_F$ and $\lambda_{FLB}$ denote weighting coefficients.

### D.3  EXPERIMENTS

**Experimental Setup.** We finetune the model for 10 epochs using 50,000 samples across all cases and employ the Adam optimizer with a learning rate of 1e-5 and fix the batch size $|\mathcal{B}| = 512$, following He et al. (2025) that MoE training requires large batch sizes and small learning rates. Ablation results for different combinations of loss functions are presented, where applicable losses are weighted with $\lambda_F = 0.001$ and $\lambda_{FLB} = 1.0$. Coefficient of Variation (CV) represents the variance divided by the mean of data samples routed to each expert. Lower values approaching 0 signify well-balanced load distribution, whereas higher values indicate the dominance of specific experts. Compared to MoRE with random routing, we use the following notation: **-R** for random initialization, **-P** for pseudo-inverse initialization, **-T** for finetuning with the remain loss $\mathcal{L}_R$ only, and **-T-B** for finetuning with both $\mathcal{L}_R$ and the forget load balancing loss $\mathcal{L}_{FLB}$.

**MoRE Constraints.** The finetuning process targets only the router training with top-1 expert selection, which aims to preserve MoRE's remapping expert layers and ensure a fair comparison with random routing. In the KR setting, we first initialize the router, then freeze the remaining components and finetune the router. Subsequently, we initialize and retrain the classification head while keeping all other components including the finetuned router frozen for evaluation.

**Initialization and Finetuning.** Table 14 demonstrates the performance impact of various initialization methods and finetuning on MoRE. Comparing initialization methods, the Non-KR setting (left) shows minimal differences, whereas the KR setting (right) reveals notable variations. In the KR setting, since the classification head is reinitialized and retrained, pseudo inverse initialization using remain prototypes provides a more favorable starting point by effectively capturing the overall remain sample space.

However, the post-initialization finetuning results show performance degradation even in **MoRE-P-T**, which occurs because the pseudo inverse initialization is conducted solely based on remain prototype information. This makes the system inherently vulnerable to forget sample information when trained only with remain loss, potentially leading to increased forget loss. Consequently, we observe that performance degradation is somewhat alleviated when forget sample balancing is incorporated.

While some cases like CIFAR-10 suggest the potential existence of conditional routers, random routing demonstrates the highest performance in most scenarios. Particularly in the KR setting, routers trained on already pretrained classification heads have fixed pathways, which diminishes the significance of utilizing multiple expert layers when retraining the classification head if those pathways are not optimal.

**Loss Ablation Study.** Our key observation is that the **forget load balancing is crucial**.

In this context, Table 15 shows that MoRE with random routing achieves good load balancing with low CV, demonstrating remarkably stable performance especially when the KR learning rate increases, while other methods essentially collapse. Although this approach may not be optimal in all cases when considering accuracy alone—which remains part of our future work—it is still rea-

Table 14: Comparison of accuracy and KR performance across conditional router initialization and finetuning methods. $N$ denotes the number of experts. Accuracy and KR performance in the KD settings are evaluated using CIFAR-10 with AllCNN, CIFAR-100 with ResNet-18, and Tiny-ImageNet with ViT. The table presents the mean and standard deviation (mean $\pm$ std) across three trials with the best value highlighted in bold.

**CIFAR-10 (N=8)**

| Method | D_f | D_r | D_ft | D_rt | HM | HM_t |
|---|---|---|---|---|---|---|
| Original | 99.92 | 99.94 | 91.80 | 91.07 | 0.16 | 15.05 |
| Retrain | 0.00 | 99.15 | 0.00 | 92.04 | 99.57 | 95.86 |
| MoRE | $0.00 \pm 0.00$ | $99.82 \pm 0.02$ | $0.00 \pm 0.00$ | $90.90 \pm 0.08$ | $\mathbf{99.91 \pm 0.01}$ | $95.23 \pm 0.05$ |
| MoRE-R | $0.00 \pm 0.00$ | $99.78 \pm 0.05$ | $0.00 \pm 0.00$ | $90.86 \pm 0.14$ | $99.89 \pm 0.03$ | $95.21 \pm 0.08$ |
| MoRE-R-T | $0.00 \pm 0.00$ | $97.78 \pm 0.15$ | $0.00 \pm 0.00$ | $88.82 \pm 0.25$ | $98.88 \pm 0.03$ | $94.08 \pm 0.08$ |
| MoRE-P | $0.00 \pm 0.00$ | $99.61 \pm 0.00$ | $0.00 \pm 0.00$ | $91.01 \pm 0.01$ | $99.81 \pm 0.01$ | $\mathbf{95.29 \pm 0.01}$ |
| MoRE-P-T | $0.00 \pm 0.00$ | $97.41 \pm 0.16$ | $0.00 \pm 0.00$ | $88.95 \pm 0.18$ | $98.55 \pm 0.03$ | $94.16 \pm 0.05$ |
| MoRE-P-T-B | $0.00 \pm 0.00$ | $97.42 \pm 0.02$ | $0.00 \pm 0.00$ | $88.73 \pm 0.07$ | $98.69 \pm 0.01$ | $94.03 \pm 0.04$ |

**CIFAR-10 (N=8, KR setting; lr = 0.1)**

| Method | D_f | D_r | D_ft | D_rt | HM | HM_t |
|---|---|---|---|---|---|---|
| Original | 99.88 | 99.95 | 91.20 | 91.07 | 0.24 | 16.05 |
| Retrain | 72.62 | 97.06 | 72.90 | 88.01 | 42.71 | 41.44 |
| MoRE | $5.25 \pm 3.77$ | $96.35 \pm 3.09$ | $13.73 \pm 6.64$ | $84.27 \pm 3.48$ | $95.54 \pm 3.42$ | $85.24 \pm 5.04$ |
| MoRE-R | $45.38 \pm 23.21$ | $95.37 \pm 3.29$ | $42.33 \pm 20.54$ | $86.21 \pm 2.96$ | $66.91 \pm 23.73$ | $67.33 \pm 18.91$ |
| MoRE-R-T | $24.59 \pm 6.35$ | $94.96 \pm 1.49$ | $24.20 \pm 6.01$ | $85.33 \pm 1.80$ | $84.00 \pm 4.75$ | $80.24 \pm 4.09$ |
| MoRE-P | $5.89 \pm 5.34$ | $96.94 \pm 2.04$ | $6.33 \pm 5.34$ | $87.31 \pm 2.05$ | $95.47 \pm 4.37$ | $90.35 \pm 4.01$ |
| MoRE-P-T | $12.43 \pm 15.93$ | $95.36 \pm 0.90$ | $11.60 \pm 12.74$ | $86.02 \pm 0.69$ | $90.81 \pm 9.56$ | $86.91 \pm 6.67$ |
| MoRE-P-T-B | $0.97 \pm 0.37$ | $95.48 \pm 0.84$ | $1.70 \pm 0.61$ | $86.08 \pm 0.85$ | $\mathbf{97.22 \pm 0.61}$ | $\mathbf{91.79 \pm 0.74}$ |

**CIFAR-100 (N=10)**

| Method | D_f | D_r | D_ft | D_rt | HM | HM_t |
|---|---|---|---|---|---|---|
| Original | 100.00 | 99.98 | 67.00 | 80.81 | 0.00 | 46.86 |
| Retrain | 0.00 | 96.64 | 0.00 | 72.00 | 98.29 | 83.72 |
| MoRE | $0.00 \pm 0.00$ | $99.98 \pm 0.00$ | $0.00 \pm 0.00$ | $80.16 \pm 0.05$ | $\mathbf{99.99 \pm 0.00}$ | $88.99 \pm 0.03$ |
| MoRE-R | $0.00 \pm 0.00$ | $99.98 \pm 0.00$ | $0.00 \pm 0.00$ | $80.24 \pm 0.07$ | $\mathbf{99.99 \pm 0.00}$ | $\mathbf{89.04 \pm 0.04}$ |
| MoRE-R-T | $0.00 \pm 0.00$ | $99.78 \pm 0.04$ | $0.00 \pm 0.00$ | $78.94 \pm 0.16$ | $99.88 \pm 0.01$ | $88.25 \pm 0.03$ |
| MoRE-P | $0.00 \pm 0.00$ | $99.98 \pm 0.00$ | $0.00 \pm 0.00$ | $79.89 \pm 0.00$ | $\mathbf{99.99 \pm 0.00}$ | $88.82 \pm 0.00$ |
| MoRE-P-T | $0.00 \pm 0.00$ | $99.78 \pm 0.04$ | $0.00 \pm 0.00$ | $78.95 \pm 0.06$ | $99.89 \pm 0.02$ | $88.24 \pm 0.04$ |
| MoRE-P-T-B | $0.00 \pm 0.00$ | $99.74 \pm 0.02$ | $0.00 \pm 0.00$ | $78.57 \pm 0.04$ | $99.87 \pm 0.01$ | $88.00 \pm 0.03$ |

**CIFAR-100 (N=10, KR setting; lr = 0.1)**

| Method | D_f | D_r | D_ft | D_rt | HM | HM_t |
|---|---|---|---|---|---|---|
| Original | 100.00 | 99.98 | 67.00 | 80.41 | 0.00 | 46.80 |
| Retrain | 57.20 | 97.50 | 58.00 | 71.60 | 59.43 | 52.96 |
| MoRE | $0.24 \pm 0.21$ | $99.73 \pm 0.22$ | $1.13 \pm 0.42$ | $78.39 \pm 0.34$ | $\mathbf{99.75 \pm 0.21}$ | $\mathbf{87.44 \pm 0.37}$ |
| MoRE-R | $25.53 \pm 8.96$ | $96.58 \pm 1.22$ | $21.87 \pm 5.44$ | $76.98 \pm 0.99$ | $83.84 \pm 6.08$ | $77.48 \pm 3.78$ |
| MoRE-R-T | $57.80 \pm 3.83$ | $98.49 \pm 0.15$ | $41.53 \pm 1.79$ | $77.03 \pm 0.09$ | $59.02 \pm 3.72$ | $66.45 \pm 1.15$ |
| MoRE-P | $17.40 \pm 4.79$ | $97.31 \pm 1.10$ | $15.83 \pm 3.52$ | $76.78 \pm 0.99$ | $90.89 \pm 2.73$ | $80.19 \pm 1.25$ |
| MoRE-P-T | $48.20 \pm 3.82$ | $99.23 \pm 0.37$ | $37.33 \pm 1.97$ | $77.57 \pm 0.25$ | $68.02 \pm 3.40$ | $69.32 \pm 1.29$ |
| MoRE-P-T-B | $32.27 \pm 3.11$ | $99.29 \pm 0.18$ | $26.60 \pm 1.47$ | $77.34 \pm 0.14$ | $80.51 \pm 2.25$ | $75.32 \pm 0.81$ |

**Tiny-ImageNet (N=20)**

| Method | D_f | D_r | D_ft | D_rt | HM | HM_t |
|---|---|---|---|---|---|---|
| Original | 98.20 | 96.45 | 96.00 | 90.29 | 3.53 | 7.66 |
| Retrain | 0.00 | 99.98 | 0.00 | 84.05 | 99.99 | 91.33 |
| MoRE | $0.02 \pm 0.01$ | $96.16 \pm 0.02$ | $0.00 \pm 0.00$ | $90.01 \pm 0.09$ | $\mathbf{98.03 \pm 0.01}$ | $\mathbf{94.74 \pm 0.05}$ |
| MoRE-R | $0.00 \pm 0.00$ | $95.19 \pm 0.05$ | $0.00 \pm 0.00$ | $89.12 \pm 0.07$ | $97.53 \pm 0.03$ | $94.25 \pm 0.04$ |
| MoRE-R-T | $0.07 \pm 0.12$ | $96.13 \pm 0.04$ | $0.00 \pm 0.00$ | $89.95 \pm 0.03$ | $97.99 \pm 0.00$ | $94.71 \pm 0.02$ |
| MoRE-P | $0.07 \pm 0.09$ | $95.90 \pm 0.01$ | $0.00 \pm 0.00$ | $89.63 \pm 0.03$ | $97.87 \pm 0.04$ | $94.53 \pm 0.02$ |
| MoRE-P-T | $0.07 \pm 0.12$ | $95.94 \pm 0.01$ | $0.00 \pm 0.00$ | $89.61 \pm 0.03$ | $97.89 \pm 0.06$ | $94.52 \pm 0.02$ |
| MoRE-P-T-B | $0.07 \pm 0.12$ | $95.88 \pm 0.03$ | $0.00 \pm 0.00$ | $89.63 \pm 0.04$ | $97.86 \pm 0.04$ | $94.53 \pm 0.02$ |

**Tiny-ImageNet (N=20, KR setting; lr = 0.1)**

| Method | D_f | D_r | D_ft | D_rt | HM | HM_t |
|---|---|---|---|---|---|---|
| Original | 98.20 | 96.83 | 96.00 | 90.10 | 3.53 | 7.66 |
| Retrain | 81.00 | 99.97 | 84.00 | 83.28 | 31.93 | 26.84 |
| MoRE | $0.50 \pm 0.07$ | $96.52 \pm 0.01$ | $0.43 \pm 0.25$ | $89.94 \pm 0.07$ | $\mathbf{97.99 \pm 0.04}$ | $\mathbf{94.51 \pm 0.14}$ |
| MoRE-R | $30.60 \pm 8.67$ | $95.60 \pm 0.80$ | $28.77 \pm 10.19$ | $89.01 \pm 0.72$ | $80.24 \pm 5.92$ | $78.90 \pm 6.52$ |
| MoRE-R-T | $24.20 \pm 1.59$ | $96.04 \pm 0.44$ | $21.27 \pm 3.30$ | $89.49 \pm 0.51$ | $84.72 \pm 1.16$ | $83.75 \pm 2.04$ |
| MoRE-P | $29.20 \pm 3.36$ | $95.32 \pm 0.32$ | $26.57 \pm 4.13$ | $88.79 \pm 0.30$ | $81.23 \pm 2.35$ | $80.35 \pm 2.58$ |
| MoRE-P-T | $20.80 \pm 3.93$ | $96.19 \pm 0.02$ | $17.87 \pm 3.31$ | $89.53 \pm 0.13$ | $86.84 \pm 2.37$ | $85.65 \pm 1.76$ |
| MoRE-P-T-B | $8.12 \pm 2.08$ | $96.56 \pm 0.06$ | $6.87 \pm 0.40$ | $89.99 \pm 0.09$ | $94.15 \pm 0.09$ | $91.53 \pm 0.23$ |

sonable given that it requires no additional initialization or training overhead, eliminates runtime routing computation, and achieves the best results among our tested approaches in most cases.

# E ROLE OF LLMs

We used LLMs to polish the writing quality and for grammar checks.

Table 15: Ablation table for loss functions across all MoRE settings. CV values are colored blue for lowest values and red for higher values. Cases with accuracy degradation are highlighted with background colors: red for severe and yellow for moderate degradation. Accuracy and KR performance in the KD settings are evaluated using CIFAR-100 with ResNet-18. The table presents the mean and standard deviation (mean ± std) across three trials with the best value highlighted in bold.

**CIFAR-100 (N=10)**

| Method | $\mathcal{L}_r$ | $\mathcal{L}_f$ | $\mathcal{L}_{f\_lb}$ | CV_t | Non-KR HM | Non-KR HM_t | KR; lr=0.001 HM | KR; lr=0.001 HM_t | KR; lr=0.01 HM | KR; lr=0.01 HM_t | KR; lr=0.1 HM | KR; lr=0.1 HM_t |
|---|---|---|---|---|---|---|---|---|---|---|---|---|
| **MoRE** | - | - | - | 0.03 ± 0.01 | **99.99 ± 0.00** | 88.99 ± 0.03 | **99.99 ± 0.00** | **89.00 ± 0.02** | **99.97 ± 0.00** | **87.47 ± 0.12** | **99.75 ± 0.21** | **87.44 ± 0.37** |
| **MoRE-R** | - | - | - | 1.22 ± 0.67 | **99.99 ± 0.00** | **89.04 ± 0.05** | 93.51 ± 1.54 | 84.57 ± 1.09 | 86.41 ± 7.61 | 78.00 ± 3.59 | 83.84 ± 7.58 | 77.48 ± 3.78 |
| **MoRE-R-T** | O | X | X | 0.67 ± 0.19 | 99.88 ± 0.01 | 88.25 ± 0.03 | 94.45 ± 1.58 | 83.72 ± 1.73 | 83.48 ± 2.01 | 76.43 ± 2.57 | 59.02 ± 3.72 | 66.45 ± 1.15 |
| | X | O | X | 1.99 ± 0.82 | 99.88 ± 0.01 | 88.27 ± 0.02 | 84.98 ± 6.35 | 78.89 ± 3.25 | 65.98 ± 8.61 | 69.27 ± 4.15 | 49.44 ± 12.23 | 62.78 ± 4.43 |
| | X | X | O | 0.79 ± 0.18 | 99.88 ± 0.01 | 88.19 ± 0.04 | 97.65 ± 1.13 | 85.93 ± 1.36 | 89.78 ± 2.75 | 79.76 ± 1.55 | 67.16 ± 2.77 | 69.12 ± 1.12 |
| | O | O | X | 1.47 ± 0.67 | 99.88 ± 0.01 | 88.29 ± 0.04 | 82.65 ± 1.48 | 77.32 ± 0.60 | 67.78 ± 3.53 | 69.48 ± 1.78 | 48.45 ± 5.38 | 61.40 ± 1.70 |
| | X | O | O | 0.79 ± 0.18 | 99.88 ± 0.01 | 88.19 ± 0.03 | 97.54 ± 1.31 | 85.86 ± 1.42 | 89.78 ± 2.75 | 79.66 ± 1.52 | 66.99 ± 2.61 | 69.04 ± 1.05 |
| | O | X | O | 0.79 ± 0.18 | 99.88 ± 0.01 | 88.19 ± 0.03 | 97.65 ± 1.13 | 85.93 ± 1.36 | 89.82 ± 2.68 | 79.75 ± 1.55 | 67.22 ± 2.69 | 69.10 ± 1.11 |
| | O | O | O | 0.79 ± 0.17 | 99.88 ± 0.01 | 88.19 ± 0.03 | 97.54 ± 1.31 | 85.86 ± 1.42 | 89.78 ± 2.64 | 79.64 ± 1.55 | 66.93 ± 2.51 | 69.06 ± 1.08 |
| **MoRE-P** | - | - | - | 0.19 ± 0.00 | **99.99 ± 0.00** | **88.82 ± 0.00** | 97.41 ± 0.24 | 86.41 ± 0.19 | 93.97 ± 2.99 | 81.92 ± 2.36 | 89.30 ± 3.98 | 80.29 ± 2.59 |
| **MoRE-P-T** | O | X | X | 1.00 ± 0.13 | 99.87 ± 0.01 | 88.05 ± 0.03 | 97.45 ± 1.41 | 85.75 ± 0.76 | 87.04 ± 6.37 | 78.79 ± 3.01 | 68.02 ± 3.40 | 69.32 ± 1.29 |
| | X | O | X | 2.89 ± 0.18 | 99.88 ± 0.01 | 88.28 ± 0.04 | 88.03 ± 9.72 | 82.56 ± 4.00 | 74.14 ± 9.71 | 73.75 ± 4.42 | 61.15 ± 8.38 | 66.47 ± 2.77 |
| | X | X | O | 0.68 ± 0.03 | 99.87 ± 0.01 | 88.00 ± 0.03 | 99.82 ± 0.06 | 88.05 ± 0.17 | 98.41 ± 0.27 | 85.00 ± 0.23 | 80.47 ± 1.98 | 75.31 ± 0.79 |
| | O | O | X | 2.21 ± 0.02 | 99.88 ± 0.01 | 88.28 ± 0.01 | 76.26 ± 0.85 | 74.64 ± 1.96 | 59.08 ± 4.62 | 64.83 ± 2.56 | 42.43 ± 7.84 | 56.85 ± 3.34 |
| | X | O | O | 0.68 ± 0.03 | 99.87 ± 0.01 | 87.99 ± 0.03 | 99.82 ± 0.06 | 88.05 ± 0.18 | 98.30 ± 0.37 | 84.95 ± 0.26 | 80.47 ± 1.79 | 75.34 ± 0.84 |
| | O | X | O | 0.68 ± 0.03 | 99.87 ± 0.01 | 88.00 ± 0.03 | 99.82 ± 0.06 | 88.05 ± 0.17 | 98.41 ± 0.27 | 84.94 ± 0.27 | 80.51 ± 2.25 | 75.32 ± 0.81 |
| | O | O | O | 0.68 ± 0.02 | 99.87 ± 0.01 | 88.00 ± 0.03 | 99.82 ± 0.06 | 88.05 ± 0.17 | 98.30 ± 0.37 | 84.97 ± 0.28 | 80.51 ± 1.97 | 75.29 ± 0.89 |

**CIFAR-100 (N=40)**

| Method | $\mathcal{L}_r$ | $\mathcal{L}_f$ | $\mathcal{L}_{f\_lb}$ | CV_t | Non-KR HM | Non-KR HM_t | KR; lr=0.001 HM | KR; lr=0.001 HM_t | KR; lr=0.01 HM | KR; lr=0.01 HM_t | KR; lr=0.1 HM | KR; lr=0.1 HM_t |
|---|---|---|---|---|---|---|---|---|---|---|---|---|
| **MoRE** | - | - | - | 0.07 ± 0.01 | **99.99 ± 0.00** | **88.98 ± 0.05** | **99.99 ± 0.00** | **88.88 ± 0.04** | **99.99 ± 0.00** | **85.60 ± 0.13** | **99.98 ± 0.00** | **84.74 ± 0.34** |
| **MoRE-R** | - | - | - | 1.49 ± 0.02 | **99.99 ± 0.00** | **88.93 ± 0.03** | 96.80 ± 1.16 | 86.29 ± 0.95 | 90.96 ± 2.95 | 80.27 ± 1.97 | 89.79 ± 1.54 | 79.04 ± 1.45 |
| **MoRE-R-T** | O | X | X | 1.02 ± 0.08 | 99.88 ± 0.01 | 88.23 ± 0.02 | 98.41 ± 0.77 | 86.86 ± 0.56 | 88.46 ± 3.18 | 79.21 ± 1.38 | 66.89 ± 5.42 | 69.63 ± 0.66 |
| | X | O | X | 3.67 ± 1.23 | 99.88 ± 0.00 | 88.27 ± 0.04 | 85.08 ± 6.21 | 78.90 ± 3.24 | 66.04 ± 8.54 | 69.29 ± 4.13 | 50.91 ± 15.90 | 64.29 ± 5.70 |
| | X | X | O | 0.87 ± 0.12 | 99.88 ± 0.01 | 88.20 ± 0.01 | 97.65 ± 1.13 | 85.93 ± 1.36 | 89.82 ± 2.67 | 79.76 ± 1.56 | 83.38 ± 2.08 | 75.74 ± 1.57 |
| | O | O | X | 2.31 ± 0.10 | 99.88 ± 0.00 | 88.23 ± 0.04 | 82.65 ± 1.48 | 77.31 ± 0.57 | 67.72 ± 3.59 | 69.46 ± 1.83 | 47.15 ± 8.87 | 60.61 ± 1.16 |
| | X | O | O | 0.87 ± 0.12 | 99.88 ± 0.01 | 88.20 ± 0.01 | 97.54 ± 1.31 | 85.86 ± 1.42 | 89.70 ± 2.67 | 79.63 ± 1.54 | 83.34 ± 2.06 | 75.73 ± 1.63 |
| | O | X | O | 0.87 ± 0.12 | 99.88 ± 0.01 | 88.20 ± 0.01 | 97.65 ± 1.13 | 85.92 ± 1.39 | 89.82 ± 2.67 | 79.75 ± 1.55 | 83.43 ± 2.00 | 75.70 ± 1.61 |
| | O | O | O | 0.87 ± 0.12 | 99.88 ± 0.01 | 88.20 ± 0.01 | 97.54 ± 1.31 | 85.86 ± 1.42 | 89.74 ± 2.60 | 79.65 ± 1.55 | 83.34 ± 2.06 | 75.73 ± 1.63 |
| **MoRE-P** | - | - | - | 0.24 ± 0.00 | **99.99 ± 0.00** | 88.70 ± 0.00 | 99.38 ± 0.45 | 87.64 ± 0.12 | 95.89 ± 1.63 | 82.33 ± 0.74 | 90.89 ± 2.73 | 80.19 ± 1.25 |
| **MoRE-P-T** | O | X | X | 1.27 ± 0.09 | 99.88 ± 0.01 | 88.04 ± 0.08 | 99.04 ± 0.38 | 86.95 ± 0.24 | 90.24 ± 3.11 | 79.77 ± 1.94 | 74.05 ± 3.91 | 72.12 ± 1.27 |
| | X | O | X | 6.25 ± 0.00 | 99.88 ± 0.01 | 88.19 ± 0.04 | 92.28 ± 0.51 | 84.58 ± 0.33 | 79.06 ± 2.58 | 76.87 ± 0.24 | 69.66 ± 5.95 | 70.39 ± 1.49 |
| | X | X | O | 0.80 ± 0.03 | 99.84 ± 0.01 | 87.82 ± 0.03 | 99.83 ± 0.00 | 87.84 ± 0.10 | 99.34 ± 0.16 | 84.33 ± 0.30 | 95.03 ± 0.93 | 81.75 ± 0.90 |
| | O | O | X | 5.95 ± 0.30 | 99.88 ± 0.01 | 88.27 ± 0.03 | 92.63 ± 1.91 | 84.14 ± 1.59 | 76.12 ± 4.82 | 75.45 ± 2.21 | 61.19 ± 3.51 | 67.49 ± 1.50 |
| | X | O | O | 0.79 ± 0.03 | 99.84 ± 0.01 | 87.83 ± 0.04 | 99.83 ± 0.01 | 87.84 ± 0.12 | 99.37 ± 0.19 | 84.33 ± 0.19 | 94.99 ± 0.87 | 81.77 ± 0.86 |
| | O | X | O | 0.79 ± 0.03 | 99.84 ± 0.01 | 87.84 ± 0.04 | 99.83 ± 0.00 | 87.85 ± 0.09 | 99.34 ± 0.16 | 84.34 ± 0.26 | 95.03 ± 0.96 | 81.85 ± 0.89 |
| | O | O | O | 0.80 ± 0.03 | 99.84 ± 0.01 | 87.84 ± 0.05 | 99.83 ± 0.01 | 87.84 ± 0.12 | 99.37 ± 0.19 | 84.37 ± 0.25 | 95.07 ± 0.81 | 81.69 ± 0.92 |

**CIFAR-100 (N=80)**

| Method | $\mathcal{L}_r$ | $\mathcal{L}_f$ | $\mathcal{L}_{f\_lb}$ | CV_t | Non-KR HM | Non-KR HM_t | KR; lr=0.001 HM | KR; lr=0.001 HM_t | KR; lr=0.01 HM | KR; lr=0.01 HM_t | KR; lr=0.1 HM | KR; lr=0.1 HM_t |
|---|---|---|---|---|---|---|---|---|---|---|---|---|
| **MoRE** | - | - | - | 0.09 ± 0.01 | 99.88 ± 0.00 | 88.99 ± 0.01 | **99.99 ± 0.00** | **88.84 ± 0.03** | **99.99 ± 0.00** | **84.81 ± 0.15** | **99.97 ± 0.01** | 82.37 ± 0.30 |
| **MoRE-R** | - | - | - | 1.95 ± 0.01 | 99.99 ± 0.00 | 89.00 ± 0.00 | 94.82 ± 1.61 | 85.04 ± 0.54 | 88.63 ± 0.79 | 78.59 ± 1.34 | 80.45 ± 3.70 | 74.76 ± 2.96 |
| **MoRE-R-T** | O | X | X | 1.34 ± 0.16 | 99.88 ± 0.00 | 88.20 ± 0.00 | 97.86 ± 1.92 | 86.40 ± 0.40 | 86.51 ± 5.03 | 77.89 ± 1.80 | 65.44 ± 8.29 | 67.34 ± 3.03 |
| | X | O | X | 4.46 ± 0.21 | 99.88 ± 0.00 | 88.21 ± 0.00 | 76.73 ± 3.71 | 75.72 ± 3.53 | 62.14 ± 5.94 | 67.33 ± 3.94 | 40.57 ± 3.45 | 57.84 ± 1.52 |
| | X | X | O | 1.09 ± 0.10 | 99.88 ± 0.00 | 88.19 ± 0.00 | 99.87 ± 0.00 | 88.05 ± 0.04 | 98.48 ± 0.58 | 82.20 ± 0.13 | 87.98 ± 2.53 | 76.20 ± 1.31 |
| | O | O | X | 3.12 ± 0.21 | 99.88 ± 0.00 | 88.25 ± 0.00 | 73.03 ± 3.14 | 73.30 ± 3.03 | 53.67 ± 2.75 | 62.54 ± 1.75 | 35.65 ± 5.65 | 55.42 ± 1.67 |
| | X | O | O | 1.09 ± 0.10 | 99.88 ± 0.01 | 88.19 ± 0.05 | 99.87 ± 0.00 | 88.05 ± 0.04 | 98.48 ± 0.58 | 82.18 ± 0.12 | 87.81 ± 2.60 | 76.17 ± 1.25 |
| | O | X | O | 1.09 ± 0.09 | 99.88 ± 0.01 | 88.19 ± 0.05 | 99.87 ± 0.00 | 88.05 ± 0.04 | 98.48 ± 0.58 | 82.19 ± 0.11 | 87.77 ± 2.54 | 76.21 ± 1.34 |
| | O | O | O | 1.09 ± 0.10 | 99.88 ± 0.01 | 88.19 ± 0.06 | 99.87 ± 0.00 | 88.05 ± 0.04 | 98.48 ± 0.58 | 82.19 ± 0.12 | 87.85 ± 2.67 | 76.17 ± 1.20 |
| **MoRE-P** | - | - | - | 0.18 ± 0.05 | **99.99 ± 0.00** | **89.26 ± 0.00** | 99.93 ± 0.00 | 88.78 ± 0.15 | 97.04 ± 1.25 | 82.73 ± 0.58 | 94.14 ± 0.88 | 81.25 ± 0.66 |
| **MoRE-P-T** | O | X | X | 0.79 ± 0.07 | 99.88 ± 0.01 | 88.32 ± 0.05 | 95.56 ± 0.26 | 87.76 ± 0.05 | 92.99 ± 1.62 | 81.30 ± 0.62 | 75.13 ± 4.56 | 73.21 ± 0.85 |
| | X | O | X | 8.89 ± 0.00 | 99.88 ± 0.01 | 88.28 ± 0.03 | 93.02 ± 1.00 | 84.37 ± 0.73 | 79.44 ± 2.48 | 76.79 ± 0.74 | 68.54 ± 5.06 | 69.30 ± 1.29 |
| | X | X | O | 0.24 ± 0.01 | 99.85 ± 0.02 | 88.39 ± 0.06 | 99.83 ± 0.01 | 88.27 ± 0.07 | 99.24 ± 0.13 | 83.92 ± 0.21 | 97.31 ± 0.75 | 82.50 ± 0.78 |
| | O | O | X | 3.76 ± 0.94 | 99.88 ± 0.01 | 88.39 ± 0.03 | 75.31 ± 3.46 | 73.10 ± 1.99 | 57.42 ± 3.63 | 63.16 ± 1.68 | 43.87 ± 3.82 | 57.19 ± 1.53 |
| | X | O | O | 0.24 ± 0.01 | 99.85 ± 0.02 | 88.40 ± 0.07 | 99.83 ± 0.01 | 88.28 ± 0.08 | 99.24 ± 0.13 | 83.93 ± 0.20 | 97.38 ± 0.72 | **82.53 ± 0.75** |
| | O | X | O | 0.24 ± 0.01 | 99.85 ± 0.02 | 88.40 ± 0.06 | 99.83 ± 0.01 | 88.26 ± 0.08 | 99.28 ± 0.18 | 83.94 ± 0.24 | 97.41 ± 0.76 | 82.44 ± 0.72 |
| | O | O | O | 0.24 ± 0.01 | 99.85 ± 0.02 | 88.40 ± 0.07 | 99.83 ± 0.01 | 88.27 ± 0.09 | 99.25 ± 0.13 | 83.93 ± 0.20 | 97.25 ± 0.51 | 82.52 ± 0.81 |

