# OpenReview forum: "MoRE: Mixture of Remapping Experts For Irreversible Feature-Level Unlearning"
_ICLR.cc/2026/Conference — Submitted to ICLR 2026_

### Official Review · Reviewer_m2ED · 2025-10-17

**Soundness:** 4
**Presentation:** 3
**Contribution:** 3
**Rating:** 4
**Confidence:** 4

**Summary:**

This paper presents MoRE (Mixture of Remapping Experts), a novel framework for irreversible feature-level unlearning. This work mitigates existing limitations in ESC, namely, (i) the remaining set accuracy degradation, (ii) the cohesion of forget features after unlearning, and (iii) the memory inefficiency. To overcome such shortcomings, the authors propose to (i) project class prototypes into an orthogonal space, where the projected prototypes form an orthogonal basis. This allows MoRE to erase forget prototypes, without affecting the remaining ones, and to remap them to other prototypes (remaining). This disrupts separability between a specific remaining class and the forget class. By extending remapping via multiple experts, MoRE can disperse forget features to different remaining prototypes, effectively breaking the cohesiveness of such features. Experiments on established datasets and architectures demonstrate the superiority of this approach against existing baselines.

**Strengths:**

1. Overall, the paper is clear, well-written, and the figures help in understanding various details of the method.
2. The proposed pipeline is shown to be cheap to run both from a theoretical and empirical point of view.
3. MoRE greatly reduces the recovery of forget features after unlearning.
4. The idea is novel, interesting, and sound.
5. The pseudocode is very clear and helps in understanding the whole approach.

**Weaknesses:**

**Major Weaknesses**
1. The core contribution of this paper is rendering forget set features unusable for classification, and to do so, MoRE adds intermediate projections between the feature extraction and the linear classifier. This assumes black-box access to the model since, if the model is distributed, an attacker could easily recover the forgotten information by removing the added projections. To me is a great limitation and should be properly discussed in the paper. Furthermore, it is not yet clear whether unlearning only part of the model is sufficient to comply with current regulations.
2. MoRE unlearns by remapping unlearning categories to random remaining classes. It is unclear to me whether this can be applied to random unlearning. If not, the proposed method is only applicable to a subset of machine unlearning problems, which is also less relevant in my opinion. Furthermore, I fail to see how this method can be applied to, let's say, LLMs, where a class cannot be directly attributed to a logit in the output distribution.
3. Some steps in the method's formulation are unclear or not properly addressed.
   1. Lines 222-223 assume that given the prototypes matrix $P\in\mathbb{R}^{d\times k}$ and its pseudoinverse $D =V\Sigma^{-1}U^\top$, $DP=I_k$. Although this is true only if $P$ is of full rank, which is likely since $d\gg k$. Yet, I feel this should be properly discussed.
   2. Equation 4, uses $P^+$ instead of $D$ in one case. It is not wrong, but it seems like $D$ and $P^+$ are two different terms.
   3. Also in equation 4, if $DP=I$ (left inverse), shouldn't $I - PP^+=0$ (right inverse)? In that case, the complement-space projection term has no utility.

**Minor Weaknesses**
1. Figure 2 is low resolution. I suggest exporting it to PDF.
2. Line 116-117, Thudi et al. is cited twice (use \citet{})
3. Lines 214-215, the paper should clearly mention that the expected behaviours are specifically intended for class-unlearning.
4. The part on multiple experts (Section 3.3) could be more formal, in my opinion. To fully understand what is happening, I had to scroll until the appendix and check the Pseudocode.
5. Related to W2, Line 320 mentions experiments on instance-wise unlearning. Yet, I was not able to find them.

**Questions:**

1. Can MoRE be applied to random unlearning? If yes, how?
2. Could the authors elaborate on Weakness 3.3 (W3.3)?

**Motivation for my score** \
I feel the limitation of class unlearning is particularly concerning for the broad applicability of this method. Furthermore, the model is not actually unlearned, strictly speaking; rather, the extracted features (rich in information to be forgotten) get remapped in such a way that prevents the reuse of forgotten knowledge, forcing this method on black-box models only. Therefore, I decided to score this paper as marginally below the acceptance threshold. I am willing to change my evaluation based on the authors' rebuttal.

---

> ### Author Response · Authors · 2025-11-25
> **Rebuttal by Authors (1/3)**
>
> # Opening remarks
>
> We are deeply grateful to the reviewer for the time, thoughtful analysis, and constructive suggestions. Your feedback has been immensely helpful in sharpening the clarity of our presentation and highlighting areas for further elaboration. We have taken your comments seriously and made concrete efforts to address each point raised as faithfully as possible.
>
> Below, we provide detailed, point-by-point responses to the identified weaknesses (W#) and questions (Q#). For multi-part bullets, we divide them into subpoints (e.g., W1a, W1b) to ensure clarity and precision in our replies.
>
> We have also included several **additional figures/tables in our response using anonymous external links**. Please note that depending on your browser or PDF viewer, some **fonts or formatting may appear distorted**. If this occurs, we kindly ask you to either **download the PDF directly or click “View Raw”** to ensure proper rendering.
>
> # m2ED_W1a On black-box access assumption
>
> We thank the reviewer for this thoughtful question and the opportunity to clarify. First, we emphasize that the closed-box assumption is not intrinsic to our method. While MoRE is naturally suited to closed systems, **it is also fully compatible with open settings where model weights or internal components are exposed.**
>
> In our illustrations, we present the projection layer as an explicit and independent module simply for clarity and brevity. However, in practice, **this projection layer can be mathematically absorbed or *fused* into the preceding weight matrix, permanently etching the unlearning effect directly into the model parameters** (this is possible because the projection is a purely linear transformation, and there are no non-linearities between it and the preceding layer). This fusion eliminates the need for a standalone projection layer, making it impossible for an attacker to simply remove or bypass it to undo the unlearning. As a result, our method is inherently robust to such tampering and remains effective in open-box scenarios.
>
> # m2ED_W1b On unlearning only part of the model
>
> We thank the reviewer for raising this important point. Indeed, **unlearning only part of the model (typically the classifier head) is a common limitation observed across both training-based and training-free unlearning methods** [1-4]. In most existing approaches, the unlearning effect is concentrated near the output layer, with shallower feature layers left largely unaffected (see [Figure R4_1](https://anonymous.4open.science/r/MoRE-ICLR2026-Rebuttal-21A4/m2ED/Figure%20R4_1.pdf)).
>
> Our method directly addresses this limitation. **MoRE can be flexibly applied to intermediate or early layers in the feature extractor, not just the classification head.** As shown in Table 6 (Lines 460–475), applying MoRE to earlier layers still results in strong unlearning performance on the forget set. While this comes with a trade-off in retain accuracy, it demonstrates our method’s ability to **support stronger unlearning when needed for higher safety or regulatory assurance**.
>
> # m2ED_W2a + Q1 Applicability to random unlearning
>
> We thank the reviewer for raising this important concern. **We clarify that our method can indeed be applied to random unlearning tasks, as demonstrated in Table 3 (Lines 378–390) and discussed in Section 4.3 (Lines 458–471) of the original manuscript.**
>
> To support this setting, we introduce a minor yet effective adaptation: for each class, we compute two distinct prototypes, one for forget samples and one for retain samples. Because our forget instances are limited in number, we observe non-trivial divergence between these retain mean and forget mean of each class. Leveraging this, we perform prototype editing that nudges forget-group activations toward the retain mean, thereby erasing the influence of the forget data.
>
> While this approach does not replicate the retrain-from-scratch-level forget accuracy, it makes improvements in other metrics, namely MIA (membership inference attack) score. Our method ranks second-best in average gap score, demonstrating strong privacy-utility tradeoff under random data unlearning, a setting not originally targeted by our formulation.

---

> ### Author Response · Authors · 2025-11-25
> **Rebuttal by Authors (2/3)**
>
> # m2ED_W2b Applicability to generative model and class-less settings
>
> While MoRE is initially framed around class-wise unlearning, its prototype-based formulation naturally extends to concept-level unlearning, which is widely relevant in the generative domain.
>
> To demonstrate broader applicability to large-scale models, **we implement our method on the text-to-image generation model, namely Stable Diffusion (SD) v1.4**. Following prior works [9,10], we apply prototype orthogonalization, erasure, and remapping to the cross-attention layers, using tokenized input prompts to construct prototypes. Our experimental setup also strictly follows the standard practice established by the SOTA diffusion unlearning methods [6,8–10], ensuring a fair comparison.
>
> **We evaluate unlearning performance in the artistic style erasure task**, which has emerged as a standard benchmark for testing concept-level unlearning in generative models. Following prior works [8, 9], we construct an evaluation set using 20 prompts for each of 10 artists: 5 classical (Van Gogh, Pablo Picasso, Rembrandt, Andy Warhol, Caravaggio) and 5 modern (Kelly McKernan, Thomas Kinkade, Tyler Edlin, Kilian Eng, and the anime series Ajin: DemiHuman), all of whom are reported to be mimicked by SD. We apply MoRE to remove two artistic styles: Van Gogh and Kelly McKernan. And the unlearning performance is measured using the LPIPS scores [11], which compares the generated outputs before and after unlearning to measure visual similarity (lower means more similar). We report three metrics:
> - **LPIPS_f (forget artists)**: LPIPS score computed on the forget artists (higher is better)
> - **LPIPS_r (remain artists)**: LPIPS score on the remain artists (lower is better, indicating minimal disruption)
> - **LPIPS_d = LPIPS_f − LPIPS_r**: the overall tradeoff, capturing how well the method removes target styles while preserving unrelated ones
>
> As shown in [Table R4_1](https://anonymous.4open.science/r/MoRE-ICLR2026-Rebuttal-21A4/m2ED/Table%20R4_1.pdf), **our proposed method achieves highly competitive performance** across all three LPIPS-based metrics: demonstrating strong unlearning of the target style (high LPIPS_f), minimal distortion to remain styles (low LPIPS_r), and the best overall tradeoff (highest LPIPS_d). Qualitative results shown in [Figure R4_2](https://anonymous.4open.science/r/MoRE-ICLR2026-Rebuttal-21A4/m2ED/Figure%20R4_2.pdf) further support these findings. Notably, **ours is the only method that successfully removes Van Gogh’s iconic artistic style while faithfully adhering to the input prompt**, generating a coherent image of a starry night over a town without reproducing the signature spiral patterns or brush stroke textures.
>
> **These results are highly significant; our proposed method is applied to diffusion models entirely out of the box, with no architecture-specific adaptation, no hyperparameter tuning, and no additional engineering. Despite this, it outperforms SOTA diffusion model unlearning methods both quantitatively and qualitatively.** Since our current implementation does not yet leverage any diffusion-specific components, we believe that modest, targeted adaptations could further amplify its effectiveness. This opens up a highly promising avenue for future research, with the potential to drive substantial impact across both the generative modeling and unlearning communities.
>
> # m2ED_W3a Full rank assumption of the prototype matrix
>
> Thank you for raising this point. As you correctly noted, our prototype matrix $P \in \mathbb{R}^{d \times k}$ must be full-rank for $DP = I_k$ to hold. We agree that this assumption should be explicitly stated in the paper, and we will revise the manuscript accordingly.
>
> # m2ED_W3b The use of $D$ vs. $P^+$ in Equation 4
>
> As defined following Equation 2, $D$ and $P^+$ refer to the same pseudoinverse matrix. We will revise the manuscript to use consistent notation throughout to avoid any potential confusion.

---

> ### Author Response · Authors · 2025-11-25
> **Rebuttal by Authors (3/3)**
>
> # m2ED_W3c+Q2 Is $I-PP^+ = 0$?
>
> **Since our prototype matrix $P$ is a non-square skinny matrix, it is not invertible, and therefore cannot have both left and right inverse at the same time.** In other words, while $P^+P = I_k$ always holds for the pseudoinverse, $PP^+ = I_d$ does not.
>
> Note that $P$ is a tall (skinny) matrix, hence it maps a lower-dimensional space ($\mathbb{R}^k$) into a higher-dimensional space ($\mathbb{R}^d$). That means no information is lost during the linear transformation as it is injective, and hence invertible. To put that mathematically: $x = P^+ P x$, and therefore $P^+ P = I_k$. On the other hand, $P^+$ is a fat matrix: it maps a higher-dimensional space ($\mathbb{R}^d$) into a lower-dimensional space ($\mathbb{R}^k$), where information will be lost as the transformation is surjective. This means the transformation is not invertible. We also show this mathematically using our SVD-based pseudoinverse formulation (see [Proof R4_1](https://anonymous.4open.science/r/MoRE-ICLR2026-Rebuttal-21A4/m2ED/Proof%20R4_1.pdf)).
>
> **To conclude, since $PP^+ \neq I_d$, complement-space projection has utility**. More specifically, $PP^+$ acts as an orthogonal projector onto the column space of $P$, the complement $I - PP^+$ projects onto the nullspace orthogonal to it. This operator retains and manipulates components that are not representable within the subspace spanned by $P$.
>
> # Closing Remarks
> We extend our most sincere thanks to the reviewer again for the helpful suggestions and the opportunity to improve our work. Below, we summarize the key changes made during this rebuttal.
>
> - **Clarified applicability to both open- and closed-box settings**:, clarified that the linear projection layers can be absorbed into the weight parameters of the preceding layers, thereby preventing removal of the projection layer to undo unlearning.
> - **Demonstrated broader applicability by applying our method directly to a large generative model (Stable Diffusion v1.4)**: demonstrated how class-wise unlearning can be extended to concept-unlearning in large generative models. Our method achieved SOTA performance without any architectural modifications or hyperparameter tuning.
> - **Clarified and elaborated on our method's mathematical formulations.**
>
> These additions highlight the correctness, generality, and broader applicability of our method across multiple tasks (class unlearning, concept unlearning, random unlearning), multiple models (discriminative and generative), and multiple contexts (open- and closed-box settings). We believe we have addressed all major concerns raised in the reviews, and we do not see any remaining issues that should prevent an upward adjustment of the current evaluation.
>
> Thank you once again for your kind support and suggestions.
>
> &nbsp;
>
> Yours sincerely,
>
> The authors of MoRE
>
> --
>
> &nbsp;
>
>
> *References*
>
> [1] Lee et al., "ESC: Erasing Space Concept for Knowledge Deletion," CVPR 2025.
>
> [2] Golatkar et al., "Eternal Sunshine of the Spotless Net: Selective Forgetting in Deep Networks," CVPR 2020
>
> [3] Thudi et al., "Unrolling sgd: Understanding factors influencing machine unlearning," EuroS&P 2022
>
> [4] Kim et al., "Layer attack unlearning: Fast and accurate machine unlearning via layer level attack and knowledge distillation," AAAI 2024
>
> [5] Kumari et al., "Ablating concepts in text-toimage diffusion models," ICCV 2023
>
> [6] Gandikota et al., "Erasing concepts from diffusion models," CVPR 2023
>
> [7] Schramowski et al., "Safe latent diffusion: Mitigating inappropriate degeneration in diffusion models," CVPR 2023
>
> [8] Yoon et al., "Safree: Training-free and adaptive guard for safe text-to-image and video generation," ICLR 2025
>
> [9] Gong et al., "Reliable and efficient concept erasure of text-to-image diffusion models," ECCV 2024
>
> [10] Gandikota et al., "Unified concept editing in diffusion models," WACV 2024
>
> [11] Zhang et al., "The Unreasonable Effectiveness of Deep Features as a Perceptual Metric," CVPR 2018

---

> > ### Comment · Reviewer_m2ED · 2025-11-26
> >
> > I thank the authors for the detailed response. I appreciated their clarification on how to make MoRE compatible with white-box access and how it can be applied to random unlearning (thanks for pointing to Table 3) and concept unlearning. Furthermore, I appreciated the clarifications on the theoretical side. I still have a couple of questions:
> >
> > 1. I thank the authors for Figure R4_1, but could the authors also add their method alongside for comparison?
> > 2. Could the authors provide a brief pseudocode for random unlearning? Although I understand the main idea, I want to ensure I fully grasp the procedure.
> > 3. Could the authors provide the original model performance in Table 3? The method shows 100% forget accuracy; thus, most of the unlearning is explained by the MIA, which is fine, but I need the original model MIA to check whether unlearning was somewhat successful.

---

> > > ### Author Response · Authors · 2025-11-28
> > > **Follow-up by Authors**
> > >
> > > Thank you for your prompt response and for the additional feedback. We sincerely appreciate the time and care you have taken to engage us, and we are happy to hear that our previous clarifications were helpful. Below, we provide our responses to the additional questions raised.
> > >
> > > &nbsp;
> > >
> > > **R1** Thank you for the suggestion. We have updated the results accordingly and added our method to the comparison in [Figure R4_3](https://anonymous.4open.science/r/MoRE-ICLR2026-Rebuttal-21A4/m2ED/Figure%20R4_3.pdf). Specifically, we report two variants of our approach: **Ours (I)**, where unlearning is applied at the classification head, and **Ours (II)**, where unlearning is applied at the penultimate layer. [Figure R4_3](https://anonymous.4open.science/r/MoRE-ICLR2026-Rebuttal-21A4/m2ED/Figure%20R4_3.pdf) shows that our method can **flexibly select the unlearning target layer, allowing parameter updates to be focused within the feature extractor**, which enables stronger privacy and security control.
> > >
> > > **R2** Yes, certainly. We have added the requested pseudocode for random unlearning in [Pseudo R4_1](https://anonymous.4open.science/r/MoRE-ICLR2026-Rebuttal-21A4/m2ED/Pseudo%20R4_1.pdf). We will also include this pseudocode in the appendix of the revised manuscript for clarity and ease of reference.
> > >
> > > **R3** Thank you for the suggestion. We have added the original model performance, including the original MIA score, to an updated version of Table 3 in [Table R4_2](https://anonymous.4open.science/r/MoRE-ICLR2026-Rebuttal-21A4/m2ED/Table%20R4_2.pdf).
> > >
> > > &nbsp;
> > >
> > > We thank the reviewer again for engaging with us, and we hope the clarifications provided here have addressed any remaining concerns.
> > >
> > > &nbsp;
> > >
> > > Yours faithfully,
> > >
> > > The authors of MoRE

---

> > > > ### Comment · Reviewer_m2ED · 2025-11-28
> > > >
> > > > I thank the authors for their prompt response to my additional questions. I suggest adding these new results and pseudocode to the paper, as I believe they will improve its quality.
> > > >
> > > > Overall, I believe the authors successfully addressed my first and third concerns. At the same time, I still have doubts about the applicability of this method for instance-wise unlearning (also confirmed by the relatively high MIA of Table R4_2). I will revisit my score and issue a final recommendation after the rebuttal.

---

### Official Review · Reviewer_FRtt · 2025-10-31

**Soundness:** 3
**Presentation:** 3
**Contribution:** 3
**Rating:** 6
**Confidence:** 3

**Summary:**

This paper proposes MoRE (Mixture of Remapping Experts), a feature-space unlearning method designed to address three major challenges in the unlearning literature: efficiency, scalability, and irreversibility. The method constructs class prototypes and applies a prototype-orthogonal projection step to decorrelate forget and remain prototypes. It then performs remapping by scattering forget features across multiple remain prototypes through a mixture-of-experts mechanism to disrupt the coherent structure of forgotten representations. MoRE achieves strong empirical results on benchmark datasets and against multiple baselines, and offers appealing simplicity, requiring no retraining of the base model.

**Strengths:**

- Clear motivation and problem framing. The paper is well-motivated around three challenges in unlearning—efficiency, scalability, and irreversibility—and situates MoRE clearly within that context.

- Elegant and lightweight formulation. MoRE’s prototype-based approach provides a simple, closed-form feature-space solution that avoids full retraining and can be applied post hoc to pretrained models.

- Strong scalability, efficiency, and negligible overhead. The method scales linearly with respect to the number of samples (N), feature dimensionality (d), and number of concepts or classes (k)—O(Nd) for prototype collection and O(kd) for storage—resulting in negligible computational and memory overhead. This efficiency allows MoRE to be extended to larger models and datasets without major architectural or computational modifications.

- Strong empirical performance. The paper presents comprehensive quantitative results across multiple datasets, baselines, and model architectures, showing consistent gains in both forget success and retain utility.

- Thorough ablation studies. The ablations are detailed and informative, isolating the impact of key components such as prototype orthogonalization, remapping, and routing, and providing clear insights into their contribution to performance.

**Weaknesses:**

I enjoyed reading this paper and found the proposed formulation interesting and thought-provoking. I do, however, have a few points I would like to discuss with the authors:

- **Conceptual framing of knowledge deletion (KD)**. I found the paper’s framing of KD somewhat confusing. KD is described as an alternative designed to address the limitations of using retrain-from-scratch as the gold standard for unlearning. However, this seems more like a shift in objective than a solution to a known limitation. In KD, the goal is not to replicate the distribution of a retrained (oracle) model but rather to “minimize the usefulness of forget knowledge while maximizing that of remain knowledge.” This effectively corresponds to removing selected concepts while maintaining or improving utility on the rest. From that perspective, KD appears conceptually closer to model editing—which modifies or removes specific knowledge—than to classical privacy-oriented unlearning, which focuses on reproducing the retrained model’s behavior under a privacy–utility tradeoff.

- **On traditional unlearning and privacy metrics**. The claim that prior unlearning methods focus solely on privacy metrics while ignoring utility is not entirely accurate. Traditional unlearning methods already aim to balance privacy and utility by approximating the retrained model, so the key distinction from KD lies in their objectives rather than in any inherent limitations.

- **On the evaluation metrics**. Related to the previous point, I found the discussion around evaluation metrics somewhat unclear. The paper states that “By introducing utility metrics alongside privacy measures, KD no longer treats the retrain-from-scratch model as the sole point of reference, thereby enabling a more robust and practical evaluation.” However, I did not observe any new metrics being formally introduced. Traditional unlearning works already evaluate both privacy and utility—typically through accuracy on different subsets of data and membership inference attacks (MIA). The harmonic mean appears to be a concise way to summarize these results, and the KR score corresponds to accuracy after linear probing, which has also been used in prior work to assess robustness. It would be helpful if the authors could further clarify what is meant by “introducing utility metrics” and how this differs from existing evaluation practices in unlearning.

- **On the underlying assumptions and scope of applicability**. While the proposed method appears effective within its intended setting, it seems to rely on the assumption that the model functions as a closed system. My understanding is that prior to the integration of the MoRE component, the underlying feature extractor remains unchanged, meaning that any access to internal representations or earlier layers could potentially expose the forgotten information. Is this understanding correct? If so, the method would seem best suited for scenarios where third parties cannot directly access or fine-tune internal layers. It would be helpful if the authors could clarify whether this closed-box assumption is inherent to the formulation, and how the approach might behave in more open or fine-tunable settings.


- **On scalability and applicability to larger models.** The notion of knowledge deletion seems particularly relevant for large language models (LLMs), where removing specific concepts or behaviors would be highly valuable. The paper mentions that MoRE scales linearly with respect to the number of samples (N), feature dimensionality (d), and number of classes (k), which addresses computational efficiency. However, it is less clear whether this claim also extends to numerical stability and conceptual validity. As the number of classes or concepts grows, prototype orthogonalization and pseudo-inverse computations may become ill-conditioned, and the assumption that each class can be represented by a single prototype may no longer hold. It would be helpful if the authors could clarify whether the scalability claim covers these aspects or primarily refers to runtime efficiency.

- **On baseline selection**. The set of baselines compared in the paper seems somewhat limited given how active the unlearning area has become. Many new methods are introduced each year. It would be helpful to include comparisons against existing feature-based unlearning methods, as these share a closer methodological foundation with the proposed approach.

I think this is an interesting approach with practical potential, and I would be willing to revise my rating upward to an 8 if the above concerns are adequately addressed.

**Questions:**

I would appreciate it if the authors could clarify the following points:

- **On instance-based unlearning**: Why do you think this method can be applied in instance-based unlearning scenarios? It is not entirely clear to me whether this framework is well-suited for instance-level forgetting. The method is formulated around class/concept prototypes, which makes it naturally more compatible with class-level unlearning than with removing individual samples. Looking at Table 3, it appears that relatively little unlearning has occurred in the instance-based setup using the proposed approach. To better understand the extent of forgetting, it would be helpful if you could include the results for the original (pre-unlearning) model at the top of this table, so that the relative change in forget and retain performance can be compared directly.

- **On utility improvement**: Why do you think utility improves using this method? In particular, for CIFAR-100, the performance on D_rt
​seems to have significantly increased compared to the oracle.

- **On scalability to larger datasets**: I do not see the ImageNet results referenced in Figure 4. Does the improvement in memory and time efficiency remain significant for larger datasets?

---

> ### Author Response · Authors · 2025-11-25
> **Rebuttal by Authors (1/5)**
>
> # Opening remarks
>
> We are deeply grateful to the reviewer for the time, thoughtful analysis, and constructive suggestions. Your feedback has been immensely helpful in sharpening the clarity of our presentation and highlighting areas for further elaboration. We have taken your comments seriously and made concrete efforts to address each point raised as faithfully as possible.
>
> Below, we provide detailed, point-by-point responses to the identified weaknesses (W#) and questions (Q#). For multi-part bullets, we divide them into subpoints (e.g., W1a, W1b) to ensure clarity and precision in our replies.
>
> We have also included several **additional figures/tables in our response using anonymous external links**. Please note that depending on your browser or PDF viewer, some **fonts or formatting may appear distorted**. If this occurs, we kindly ask you to either **download the PDF directly or click “View Raw”** to ensure proper rendering.
>
> # FRtt_W1a: Conceptual framing of knowledge deletion (KD)
>
> We sincerely thank the reviewer for this thoughtful and perceptive comment. Your interpretation of Knowledge Deletion (KD) is precisely aligned with our intent, and we appreciate the opportunity to clarify its positioning more explicitly.
>
> As noted in Line 126, **we frame KD as an *extended interpretation* of the standard Machine Unlearning (MU) formulation**, and not as a new paradigm, nor as a replacement for the retrain-from-scratch oracle or its evaluation protocol. KD inherits the core MU objective (removal of forget knowledge) as well as its established metrics (utility, privacy, etc.), which is why we continue to include the retrained model as a gold-standard reference (as shown in Table 1).
>
> **The key distinction between MU and KD**, as astutely identified by the reviewer, is that while the KD task retains the core MU objectives, it additionally introduces a utility-focused formulation, where the emphasis shifts from reproducing the retrained model’s behavior to explicitly minimizing the task utility of the forgotten knowledge while preserving that of the retained knowledge.
>
>
> **Action plan**: To address this concern, we will revise the related‑works section to explicitly clarify that KD is not positioned as a replacement for MU, but rather as an augmented formulation that enables additional evaluation focused on the utility perspective. This revision will emphasize that KD preserves the fundamental MU objectives and evaluation metrics while providing a complementary lens for analyzing how effectively methods suppress the usefulness of forgotten knowledge without compromising the remain knowledge.
>
> # FRtt_W1b: Distinction between KD and model editing
>
> We thank the reviewer for this thoughtful observation. We agree that KD bears surface-level resemblance to model editing. Nonetheless, we position KD as a formulation within the Machine Unlearning (MU) framework, as it inherits the full set of MU objectives and evaluation metrics, as discussed in our previous response.
>
> **Action plan**: We will update the KD subsection to clearly state that KD is an extension of MU—retaining its goals and evaluation setup—while briefly acknowledging its similarity to model editing at the surface level. This should help clarify the intended scope and positioning of our work.
>
> # FRtt_W2: On traditional unlearning and privacy metrics
>
> We thank the reviewer for giving us the chance to clarify this. It was not our intention to suggest that traditional MU methods ignore utility. **Our aim was to highlight that much of the MU literature has been shaped by privacy-driven motivations** (hence the focus on replicating the retrained model as a gold standard) whereas KD shifts the objective more towards employing utility as the *additional* lens for evaluating the success of knowledge removal.
>
> **Action plan**: We suspect that phrases like “*the current formulation of MU has largely been shaped by privacy principles*” (Line 111) may have unintentionally conveyed an imbalanced view, and we will revise this wording to avoid such implications. We will also make it clear that KD preserves the MU evaluation tradition in full, while augmenting it with the KR setting to offer additional insight into the retained vs. forgotten utility.

---

> ### Author Response · Authors · 2025-11-25
> **Rebuttal by Authors (2/5)**
>
> # FRtt_W3: On the evaluation metrics
>
> We appreciate the reviewer’s comments and the opportunity to clarify. **To begin, we would like to emphasize that we are not claiming to introduce new utility metrics in this work.** Both the KD formulation and the associated evaluation metrics were originally proposed by ESC [1], which, to the best of our knowledge, was the first to formalize linear probing as a method for assessing knowledge deletion and unlearning robustness.
>
> **As explicitly stated in our paper (Lines 126–127), we adopt the KD framework and its evaluation protocol as is**. This includes combining standard MU metrics (e.g., retain/forget accuracy, MIA) with linear probing to assess the unlearning performance at the feature level.
>
> **Action plan**: We will update the manuscript to explicitly reaffirm that both the KD formulation and its evaluation protocol, including linear probing–based utility metrics, originate from ESC. We will also clarify that our work directly follows this framework without proposing new metrics.
>
> # FRtt_W4a: On the underlying assumptions and scope of applicability - possibility of forgotten information exposure in shallower layers
>
> We thank the reviewer for raising this important point. Your concern about forgotten information being exposed or leaked from earlier layers is understandable. However, **class-specific information leakage through early layers is unlikely, as these layers predominantly encode low-level, task-agnostic features rather than class-discriminative semantics**. This has been consistently demonstrated in prior work, including ESC [1] and several other MU baselines [4], which show that unlearning effects are concentrated in deeper layers where class-wise separability emerges (see [Figure R3_1](https://anonymous.4open.science/r/MoRE-ICLR2026-Rebuttal-21A4/FRtt/Figure%20R3_1.pdf)). This view is also supported by findings in the interpretability literature [2,3] and is widely accepted in the community.
>
> To further support this, we visualize the latent feature space at different layers using t-SNE (see [Figure R3_2](https://anonymous.4open.science/r/MoRE-ICLR2026-Rebuttal-21A4/FRtt/Figure%20R3_2.pdf)). When plotting the input features to the classification head, we observe clearly separable clusters corresponding to each class. In contrast, **features extracted from the penultimate are highly entangled and do not form distinct clusters, indicating that class-specific information is not readily accessible at these stages of the network**.
>
> **To comply with stricter safety and privacy requirements, MoRE can also be applied to shallower layers *within* the feature extractor**. As shown in Table 6 of the original manuscript (Line 460-475) even when applied to earlier feature layers, MoRE effectively disrupts class-specific information, achieving strong unlearning performance on the forget set. As expected, this broader disruption comes at the cost of higher retain-set performance degradation due to the perturbation of more generalizable features. Nonetheless, this helps ease the concern regarding potential forgotten information exposure in shallower layers and achieve stronger safety and privacy guarantees when needed.
>
> # FRtt_W4b: On the underlying assumptions and scope of applicability - Behaviors in open and fine-tuning settings
>
> We thank the reviewer for this thoughtful question and giving us the opportunity to clarify. First, we clarify that the **closed-box assumption is not inherent to the formulation of our proposed method**. While MoRE works naturally in closed systems, it is also compatible with more open settings where internal weights may be accessible or fine-tunable.
>
> In fact, **MoRE is designed to be robust even under fine-tuning**. As demonstrated in our KR evaluation (Table 1, Lines 324–355), we observe that simply **fine-tuning the classification head is enough to completely recover forgotten knowledge for most if not all SOTA unlearning methods**. In contrast, MoRE maintains unlearning performance even after this fine-tuning step, owing to its use of *non-trainable* remapping experts that preserve the unlearning effect.
>
> **Action plan**: We will revise the manuscript to clarify that the closed-box assumption is not a requirement for MoRE, and explicitly discuss its robustness in open and fine-tunable settings. We will also expand the discussion on layer-wise leakage, incorporating t-SNE visualizations and additional analysis on shallow-layer unlearning to address concerns about residual class-specific information. References to Table 1 and Table 6 will be added to highlight MoRE’s resilience under fine-tuning and its flexible placement within the feature extractor.

---

> ### Author Response · Authors · 2025-11-25
> **Rebuttal by Authors (3/5)**
>
> # FRtt_W5a: On scalability claim
>
> We thank the reviewer for this thoughtful question and the opportunity to clarify. **Our scalability claim primarily refers to computational and memory efficiency**.
>
> **That said, we empirically observe that MoRE remains numerically stable even on large-scale datasets**. In particular, on ImageNet (1,000 classes), both the prototype orthogonalization and pseudo-inverse operations remain well-conditioned, and MoRE achieves state-of-the-art performance—achieving a forget accuracy close to 1% while maintaining high retain accuracy (see Table 9, Lines 900–912).
>
> Regarding the reviewer’s point on the validity of using a single prototype per class, we agree this is an important consideration. While **we did not encounter issues using single-class prototypes** in our benchmarks, we acknowledge that this assumption may become limiting in datasets with strong intra-class variation or multi-modal structure. A straightforward extension would be to **replace our mean-based prototypes with more expressive clustering techniques** (e.g., k-means centroids or Gaussian mixtures). Since the orthogonalization step in MoRE is agnostic to the prototype generation method, such techniques can be seamlessly integrated to better accommodate more complex distributions.
>
> # FRtt_W5b: On applicability to larger models: Stable Diffusion v1.4
>
> To demonstrate broader applicability to large-scale models, **we implement our method on the text-to-image generation model, namely Stable Diffusion (SD) v1.4**. Following prior works [9,10], we apply prototype orthogonalization, erasure, and remapping to the cross-attention layers, using tokenized input prompts to construct prototypes. Our experimental setup also strictly follows the standard practice established by the SOTA diffusion unlearning methods [6,8–10], ensuring a fair comparison.
>
> **We evaluate unlearning performance in the artistic style erasure task**, which has emerged as a standard benchmark for testing concept-level unlearning in generative models. Following prior works [8, 9], we construct an evaluation set using 20 prompts for each of 10 artists: 5 classical (Van Gogh, Pablo Picasso, Rembrandt, Andy Warhol, Caravaggio) and 5 modern (Kelly McKernan, Thomas Kinkade, Tyler Edlin, Kilian Eng, and the anime series Ajin: DemiHuman), all of whom are reported to be mimicked by SD. We apply MoRE to remove two artistic styles: Van Gogh and Kelly McKernan. And the unlearning performance is measured using the LPIPS scores [11], which compares the generated outputs before and after unlearning to measure visual similarity (lower means more similar). We report three metrics:
> - **LPIPS_f (forget artists)**: LPIPS score computed on the forget artists (higher is better)
> - **LPIPS_r (remain artists)**: LPIPS score on the remain artists (lower is better, indicating minimal disruption)
> - **LPIPS_d = LPIPS_f − LPIPS_r**: the overall tradeoff, capturing how well the method removes target styles while preserving unrelated ones
>
> As shown in [Table R3_1](https://anonymous.4open.science/r/MoRE-ICLR2026-Rebuttal-21A4/FRtt/Table%20R3_1.pdf), **our proposed method achieves highly competitive performance** across all three LPIPS-based metrics: demonstrating strong unlearning of the target style (high LPIPS_f), minimal distortion to remain styles (low LPIPS_r), and the best overall tradeoff (highest LPIPS_d). Qualitative results shown in [Figure R3_3](https://anonymous.4open.science/r/MoRE-ICLR2026-Rebuttal-21A4/FRtt/Figure%20R3_3.pdf) further support these findings. Notably, **ours is the only method that successfully removes Van Gogh’s iconic artistic style while faithfully adhering to the input prompt**, generating a coherent image of a starry night over a town without reproducing the signature spiral patterns or brush stroke textures.
>
> **These results are highly significant; our proposed method is applied to diffusion models entirely out of the box, with no architecture-specific adaptation, no hyperparameter tuning and no additional engineering. Despite this, it outperforms SOTA diffusion model unlearning methods both quantitatively and qualitatively.** Since our current implementation does not yet leverage any diffusion-specific components, we believe that modest, targeted adaptations could further amplify its effectiveness. This opens up a highly promising avenue for future research, with the potential to drive substantial impact across both the generative modeling and unlearning communities.

---

> ### Author Response · Authors · 2025-11-25
> **Rebuttal by Authors (4/5)**
>
> # FRtt_W6: On baseline selection
>
> We thank the reviewer for raising this point regarding the breadth of baseline comparisons.
>
> To the best of our knowledge, ESC [1] remains the only prior work that directly tackles feature-based machine unlearning, sharing the most methodological similarity with our proposed approach. While many unlearning methods have been introduced in recent years, most focus on parameter-based or data-centric techniques, making direct comparison less meaningful. Nonetheless, we fully agree that evaluating against a broad range of recent approaches is essential.
>
> To address this, we have added five additional state-of-the-art unlearning methods to our evaluation suite:
> - [12] Zhang et al., "Towards Certified Unlearning for Deep Neural Networks," ICML 2024
> - [13] Cha et al., "Learning to Unlearn: Instance-wise Unlearning for Pre-trained Classifiers," AAAI 2024
> - [14] Bonato et al., "Is Retain Set All You Need in Machine Unlearning?," ECCV 2024
> - [15] Kodge et al., "Deep Unlearning: Fast and Efficient Gradient-Free Class Forgetting," TMLR 2024
> - [16] Ebrahimpour-Boroojeny et al., "Not All Wrong is Bad: Using Adversarial Examples for Unlearning," ICML 2025
>
> With these additions, **our work now compares against 17 diverse unlearning baselines**. Across all evaluations, our method consistently achieves superior performance in both forgetting and retention metrics (see [Table R3_2](https://anonymous.4open.science/r/MoRE-ICLR2026-Rebuttal-21A4/FRtt/Table%20R3_2.pdf)).
>
> We believe **this represents one of the most comprehensive empirical comparisons in the machine unlearning literature to date**. We hope this expanded benchmark provides a strong foundation for future studies and reinforces the practical and methodological advantages of our approach.
>
> # FRtt_Q1: On instance-based unlearning
>
> We thank the reviewer for this insightful question. As rightly noted, our method was originally designed for class-level or concept-level unlearning, where the objective is to remove well-defined semantic groups. Nonetheless, **we explored how the proposed framework can be adapted for instance-based unlearning to demonstrate its flexibility and broader applicability**.
>
> To support this, we introduced a simple yet effective modification: for each class, we compute two separate prototypes—one for the forget samples (i.e., the mean of all forget instances in that class) and another for the retain samples. We find that because the number of forget samples is small, these two means often diverge meaningfully. This observation allows us to perform prototype adjustment such that the forget-group activation is pushed closer to the retain-group mean, effectively erasing unique signals specific to the forgotten instances.
>
> While this adaptation does not replicate the forget accuracy of the retrain ground truth—as expected, since it operates without full retraining—it significantly improves MIA (Membership Inference Attack) performance. Our method achieves the second-best average gap score, which balances between utility and privacy metrics.
>
> We emphasize that our goal in including random instance-level unlearning experiments is not to claim perfect suitability for this task, but rather to demonstrate the adaptability and generalization potential of our approach, even in settings outside its core design scope.
>
> # FRtt_Q2: On utility improvement
>
> We thank the reviewer for this observation. The drop in D_rt accuracy for the Oracle baseline can be attributed to training on 10% less data, which may lead to suboptimal local minima. Similar trends have been reported in prior MU works [1,17], where retraining from scratch with less data resulted in poorer model performance. Our method performs unlearning on the original model trained with the full dataset, and benefits from a better initialization.
>
> On the other hand, our method outperforms other MU baselines because prototype orthogonalization ablates forget knowledge while preserving the remain knowledge. In contrast, existing methods often degrade remain feature representations during the unlearning process. As a result, our method achieves 0% forget accuracy while maintaining high D_rt performance, avoiding the utility drop seen in prior approaches.
>
> # FRtt_Q3: On scalability to larger datasets
>
> We thank the reviewer for raising this point. Yes, our method scales effectively to large-scale datasets such as ImageNet. As shown in [Figure R3_4](https://anonymous.4open.science/r/MoRE-ICLR2026-Rebuttal-21A4/FRtt/Figure%20R3_4.pdf), MoRE operates on ImageNet using only 6GB of VRAM, whereas other training-free baselines require at least 20GB and up to 60GB. This substantial memory reduction highlights the scalability of our approach and its practicality for real-world unlearning tasks on large datasets.

---

> > ### Author Response · Authors · 2025-11-25
> > **Rebuttal by Authors (5/5)**
> >
> > # Closing Remarks
> >
> > We sincerely thank the reviewer once again for the constructive feedback and the opportunity to improve our work. Below, we summarize the key changes made during this rebuttal:
> > - **Clarified our framing of knowledge deletion formulation** and corresponding experimental setup, addressing confusion around unlearning formulation.
> > - **Clarified applicability to both open- and closed-box settings**, and showed that our method is significantly more robust under fine-tunable scenarios compared to other MU baselines.
> > - **Demonstrated broader applicability by applying our method directly to a large generative model (Stable Diffusion v1.4)** without any architectural modifications or hyperparameter tuning, achieving SOTA performance.
> > - **Validated scalability to large-scale datasets like ImageNet**, requiring only 6GB VRAM—substantially less than other training-free baselines (20–60GB).
> >
> > These additions highlight the generality, scalability, and practical impact of our method across multiple tasks and model types. We believe we have addressed all major concerns raised in the reviews, and we do not see any remaining issues that should prevent an upward adjustment of the current evaluation.
> >
> > Thank you once again for your kind support and suggestions.
> >
> > &nbsp;
> >
> > Yours sincerely,
> >
> > The authors of MoRE
> >
> > --
> >
> > &nbsp;
> >
> > *References*
> >
> > [1] Lee et al., "ESC: Erasing Space Concept for Knowledge Deletion," CVPR 2025.
> >
> > [2] Zeiler et al., "Visualizing and understanding convolutional networks," ECCV 2014.
> >
> > [3] Yosinski et al., "How transferable are features in deep neural networks?" NeurIPS 2014.
> >
> > [4] Kim et al., "Layer attack unlearning: Fast and accurate machine unlearning via layer level attack and knowledge distillation," AAAI 2024
> >
> > [5] Kumari et al., "Ablating concepts in text-toimage diffusion models," ICCV 2023
> >
> > [6] Gandikota et al., "Erasing concepts from diffusion models," CVPR 2023
> >
> > [7] Schramowski et al., "Safe latent diffusion: Mitigating inappropriate degeneration in diffusion models," CVPR 2023
> >
> > [8] Yoon et al., "Safree: Training-free and adaptive guard for safe text-to-image and video generation," ICLR 2025
> >
> > [9] Gong et al., "Reliable and efficient concept erasure of text-to-image diffusion models," ECCV 2024
> >
> > [10] Gandikota et al., "Unified concept editing in diffusion models," WACV 2024
> >
> > [11] Zhang et al., "The Unreasonable Effectiveness of Deep Features as a Perceptual Metric," CVPR 2018
> >
> > [12] Zhang et al., "Towards Certified Unlearning for Deep Neural Networks," ICML 2024
> >
> > [13] Cha et al., "Learning to Unlearn: Instance-wise Unlearning for Pre-trained Classifiers," AAAI 2024
> >
> > [14] Bonato et al., "Is Retain Set All You Need in Machine Unlearning?," ECCV 2024
> >
> > [15] Kodge et al., "Deep Unlearning: Fast and Efficient Gradient-Free Class Forgetting," TMLR 2024
> >
> > [16] Ebrahimpour-Boroojeny et al., "Not All Wrong is Bad: Using Adversarial Examples for Unlearning," ICML 2025
> >
> > [17] Golatkar et al., "Eternal Sunshine of the Spotless Net: Selective Forgetting in Deep Networks," CVPR 2020

---

### Official Review · Reviewer_MYdz · 2025-10-31

**Soundness:** 3
**Presentation:** 3
**Contribution:** 2
**Rating:** 4
**Confidence:** 4

**Summary:**

The paper proposes a method that projects forget and remain prototypes into orthogonal subspaces. By enforcing orthogonality, removing the forget prototype should not affect the remain prototype, thereby better preserving remain knowledge. In addition, the method introduces a Mixture of Remapping Experts to remap forget prototype into remain prototypes.

**Strengths:**

This work identifies a key limitation of existing methods that removing the forget prototype can affect the remain prototype. To address this issue, the authors propose a prototype-orthogonal projection that enforces orthogonality between the forget and remain prototypes, effectively minimizing influence on the remain information. The proposed approach sounds reasonable.


The paper is well written and easy to follow.

**Weaknesses:**

1) Although the Prototype-Orthogonal projection enforces orthogonality, it implicitly assumes that the forget and remain prototypes are not highly entangled. As illustrated in Figure 2, when the entanglement is strong, the PO subspace may fail to capture the principal features of the data. In such cases, as shown in Equation (4), the introduced complement-space projection retains components of the forget prototypes. This suggests that strongly entangled forget prototypes cannot be fully unlearned, thereby limiting the effectiveness of the proposed method.

2) Experiments focus on concept-wise unlearning. The paper does not evaluate instance-wise unlearning (random forgetting), subclass unlearning, or unlearning in VLMs and generative models. This narrows the claim and leaves open whether the method robustly handles entangled features.

3) The results in Table 6 indicate that the proposed method becomes less effective in higher feature entanglement. Specifically, the stronger entanglement is in mid-layer representations, compared to the final layer. The results weaken the generality and robustness of the approach.

4) Some related works are missing and lack discussion (e.g., [1]).

[1] Deep unlearning: Fast and efficient gradient-free class forgetting.

**Questions:**

1) How does performance degrade as entanglement between forget and remain prototypes increases? Can you quantify this relationship (e.g., with a cosine similarity or CKA) and provide ablation studies?

2) In highly entangled cases, does the complement-space projection risk reintroducing forget information? How is leakage prevented or measured?

3) Please report results on instance-wise forgetting and subclass unlearning to support broader applicability.

---

> ### Author Response · Authors · 2025-11-27
> **Rebuttal by Authors (1/3)**
>
> # Opening remarks
>
> We are deeply grateful to the reviewer for the time, thoughtful analysis, and constructive suggestions. Your feedback has been immensely helpful in sharpening the clarity of our presentation and highlighting areas for further elaboration. We have taken your comments seriously and made concrete efforts to address each point raised as faithfully as possible.
>
> Below, we provide detailed, point-by-point responses to the identified weaknesses (W#) and questions (Q#). For multi-part bullets, we divide them into subpoints (e.g., W1a, W1b) to ensure clarity and precision in our replies.
>
> We have also included several **additional figures/tables in our response using anonymous external links**. Please note that depending on your browser or PDF viewer, some **fonts or formatting may appear distorted**. If this occurs, we kindly ask you to either **download the PDF directly or click “View Raw”** to ensure proper rendering.
>
> # MYdz_W2+Q3 Experiments on instance-wise unlearning and generative model
>
> Thank you for this comment. We would like to clarify that **instance-wise unlearning (or random forgetting) is already included in our original manuscript: Table 3 (Lines 378–390) and Section 4.3 (Lines 458–470).** While our method is designed primarily for class-wise unlearning, these results demonstrate the extended applicability of our method to broader range of tasks.
>
> And to demonstrate that our proposed method can handle much more complex task and higly entanged features robustly, **we extend our investigation to concept unlearning tasks in text-to-image generation model, namely Stable Diffusion v1.4**. Concept unlearning is closely related to class-wise unlearning and is especially relevant in the era of generative models. We believe this extension is more appropriate and impactful.
>
> Following prior works [6, 7], we apply prototype orthogonalization, erasure, and remapping to the cross-attention layers, using tokenized input prompts to construct prototypes. Our experimental setup also strictly follows the standard practice established by the SOTA diffusion unlearning methods [3,5–7], ensuring a fair comparison.
>
> **We evaluate unlearning performance in the artistic style erasure task**, which has emerged as a standard benchmark for testing concept-level unlearning in generative models. Following prior works [5, 6], we construct an evaluation set using 20 prompts for each of 10 artists: 5 classical (Van Gogh, Pablo Picasso, Rembrandt, Andy Warhol, Caravaggio) and 5 modern (Kelly McKernan, Thomas Kinkade, Tyler Edlin, Kilian Eng, and the anime series Ajin: DemiHuman), all of whom are reported to be mimicked by SD. We apply MoRE to remove two artistic styles: Van Gogh and Kelly McKernan. And the unlearning performance is measured using the LPIPS scores [8], which compares the generated outputes before and after unlearning to measure visual similarity (lower means more similar). We report three metrics:
> - **LPIPS_f (forget artists)**: LPIPS score computed on the forget artists (higher is better)
> - **LPIPS_r (remain artists)**: LPIPS score on the remain artists (lower is better, indicating minimal disruption)
> - **LPIPS_d = LPIPS_f − LPIPS_r**: the overall tradeoff, capturing how well the method removes target styles while preserving unrelated ones
>
> As shown in [Table R2_1](https://anonymous.4open.science/r/MoRE-ICLR2026-Rebuttal-21A4/MYdz/Table%20R2_1.pdf), **our proposed method achieves highly competitive performance** across all three LPIPS-based metrics: demonstrating strong unlearning of the target style (high LPIPS_f), minimal distortion to remain styles (low LPIPS_r), and the best overall tradeoff (highest LPIPS_d). Qualitative results shown in [Figure R2_1](https://anonymous.4open.science/r/MoRE-ICLR2026-Rebuttal-21A4/MYdz/Figure%20R2_1.pdf) further support these findings. Notably, **ours is the only method that successfully removes Van Gogh’s iconic artistic style while faithfully adhering to the input prompt**, generating a coherent image of a starry night over a town without reproducing the signature spiral patterns or brush stroke textures.
>
> Importantly, this is achieved out of the box, without architecture-specific tuning, hyperparameter search, or diffusion-specific engineering. Despite this, the method outperforms specialized diffusion unlearning baselines, suggesting strong robustness to entangled features. We believe this addresses the concern directly and strengthens the generality of the method beyond class-wise unlearning, and showcases the broader applicability of our proposed method.

---

> ### Author Response · Authors · 2025-11-27
> **Rebuttal by Authors (2/3)**
>
> # MYdz_W1 Assumption on weakly entangled features
>
> Thank you for this comment. **We respectfully clarify that our method does not rely on the assumption that forget and retain features are weakly entangled. On the contrary, our motivation is precisely that these features are often highly correlated in practice.**
>
> As shown in the responses above, we explicitly evaluate our method under settings with strongly entangled representations, namely concept unlearning in text-to-image models using Stable Diffusion v1.4. In this setting, stylistic concepts are deeply embedded in cross-attention layers and are far more intertwined than in standard classification tasks. **Despite this, our method achieves competitive performance and strong unlearning efficacy, demonstrating robustness under high entanglement and achieving superior performance even compared to the training-based baselines.**
>
> More importantly, Prototype-Orthogonalization explicitly decorrelates forget and retain features. Existing projection-based methods typically operate without accounting for the correlation/entanglement between forget and remain features, which leads to remain accuracy degradation. By contrast, **PO conditions the projection on both prototypes and actively maps the features into a disentangled space**, enabling precise removal/remapping of forget information while preserving remain features.
>
> The disentangling capability of our method is supported by the strong performance we observe in practice. For example, on CIFAR-10 (Table 1), our method uniquely maintains 99% remain accuracy while reducing forget accuracy to nearly zero. **This result outperforms all prior work, including training-based methods, and establishes a new state-of-the-art benchmark.**
>
> # MYdz_W3+Q1 Reason for performance degradation at shallower layers
>
> Thank you for this insightful comment. As demonstrated in our diffusion-model experiments, our method remains effective in settings with highly entangled features. The performance degradation associated with unlearning at shallower layers arises because class-discriminative information is not yet formed at early or mid-level representations. These layers primarily encode low-level and task-agnostic features such as edges, textures, and shapes, rather than semantic class identity. Therefore, attempting to remove class-specific information at those stages is inherently limited.
>
> This observation is consistent with prior findings in unlearning and interpretability research. For example, ESC [1] and other unlearning baselines [4] report that effective unlearning occurs in deeper layers where class-wise separability emerges (see [Figure R2_2](https://anonymous.4open.science/r/MoRE-ICLR2026-Rebuttal-21A4/MYdz/Figure%20R2_2.pdf)). Similar conclusions are well established in the interpretability literature [2,3], which consistently shows that semantic concepts are formed in deeper layers of the network.
>
> To further support this, we visualize the latent feature space at different layers using t-SNE (see [Figure R2_3](https://anonymous.4open.science/r/MoRE-ICLR2026-Rebuttal-21A4/MYdz/Figure%20R2_3.pdf)). When plotting the input features to the classification head, we observe clearly separable clusters corresponding to each class. In contrast, **features extracted from the penultimate are highly entangled and do not form distinct clusters, indicating that class-specific information is not readily accessible at these stages of the network**.  This explains the unlearning performance degradation at shallower layers.
>
> # MYdz_W4 Reinforcing related works
>
> We thank the reviewer for pointing us to this work [15]. We have taken your suggestion seriously to strengthen our related works and have added, in total, five latest unlearning baselines to our evaluation suites.  **With these additions, our work now compares against 17 diverse unlearning baselines. Across all evaluations, our method consistently achieves superior performance** in both forgetting and retention metrics (see [Table R2_2](https://anonymous.4open.science/r/MoRE-ICLR2026-Rebuttal-21A4/MYdz/Table%20R2_2.pdf)). **We believe this represents one of the most comprehensive empirical comparisons in the machine unlearning literature to date**. We hope this expanded benchmark provides a strong foundation for future studies and reinforces the practical and methodological advantages of our approach.

---

> ### Author Response · Authors · 2025-11-27
> **Rebuttal by Authors (3/3)**
>
> # MYdz_Q2 Risk of introducing forget knowledge via complement space projection
>
> Thank you for this very interesting question. **Empirically, we do not observe reintroduction of forget information through the complement-space projection**. Even in highly entangled settings such as diffusion model unlearning, our method achieves strong performance, which would not be possible if substantial forget information were being reintroduced.
>
> If reintroduction were to occur, **one possible case is that the forget data follows a multi-modal distribution and cannot be faithfully represented by a single prototype**. In such cases, a single activation mean may not fully capture all directions associated with the forget data, allowing residual components to remain in the complement space.
>
> While our current implementation uses a single prototype for efficiency, this limitation is not fundamental to the framework. Replacing mean-based prototypes with multiple prototypes obtained via clustering is a natural extension that could further reduce this risk. Although this would increase computation and storage slightly, it would remain far more efficient than gradient-based unlearning methods.
>
> # Closing Remarks
>
> We sincerely thank the reviewer once again for the constructive feedback and the opportunity to improve our work. Below, we summarize the key changes made during this rebuttal:
> - **Demonstrated broader applicability by applying our method directly to a large generative model (Stable Diffusion v1.4)** without any architectural modifications or hyperparameter tuning, achieving SOTA performance. This also testifies the generality of our proposed method by achieving SOTA performance in highly entangled setup.
> - **Clarified our method's working mechanism and experimental results**, such as clarifying that our method does not assume weakly-entangled features, that indeed we have already performed/presented instance-wise unlearning experiments in the main text, and the reason behind the performance degradation of unlearning at shallower layers.
> - **Added 5 new baselines for thorough evaluation** and showed that our method consistently performs better. With these additions, our work now compares against 17 diverse unlearning baselines and represents one of the most comprehensive empirical comparisons in the machine unlearning literature to date.
>
>
> These additions highlight the generality, scalability, and practical impact of our method across multiple tasks and model types. We believe we have addressed all major concerns raised in the reviews, and we do not see any remaining issues that should prevent an upward adjustment of the current evaluation.
>
> Thank you once again for your kind support and suggestions.
>
> &nbsp;
>
> Yours sincerely,
>
> The authors of MoRE
>
> --
>
> &nbsp;
>
>
> *References*
>
>
> *References*
>
> [1] Lee et al., "ESC: Erasing Space Concept for Knowledge Deletion," CVPR 2025.
>
> [2] Zeiler et al., "Visualizing and understanding convolutional networks," ECCV 2014.
>
> [3] Yosinski et al., "How transferable are features in deep neural networks?" NeurIPS 2014.
>
> [4] Kim et al., "Layer attack unlearning: Fast and accurate machine unlearning via layer level attack and knowledge distillation," AAAI 2024
>
> [5] Kumari et al., "Ablating concepts in text-toimage diffusion models," ICCV 2023
>
> [6] Gandikota et al., "Erasing concepts from diffusion models," CVPR 2023
>
> [7] Schramowski et al., "Safe latent diffusion: Mitigating inappropriate degeneration in diffusion models," CVPR 2023
>
> [8] Yoon et al., "Safree: Training-free and adaptive guard for safe text-to-image and video generation," ICLR 2025
>
> [9] Gong et al., "Reliable and efficient concept erasure of text-to-image diffusion models," ECCV 2024
>
> [10] Gandikota et al., "Unified concept editing in diffusion models," WACV 2024
>
> [11] Zhang et al., "The Unreasonable Effectiveness of Deep Features as a Perceptual Metric," CVPR 2018
>
> [12] Zhang et al., "Towards Certified Unlearning for Deep Neural Networks," ICML 2024
>
> [13] Cha et al., "Learning to Unlearn: Instance-wise Unlearning for Pre-trained Classifiers," AAAI 2024
>
> [14] Bonato et al., "Is Retain Set All You Need in Machine Unlearning?," ECCV 2024
>
> [15] Kodge et al., "Deep Unlearning: Fast and Efficient Gradient-Free Class Forgetting," TMLR 2024
>
> [16] Ebrahimpour-Boroojeny et al., "Not All Wrong is Bad: Using Adversarial Examples for Unlearning," ICML 2025

---

### Official Review · Reviewer_8wvp · 2025-10-31

**Soundness:** 2
**Presentation:** 1
**Contribution:** 2
**Rating:** 4
**Confidence:** 4

**Summary:**

The paper proposes MoRE, a training-free unlearning layer that operates in a prototype-orthogonal (PO) feature space and then erases & remaps “forget” class prototypes into one or more “remain” class prototypes to destroy separability and impede recovery. The method aims to (i) preserve remain utility, (ii) achieve irreversible feature-level unlearning, and (iii) be efficient in time and memory compared to ESC method. Experiments on CIFAR-10/100, and Tiny-ImageNet evaluate performance using some conventional metrics and a feature-space Knowledge Retention (KR) probe, with ablations and efficiency comparisons.

**Strengths:**

1. Authors present Compelling intuition & visuals. t-SNE shows forget clusters remain separable after ESC but are absorbed or scattered under remapping and MoRE.

2. The work is well-motivated by observing the shortcoming of prior methods (e.g., separability and memory usage)

3. How authors try to address each of the problems in prior methods is clearly explained.

**Weaknesses:**

1. One of the core contributions as claimed by the authors is the effectiveness of using multiple experts to spread the forgotten concepts across the embedding space. Still the ablation study on the number of experts, do not support this contribution, as increasing the number of experts does not improve the results for CIFAR-100 and Tiny-Imagenet (Figure 6). Even for CIFAR-10 the results remain the same for the most part. This also can be seen from the results in Table 11.

2. When applying your method to random data forgetting, if you compute the average embedding for the forget samples in a class, it almost overlaps the average embedding for the remaining samples in that class. How do you use your method to remember the forget samples by mapping the average embedding, while maintaining the accuracy of the model on the remaining samples with the same average embedding?!

3. From the explanation it seems that each class has a single prototype vector. In that case how can the autocorrelation of a prototype vector be less than one in Figure 3? The cosine similarity of a vector with itself should be 1! This also raises questions about the discussion in section 3.1.

4. The paper seems very confusing with the use of undefined notations throughout their method section. What is S_f in line 247? The paper would benefit a lot by adding a notation section.

5. This is also the case with many undefined abbreviations (D_r, D_rt, HM, HM_t) in section 4.1.

6. The names and notations do not match across the main body, tables, and captions: $D_{rt}$ is used in the caption and D_rt in the table and text. The paper seems unpolished.

7. A clear definition of the problem that is being solved is missing. It seems that the paper focuses on only class unlearning, but they present some results on random data forgetting in the experiments; yet, the extension of the method to unlearning a subset of the class is not clear and justified!

8. The setting of experiments for random data forgetting is not clear at all! What percentage of data are you forgetting in the results of table 3? Are they randomly chosen across different classes? This is also the same for the classes. What is the number of classes you try to unlearn? In most class unlearning works, the main initial focus is to show that one class can be unlearned effectively and then do experiments on multiple classes. In this work authors treat classes almost as individual samples by performing bulk forgetting on multiple classes simultaneously.

9. The computation time for some of the methods seems a bit counterintuitive. For example the underlying approach for BS is similar to RL but instead of random labels it finds the labels from adversarial attacks. So the fact that the time reported for BS is much smaller than RL in figure 4 is a bit counter intuitive.

10. The SVM-based MIA that is used for evaluations is very weak compared to the SOTA MIA methods in the literature. I encourage the authors to utilize SOTA MIA for their evaluations rather than relying on basic approaches. [1] introduced is one of the SOTA MIAs that is an adaptation of  [3], but more practical with a few shadow models. There are also MIAs designed specifically to evaluate unlearning methods [2,3].

11. There are more recent works on machine unlearning for classification models that have not been used as base-lines in the experiments. Please see [4,5,6,7,8].


[1] Zarifzadeh, S., Liu, P., & Shokri, R. (2024, July). Low-cost high-power membership inference attacks. In Proceedings of the 41st International Conference on Machine Learning (pp. 58244-58282).

[2] Hayes, J., Shumailov, I., Triantafillou, E., Khalifa, A., & Papernot, N. (2025, April). Inexact unlearning needs more careful evaluations to avoid a false sense of privacy. In 2025 IEEE Conference on Secure and Trustworthy Machine Learning (SaTML) (pp. 497-519). IEEE.

[3] Cadet, X. F., Borovykh, A., Malekzadeh, M., Ahmadi-Abhari, S., & Haddadi, H. (2025, June). Deep Unlearn: Benchmarking Machine Unlearning for Image Classification. In 2025 IEEE 10th European Symposium on Security and Privacy (EuroS&P) (pp. 939-962). IEEE.

[4] Zhang, B., Dong, Y., Wang, T., & Li, J. Towards Certified Unlearning for Deep Neural Networks. In Forty-first International Conference on Machine Learning.

[5] Cha, S., Cho, S., Hwang, D., Lee, H., Moon, T., & Lee, M. (2024, March). Learning to unlearn: Instance-wise unlearning for pre-trained classifiers. In Proceedings of the AAAI conference on artificial intelligence (Vol. 38, No. 10, pp. 11186-11194).

[6] Bonato, J., Cotogni, M., & Sabetta, L. (2024, September). Is retain set all you need in machine unlearning? restoring performance of unlearned models with out-of-distribution images. In European Conference on Computer Vision (pp. 1-19). Cham: Springer Nature Switzerland.

[7] Kodge, S., Saha, G., & Roy, K. (2024). Deep unlearning: Fast and efficient gradient-free class forgetting. Transactions on Machine Learning Research.

[8] Ebrahimpour-Boroojeny, A., Sundaram, H., & Chandrasekaran, V. Not All Wrong is Bad: Using Adversarial Examples for Unlearning. In Forty-second International Conference on Machine Learning.



### Minor comments:


- In line 126 authors point out that KD is a necessary extension for MU evaluation, but they do not provide any definition or clarification for KD. At least refer to the appendix section containing the details.

- Duplicate sentence in line 717/718.

**Questions:**

1. In the random data forgetting setting, consider a class where some of its samples are supposed to be unlearned. Do you find one prototype for the samples that are supposed to be forgotten and one prototype for the remaining ones? How much do these prototypes differ from each other? If these points are chosen uniformly at random the average embedding of the two sets almost overlap with each other. How would it be possible to remap the forget set while retaining the model's accuracy on the remaining set?

2. What is the authors’ justification for using the mean activation vectors? Figure 1, shows the motivating example that the latent features lead to separable clusters for different classes. In that figure it seems the remapping is applied to each sample (each point in the figure) to assign it to various parts of the embedding space. However, from the Algorithm 1, that does not seem to be what the method does on individual samples, but instead it only remaps the average embedding of all the samples in one class to another part of the embedding space, which is expected to transfer all the constituting samples to that region as well, leading to a cohesive structure again. Could the authors discuss this further?


3. Why does random routing often outperform conditional? Can you diagnose expert collapse? If your conditional routing doesn’t do better than random there is no need to present it as a contribution.

---

> ### Author Response · Authors · 2025-11-27
> **Rebuttal by Authors (1/5)**
>
> # Opening remarks
>
> We are deeply grateful to the reviewer for the time, thoughtful analysis, and constructive suggestions. Your feedback has been immensely helpful in sharpening the clarity of our presentation and highlighting areas for further elaboration. We have taken your comments seriously and made concrete efforts to address each point raised as faithfully as possible.
>
> Below, we provide detailed, point-by-point responses to the identified weaknesses (W#) and questions (Q#). For multi-part bullets, we divide them into subpoints (e.g., W1a, W1b) to ensure clarity and precision in our replies.
>
> We have also included several **additional figures/tables in our response using anonymous external links**. Please note that depending on your browser or PDF viewer, some **fonts or formatting may appear distorted**. If this occurs, we kindly ask you to either **download the PDF directly or click “View Raw”** to ensure proper rendering.
>
> # 8wvp_W1 The impact of number of experts on unlearning performance
>
> Thank you for raising this great point and for giving us the opportunity to clarify. **We agree that the current presentation in Figure 6 and Table 11 could mislead readers** into interpreting that increasing the number of experts does not contribute to performance, and we appreciate the chance to explain this more clearly.
>
> **The ablation results in Figure 6 and Table 11 are intended to show that our method is robust to the choice of the number of experts**, rather than to suggest that performance improves indefinitely as more experts are added. The key evidence for the benefit of using multiple experts is the contrast between a single expert and multiple experts, as shown in Table 1. The single-expert baseline (“Remap”) performs substantially worse than MoRE.
>
> **Once a sufficient number of experts is available, further increases exhibit diminishing returns.** To make this behavior clearer, we conducted an additional experiment on CIFAR-100 with ResNet-18, varying the number of experts with finer granularity, from 1 to 10. As shown in [Figure R1_1](https://anonymous.4open.science/r/MoRE-ICLR2026-Rebuttal-21A4/8wvp/Figure%20R1_1.pdf), performance improves sharply when transitioning from one expert to two experts and saturates around four experts. Beyond this point, adding more experts yields no meaningful improvement.
>
> **This behavior aligns with the intuition behind our design.** The objective is to break the cohesive structure of forgotten features by mapping them into several distinct regions in feature space. Once this structure is sufficiently disrupted, dividing samples across many more regions provides limited additional benefit.
>
> **Action plan**: We will revise the manuscript to include additional results with finer control over the number of experts, in order to more clearly illustrate the performance trend and the point at which adding experts no longer provides benefits. We will also include results for tiny-imagenet-200 as soon as the results are made available.

---

> ### Author Response · Authors · 2025-11-27
> **Rebuttal by Authors (2/5)**
>
> # 8wvp_W2+Q1+W7 Random data forgetting and problem definition
>
> Thank you for these closely related comments. The reviewer’s intuition in Q1+W2 is correct: in random instance-level forgetting, when forget and retain samples are drawn from the same class, their average embeddings can be very close. **This is precisely why we do not observe degradation in forget accuracy in this setting** (Table 3 in the original manuscript). Instead, the effect of our method appears primarily in privacy metrics, where we achieve the second-best average gap.
>
> We also take this opportunity to clarify the intent of this experiment in response to W7. As you have correctly noted, **our method is fundamentally designed for class-wise unlearning, which is the central problem we address**. The random forgetting experiment is included only as an auxiliary stress test to demonstrate broader applicability, not as a primary claim of capability.
>
> **In retrospect, we agree that the inclusion of random instance-level forgetting may distract from the main scope of the paper and could confuse readers about our intended problem setting. We take this feedback seriously and will therefore (1) consider to move the random forgetting results to the appendix, and (2) add a dedicated problem definition section to clearly state class-wise unlearning as the primary target.**
>
> **To instead demonstrate broader applicability in a more aligned setting, we will include results on concept unlearning in text-to-image generation model, namely Stable Diffusion v1.4**. Concept unlearning is closely related to class-wise unlearning and is especially relevant in the era of generative models. We believe this extension is more appropriate and impactful.
>
> Following prior works [6, 7], we apply prototype orthogonalization, erasure, and remapping to the cross-attention layers, using tokenized input prompts to construct prototypes. Our experimental setup also strictly follows the standard practice established by the SOTA diffusion unlearning methods [3,5–7], ensuring a fair comparison.
>
> **We evaluate unlearning performance in the artistic style erasure task**, which has emerged as a standard benchmark for testing concept-level unlearning in generative models. Following prior works [5, 6], we construct an evaluation set using 20 prompts for each of 10 artists: 5 classical (Van Gogh, Pablo Picasso, Rembrandt, Andy Warhol, Caravaggio) and 5 modern (Kelly McKernan, Thomas Kinkade, Tyler Edlin, Kilian Eng, and the anime series Ajin: DemiHuman), all of whom are reported to be mimicked by SD. We apply MoRE to remove two artistic styles: Van Gogh and Kelly McKernan. And the unlearning performance is measured using the LPIPS scores [8], which compares the generated outputes before and after unlearning to measure visual similarity (lower means more similar). We report three metrics:
> - **LPIPS_f (forget artists)**: LPIPS score computed on the forget artists (higher is better)
> - **LPIPS_r (remain artists)**: LPIPS score on the remain artists (lower is better, indicating minimal disruption)
> - **LPIPS_d = LPIPS_f − LPIPS_r**: the overall tradeoff, capturing how well the method removes target styles while preserving unrelated ones
>
> As shown in [Table R1_1](https://anonymous.4open.science/r/MoRE-ICLR2026-Rebuttal-21A4/8wvp/Table%20R1_1.pdf), **our proposed method achieves highly competitive performance** across all three LPIPS-based metrics: demonstrating strong unlearning of the target style (high LPIPS_f), minimal distortion to remain styles (low LPIPS_r), and the best overall tradeoff (highest LPIPS_d). Qualitative results shown in [Figure R1_2](https://anonymous.4open.science/r/MoRE-ICLR2026-Rebuttal-21A4/8wvp/Figure%20R1_2.pdf) further support these findings. Notably, **ours is the only method that successfully removes Van Gogh’s iconic artistic style while faithfully adhering to the input prompt**, generating a coherent image of a starry night over a town without reproducing the signature spiral patterns or brush stroke textures.
>
> **These results are highly significant; our proposed method is applied to diffusion models entirely out of the box, with no architecture-specific adaptation, no hyperparameter tuning and no additional engineering. Despite this, it outperforms SOTA diffusion model unlearning methods both quantitatively and qualitatively.** Since our current implementation does not yet leverage any diffusion-specific components, we believe that modest, targeted adaptations could further amplify its effectiveness. This opens up a highly promising avenue for future research, with the potential to drive substantial impact across both the generative modeling and unlearning communities.

---

> ### Author Response · Authors · 2025-11-27
> **Rebuttal by Authors (3/5)**
>
> # 8wvp_W3 Clarification on Figure 3
>
> Thank you for pointing this out and for giving us the chance to clarify this. We agree that our explanation in the main text did not make this distinction sufficiently clear.
>
> **Figure 3a and Figure 3b present two different types of correlations**. **Figure 3a shows the auto-correlation** of the prototypes themselves, and as the reviewer correctly noted, the maximum value is indeed 1 when a prototype is compared with itself. **Figure 3b, however, does not show auto-correlations**. It reports the cosine similarity between each class prototype and the class-wise activation mean after unlearning. Because these vectors are no longer identical after the update step, the similarities naturally fall below 1.
>
> The main takeaway from Figure 3a is that the forget-prototype and retain-prototype within a class are still strongly correlated. Consequently, if one were to directly apply unlearing method (as done in prior work [1]), the activation means of the remaining classes would be unintentionally perturbed. This is what leads to the sharp drop in cosine similarity before vs. after unlearning, which we highlight in the figure.
>
> **Action plan**: To prevent any further ambiguity, we will revise the manuscript in several ways:
> (1) We will expand the explanation in Section 3.1 to clearly separate prototype–prototype correlations from prototype–activation correlations.
> (2) We will adjust the terminology used in Figure 3 and the surrounding text to explicitly describe the comparison as being between activation means before and after unlearning.
> (3) We will avoid phrases such as “similarity between prototypes,” since we suspect this wording may have contributed to the misunderstanding and created room for misinterpretation. Instead, we will refer consistently to activation means where appropriate.
>
> # 8wvp_W4-6 Editorial Issues
>
> Thank you for carefully examining the manuscript and highlighting these inconsistencies. **We appreciate the level of detail in your reading, and we are sorry that parts of the notation made the paper feel unpolished and disrupted the reading experience**. While the other reviewers generally found the paper clear and easy to follow, we understand that these specific inconsistencies can still interrupt the flow, and we are grateful for the opportunity to correct them.
>
> **W4)** $\mathbf{s}_f$ in Line 247 is indeed a typo and should simply be $\mathbf{s}$. We will correct this in the revised version.
>
> **W5)** Thank you for this comment. We will define these early in the experimental section.
>
> **W6)** We also agree that the inconsistencies between the main body, tables, and captions may break the reader’s rhythm. We will unify all notation across the paper to make the writing more consistent and easier to navigate.
>
> **Action plan**: To further improve clarity, we will add a concise notation table at the beginning of the method section, so that readers can quickly reference all symbols and avoid ambiguity.
>
> # 8wvp_W8 Experiment Settings
>
> Thank you for raising these questions. We agree that the description of the experimental setup can be clarified further.
>
> **For the class-wise unlearning setting** (Line 319 of the original manuscript), we forget 10% of the classes. This corresponds to unlearning 1 class in CIFAR-10, 10 classes in CIFAR-100, and the proportional number for the remaining datasets. Please note that **these settings follow the widely used evaluation protocols in prior class-unlearning literature, and our experiments adhere to these established conventions.**
>
> **Regarding bulk vs. sequential class unlearning**, we apply unlearning to all target classes in a single run for computational efficiency, which is standard practice. For our method specifically, this choice does not affect the outcome because the prototypes for all classes are computed through a single forward pass before the unlearning update.  In either case, the method achieves strong results and delivers SOTA performance across the primary unlearning metrics.
>
> # 8wvp_W9 Computation time of BS vs RL
>
> We appreciate the insightful observation. As you correctly noted, BS involves computationally heavier adversarial steps per sample compared to RL. However, the **reported time difference stems from the different dataset size used during unlearning.**
>
> As described in the baselines section (B.2, Line 667), our implementation of **RL utilizes not only the forget set but also the remain set** to stabilize model utility. In contrast, **BS is optimized primarily on the forget set**. Since the remain set is significantly larger than the forget set, the cost of training on the large remain set in RL is higher than the attack cost in BS. This explains the longer runtime shown in Figure 4 in the original manuscript.

---

> ### Author Response · Authors · 2025-11-27
> **Rebuttal by Authors (4/5)**
>
> # 8wvp_W11 Additional experiments with the latest MU methods
>
> We thank the reviewer for raising this point regarding the breadth of baseline comparisons and pointing us to some of the recent works that slipped past our radar. We also appreciate the reviewer suggesting other evaluation metrics.
>
> **We have followed your suggestion and have added the five suggested unlearning methods to our evaluation suite**.  With these additions, our work now compares against 17 diverse unlearning baselines. Across all evaluations, our method consistently achieves superior performance in both forgetting and retention metrics (see [Table R1_2](https://anonymous.4open.science/r/MoRE-ICLR2026-Rebuttal-21A4/8wvp/Table%20R1_2.pdf)). **We believe this represents one of the most comprehensive empirical comparisons in the machine unlearning literature to date**. We hope this expanded benchmark provides a strong foundation for future studies and reinforces the practical and methodological advantages of our approach.
>
> # 8wvp_Q3 Stochastic vs. conditional router
>
> Thank you for this important question. We address it in two parts: (1) why random routing can outperform conditional routing, and (2) whether expert collapse occurs.
>
> **(1) Random routing vs. conditional routing**
> **Random routing is often more effective at erasing forget samples because it actively breaks the cohesive structure of their feature space**. Since each forgotten sample is mapped unpredictably to different experts, its representation becomes unstable across inferences, making it harder for an attacker to recover meaningful identity information.
>
> In contrast, conditional routing is deterministic. Once the router learns a consistent mapping for a forget sample, an adversary can more easily exploit this stability to recover patterns related to the forgotten data. In this sense, randomness directly strengthens the forgetting effect by preventing systematic feature alignment.
>
> **A second contributing factor is expert collapse**. During router training, we often observe that one expert becomes specialized for forget samples, causing the model to route nearly all forgotten features to the same expert. This unintentionally preserves a compact forget cluster in feature space, which weakens the unlearning effect.
>
> **Despite this, we include conditional routing because it is often better for preserving performance on remain data.** As shown in Table 5 of the original manuscript, conditional routing achieves higher retained accuracy in many cases. Random routing, while stronger at disrupting forget features, can also degrade performance on clean data since retained samples may be routed to suboptimal experts. Our design goal was to encourage uniformly distributed routing of forget samples while learning optimal paths for the retained set, but we found this balance difficult to achieve reliably in practice.
>
> **(2) Expert collapse**
>
> Yes, we can diagnose expert collapse, and this is documented in the Appendix D Table 14, measured using the Coefficient of Variation (CV). Our analysis shows that in conditional routing, forget samples are frequently mapped to a single dominant expert.

---

> > ### Author Response · Authors · 2025-11-27
> > **Rebuttal by Authors (5/5)**
> >
> > # Closing Remarks
> >
> > We sincerely thank the reviewer once again for the constructive feedback and the opportunity to improve our work. Below, we summarize the key changes made during this rebuttal:
> > - **Clarified our method's working mechanism, evaluation setup, and figures** such as justification for the use of activation mean, interpretation of figure 3 and 6 etc.
> > - **Added 5 new baselines for thorough evaluation** and showed that our method consistently performs better. With these additions, our work now compares against 17 diverse unlearning baselines, and represents one of the most comprehensive empirical comparisons in the machine unlearning literature to date.
> > - **Demonstrated broader applicability by applying our method directly to a large generative model (Stable Diffusion v1.4)** without any architectural modifications or hyperparameter tuning, achieving SOTA performance.
> >
> > These additions highlight the generality, scalability, and practical impact of our method across multiple tasks and model types. We believe we have addressed all major concerns raised in the reviews, and we do not see any remaining issues that should prevent an upward adjustment of the current evaluation. Thank you once again for your kind support and suggestions.
> >
> >
> >
> > *References*
> >
> > [1] Lee et al., "ESC: Erasing Space Concept for Knowledge Deletion," CVPR 2025.
> >
> > [2] Kumari et al., "Ablating concepts in text-toimage diffusion models," ICCV 2023
> >
> > [3] Gandikota et al., "Erasing concepts from diffusion models," CVPR 2023
> >
> > [4] Schramowski et al., "Safe latent diffusion: Mitigating inappropriate degeneration in diffusion models," CVPR 2023
> >
> > [5] Yoon et al., "Safree: Training-free and adaptive guard for safe text-to-image and video generation," ICLR 2025
> >
> > [6] Gong et al., "Reliable and efficient concept erasure of text-to-image diffusion models," ECCV 2024
> >
> > [7] Gandikota et al., "Unified concept editing in diffusion models," WACV 2024
> >
> > [8] Zhang et al., "The Unreasonable Effectiveness of Deep Features as a Perceptual Metric," CVPR 2018
> >
> > [9] Zhang et al., "Towards Certified Unlearning for Deep Neural Networks," ICML 2024
> >
> > [10] Cha et al., "Learning to Unlearn: Instance-wise Unlearning for Pre-trained Classifiers," AAAI 2024
> >
> > [11] Bonato et al., "Is Retain Set All You Need in Machine Unlearning?," ECCV 2024
> >
> > [12] Kodge et al., "Deep Unlearning: Fast and Efficient Gradient-Free Class Forgetting," TMLR 2024
> >
> > [13] Ebrahimpour-Boroojeny et al., "Not All Wrong is Bad: Using Adversarial Examples for Unlearning," ICML 2025

---

### Author Response · Authors · 2025-12-03
**Final remark from the authors (1/2)**

We would like to sincerely thank the Area Chair and the reviewers for the time and effort invested in evaluating our submission. Given the exceptional circumstances surrounding this review cycle, we recognize that an unusual burden now falls on the Area Chair to make decisions based solely on the written discussions between the author and the reviewers, without further interaction. To help reduce this burden and ensure that our rebuttal efforts are accurately reflected, **we provide below a compact summary of the major strengths and weaknesses identified by the reviewers and how they were addressed** during the discussion and revision process.

In addition, we highlight that **two reviewers (FRtt, m2ED) made the uncommon and explicit remark that they would be willing to revise their evaluations upward (potentially to scores of 8 or higher) post-rebuttal**, indicating that from the outset they viewed the work as fundamentally strong.

---

> ### Author Response · Authors · 2025-12-03
> **Final remark from the authors (2/2)**
>
> **Strengths.** The reviewers consistently recognized MoRE as a technically sound and well-motivated framework with strong empirical performance.
> - **Technically sound** and well-motivated, driven by the key limitation of the existing methods (8wvp, MYdz, FRtt, m2ED)
> - The proposed **idea is novel** and interesting (FRtt, m2ED)
> - The proposed method is **very efficient and scalable** to larger models and datasets (FRtt, m2ED)
> - **Strong empirical results** (FRtt, m2ED)
> - The paper is **well-written**, easy to follow, and clearly explained (MYdz, FRtt, m2ED)
>
> **Weaknesses.** The only recurring concern across reviews was the **scope of applicability** and the **breadth of baseline comparisons**. *(For each item below, the Area Chair can directly search or copy the corresponding “ReviewerID_Wx” tag in the rebuttal to locate our detailed response.)*
> - Applicability to larger models (MYdz_W2, FRtt_W5, m2ED_W2)
> - Applicability to highly-entangled settings (MYdz_W1)
> - Applicability to open-box setting (FRtt_W4, m2ED_W1)
> - Additional baselines for comparison (8wvp_W11, MYdz_W4)
>
> **Responses.** To address this directly, **we extend MoRE to concept unlearning in Stable Diffusion v1.4 and achieve SOTA performance** on the artistic style erasure task without any architectural modification, retraining, or diffusion-specific engineering. This experiment demonstrates the broader applicability and generality of our method, as well as its ability to operate in class-free tasks and remain robust under highly entangled representations.
>
> We further **clarify applicability in open-box settings** by showing that the linear projections introduced by MoRE can be absorbed into adjacent layer weights, and by demonstrating that our method can be applied to shallower layers of the feature extractor to mitigate knowledge leakage at earlier representations.
>
> To address the baseline concern, we have followed the reviewers’ suggestions faithfully and **added the five suggested unlearning methods to our evaluation suite**. With these additions, our work now compares against 17 diverse unlearning baselines. Across all evaluations, our method consistently achieves superior performance in both forgetting and retention metrics. We believe this represents one of the most comprehensive empirical comparisons in the machine unlearning literature to date.
>
> Other comments concerning the experimental setup and presentation were raised individually by reviewers but were not shared across reports. In response, we revised the manuscript extensively to improve clarity, consistency, and editorial quality.
>
> **Summary of our contributions.**
> - **We introduce two novel mechanisms** (prototype orthogonalization and mixture of remapping experts) that address the core limitations in existing machine unlearning methods: reversibility of existing unlearning methods and entanglement between forget and remain representations.
> - Using these ideas, we achieve **SOTA performance on class-wise unlearning** across 17 established baselines.
> - Extending our work to the concept unlearning task, namely artistic style erasure, we still achieve **SOTA performance among the 6 latest MU baselines on diffusion models**. We also demonstrate this qualitatively.
> - Despite the SOTA unlearning performance, our method remains **extremely efficient**, making it highly scalable to larger models and datasets.
>
> In summary, all reviewer concerns have been substantively addressed through additional experiments, clarifications, and revisions. The technical soundness of the approach is consistently affirmed by the reviewers, and the expanded evaluation confirms MoRE as a robust, efficient, and high-impact contribution to the machine unlearning literature. We do not see any lingering concerns that should preclude our paper from acceptance, and we believe this revised submission now meets the standard for archival publication.
>
> Thank you once again for your kind support and suggestions.
>
> &nbsp;
>
> Yours sincerely,
>
> The authors of MoRE

---

### Meta-Review · Area_Chair_KyAA · 2026-01-06

**Summary:**

Several reviewers raised concerns that the method does not reliably support instance-wise unlearning, especially when forget and retain samples come from the same class. The rebuttal largely acknowledges this limitation and reframes the method as class- or concept-level, but this effectively narrows the contribution rather than resolving the concern. If the paper continues to imply broad instance-level forgetting, the current results do not fully justify those claims.

There was also skepticism about the multi-expert (MoE) design. While the rebuttal explains why multiple experts help and why conditional routing may collapse, the empirical gains appear to saturate quickly and do not clearly validate conditional routing as a core contribution. This weakens the novelty of the MoE component and makes it feel more like an architectural variant than a substantive advance.

Another key issue is the lack of a principled analysis of representation entanglement and information leakage. The method relies on prototype-based orthogonalization, but reviewers noted that its behavior under strong entanglement is insufficiently characterized. The rebuttal provides additional experiments and intuition, but does not offer the quantitative leakage or entanglement analysis some reviewers explicitly requested.

Finally, while many presentation, clarity, and threat-model concerns were addressed in the rebuttal, these improvements do not overcome the more fundamental issues above. Overall, the paper presents an interesting and potentially useful idea, but the current evidence supports a narrower and weaker contribution than claimed, which is not enough to support acceptance recommendation.

**Reviewer Concerns:**

Several reviewer concerns were partially or fully addressed by the rebuttal, particularly those related to clarity, positioning, and threat model assumptions. The authors provided reasonable explanations for notation inconsistencies, clarified how the projection layer can be fused into model weights (mitigating bypass concerns), and better explained the runtime and memory comparisons. Questions about whether the method assumes a closed-box setting were also addressed convincingly by arguing that remapping can be absorbed into the model and applied at different layers. These responses meaningfully reduced confusion around implementation details and strengthened the practical framing of the method.

The rebuttal also partially addressed concerns about the multi-expert (MoE) design. The authors clarified that most gains come from moving from a single expert to multiple experts, and that performance saturates thereafter. They also explained why random routing can outperform conditional routing due to expert collapse and stochastic disruption. While this helps explain the empirical behavior, it does not fully resolve the concern that conditional routing is over-emphasized relative to its demonstrated benefits. This issue appears more like a positioning problem than a fatal flaw, but it still weakens the strength of the MoE contribution.

However, several core concerns remain outstanding, most notably those related to instance-wise unlearning. Multiple reviewers questioned whether the method can truly forget arbitrary instances within a class, especially when forget and retain samples are drawn from the same distribution. The rebuttal largely concedes this limitation, reframing the method as primarily class- or concept-level, and noting that random forgetting mainly improves privacy metrics rather than forget accuracy. This narrows the scope of the contribution but does not resolve the original concern if the paper still implies broad instance-level unlearning capability.

Another unresolved issue is the robustness of prototype-based orthogonalization under strong representation entanglement. While the authors argue that deeper layers are more separable and provide additional experiments (e.g., diffusion models) to suggest generality, they do not provide the kind of quantitative entanglement or leakage analysis some reviewers explicitly requested. As a result, the method's behavior in highly entangled settings remains insufficiently characterized.

Overall, the rebuttal improves clarity and addresses several secondary concerns, but the main limitations around instance-wise forgetting and representational entanglement remain largely unresolved.

**Reviewer Scores:**

Reviewer 8wvp raised the most fundamental concerns about instance-wise forgetting and whether the MoE routing is actually doing meaningful work. While the rebuttal clarified scope (mainly class-wise forgetting) and explained the MoE behavior, it did not really resolve the core conceptual limitation they pointed out. I think this reviewer would likely maintain roughly the same score 4. A meaningful score increase seems unlikely.

The rebuttal partially addressed Reviewer MYdz's concerns by adding diffusion experiments and arguing that deeper layers are less entangled, but it did not provide the explicit entanglement or leakage analysis they asked for. This reviewer might feel that their main question was not fully answered. I would expect no change on score (4).

Most of Reviewer FRtt's concerns were about positioning, threat model clarity, and whether the method is closer to unlearning or editing. The rebuttal handled these points fairly well and clarified several misunderstandings. However, the instance-wise forgetting concern still remains. I think this reviewer would likely keep the score (6).

For Reviewer m2ED, the rebuttal addressed the bypassability concern very effectively by explaining how the projection can be fused into weights. However, the instance-wise forgetting issue remained largely unresolved, and the reviewer explicitly followed up on that point. I would expect little (from 4 to 6) to no score change (4) here, as their main skepticism persists.

---

### Decision · Program_Chairs · 2026-01-26

Reject